# Inhibition of iRhom1 by CD44-targeting nanocarrier for improved cancer immunochemotherapy

Zhangyi Luo[1,2], Yixian Huang[1,2], Neelu Batra[3], Yuang Chen [1,2], Haozhe Huang[1,2], Yifei Wang[1,2], Ziqian Zhang [1,2], Shichen Li[1,2], Chien-Yu Chen[1,2], Zehua Wang [1,2], Jingjing Sun [1,2], Qiming Jane Wang [4], Da Yang [1,2], Binfeng Lu [5], James F. Conway [6], Lu-Yuan Li [2,7], Ai-Ming Yu [3] & Song Li [1,2] ✉

The multifaceted chemo-immune resistance is the principal barrier to achieving cure in cancer patients. Identifying a target that is critically involved in chemo-immune-resistance represents an attractive strategy to improve cancer treatment. iRhom1 plays a role in cancer cell proliferation and its expression is negatively correlated with immune cell infiltration. Here we show that iRhom1 decreases chemotherapy sensitivity by regulating the MAPK14-HSP27 axis. In addition, iRhom1 inhibits the cytotoxic T-cell response by reducing the stability of ERAP1 protein and the ERAP1-mediated antigen processing and presentation. To facilitate the therapeutic translation of these findings, we develop a biodegradable nanocarrier that is effective in codelivery of iRhom pre-siRNA (pre-siiRhom) and chemotherapeutic drugs. This nanocarrier is effective in tumor targeting and penetration through both enhanced permeability and retention effect and CD44-mediated transcytosis in tumor endothelial cells as well as tumor cells. Inhibition of iRhom1 further facilitates tumor targeting and uptake through inhibition of CD44 cleavage. Co-delivery of pre-siiRhom and a chemotherapy agent leads to enhanced antitumor efficacy and activated tumor immune microenvironment in multiple cancer models in female mice. Targeting iRhom1 together with chemotherapy could represent a strategy to overcome chemo-immune resistance in cancer treatment.

Resistance to cancer therapy such as chemotherapy and immunotherapy remains a challenge in curing cancer patients. The improvement of treatment regimens, especially through rational design of combination therapy, has resulted in advancement in overcoming chemoresistance[1,2]. The discovery of immune checkpoint molecules such as PD-1/PD-L1 and CTLA4 has also changed the landscape of cancer immunotherapy. However, only a small population of cancer patients benefit from immune checkpoint blockade (ICB)-based immunotherapy such as colorectal cancer (CRC) patients with deficient mismatch repair (dMMR)[3]. Strong interests remain in

[1]Center for Pharmacogenetics, Department of Pharmaceutical Sciences, University of Pittsburgh School of Pharmacy, Pittsburgh, PA, USA. [2]UPMC Hillman Cancer Center, University of Pittsburgh, Pittsburgh, PA, USA. [3]Department of Biochemistry and Molecular Medicine, University of California, Davis, School of Medicine, Sacramento, CA, USA. [4]Department of Pharmacology and Chemical Biology, University of Pittsburgh School of Medicine, Pittsburgh, PA, USA. [5]Center for Discovery and Innovation, Hackensack Meridian Health, Nutley, NJ, USA. [6]Department of Structural Biology, University of Pittsburgh School of Medicine, Pittsburgh, PA, USA. [7]Present address: State Key Laboratory of Medicinal Chemical Biology and College of Pharmacy, Haihe Laboratory of Cell Ecosystem, Tianjin Key Laboratory of Molecular Drug Research, Nankai University, Tianjin, China. ✉e-mail: sol4@pitt.edu

developing new mechanism-based therapies to benefit more cancer patients.

IRhom proteins (iRhom1 and iRhom2) are catalytically inactive relatives of rhomboid intramembrane proteases[4,5]. They are widely distributed in many tissues and play an important role in regulating the stability and trafficking of other membrane proteins. IRhom proteins have seven transmembrane domains and are predominantly localized in endoplasmic reticulum (ER)[6] but also distributed on the route of protein trafficking, including Golgi and cell surface[7,8]. They have been shown to regulate (enhance) the activity of ADAM17, a membrane-tethered metalloprotease and the primary shedding enzyme responsible for the release of the proinflammatory cytokine TNFα and several EGF receptor ligands[6,9–12]. IRhom proteins are also found to be involved in regulating the homeostasis and functions of other proteins such as HIF1α and JNK2/3[7,13,14]. IRhom1 is overexpressed in breast cancer (BC) and head and neck cancer, and knockdown (KD) of iRhom1 causes apoptosis or autophagy in epithelial cancer cells, and attenuation of fibrotic stroma formation through inhibiting the endothelial–mesenchymal transition[14–16]. Moreover, a negative correlation between CD8+ T cell infiltration and iRhom1 expression is shown in several types of cancers including BC and CRC[14]. Knockout (KO) of iRhom1 in 4T1 cells (murine BC cell line) is associated with delayed tumor growth and more infiltration of immune cells[14]. The potentially multifunctional role of iRhom in regulating both tumor survival and immune microenvironment suggests an effective therapeutic target for improving cancer treatment.

In this work, we investigate the mechanism of how iRhom1 regulates the antitumor immune response. The role of iRhom1 in chemosensitivity is also examined. Finally, systematic studies are conducted to develop and evaluate immunochemotherapy based on nanoparticle (NP)-mediated co-delivery of iRhom1 pre-siRNA and a chemotherapeutic drug.

## Results

### Role of iRhom1 in regulating sensitivity to chemotherapy

IRhom1 has previously been shown to be overexpressed in breast and cervical cancers and its expression is negatively correlated with patient prognosis[13,17]. Our analysis of the TCGA data shows that iRhom1 is also significantly upregulated in several other types of cancer including CRC, pancreatic cancer, and liver cancer (Supplementary Fig. 1A). The levels of iRhom1 expression are also negatively correlated with the clinical prognosis of those cancer patients (Supplementary Fig. 1B). More importantly, iRhom1 shows negative correlation with prognosis in those patient cohorts with chemotherapy treatment in both triple negative breast cancer (TNBC) and CRC (Fig. 1a), suggesting a likely role of iRhom1 in chemosensitivity.

As an initial step to understand the role of iRhom1 in regulating drug sensitivity, we analyzed the data in the Cancer Therapeutics Response Portal (CTRP)[18], a database of large-scale cancer cell line drug screening that correlates genetic biomarkers with the response to over 300 anti-cancer agents. Those agents are categorized by their mechanisms of action (protein target or activity) based on the Informer Set of CTRP[18]. This analysis shows that histone deacetylases (HDAC), microtube assembly, and topoisomerase inhibitors are the top ranked drugs whose sensitivity is likely to be regulated by iRhom1 (Fig. 1b). Therefore doxorubicin (DOX) and Vorinostat (SAHA)/camptothecin (CPT) were chosen as model drugs to test drug sensitivity as they are currently used in the treatment of BC and CRC, respectively. We also prepared a prodrug conjugate (CPT-SAHA) of CPT and SAHA to synergistically improve the anti-tumor activity[19] and to facilitate the incorporation of both drugs into a nanocarrier as detailed later. Indeed, cytotoxicity assays showed that iRhom1 siRNA KD led to sensitization of BC or CRC cells to DOX or CPT-SAHA (Fig. 1c). Interestingly, treatment of 4T1 or CT26 cells with DOX or CPT-SAHA also led to increased expression of iRhom1 in a dose-dependent manner (Fig. 1d).

These data suggest a role of iRhom1 in both intrinsic and possibly acquired drug resistance.

To gain more insights into the role of iRhom1 in oncogenesis and drug response, we generated CT26 and 4T1.2 iRhom1 KO sublines, respectively by using CRISPR-Cas9 technology. Similar to what was seen in the iRhom1 KD cells, iRhom1 KO led to sensitization to DOX or CPT-SAHA treatment (Supplementary Fig. 2A). Figure 1E shows the changes in several cancer hallmark signaling pathways from bulk RNAseq data following KO of iRhom1 in CT26 cells. A number of signaling pathways were downregulated including MAPK, Myc, Kras, and hypoxia while the apoptosis pathway was upregulated. This result is consistent with the notion that iRhom1 supports the survival and proliferation of cancer cells. The downregulation of major survival signaling pathways was further confirmed by Western blot (Supplementary Fig. 3A), and likely contributes to both slowdown of cell proliferation (Supplementary Fig. 3B) and enhanced drug sensitivity. Furthermore, iRhom1 KO led to decreased expression of phosphorylated MAPK14 (p-MAPK14) and significantly reduced phosphorylation of HSP27 (p-HSP27) (Fig. 1f), one of the MAPK14 (also called p38-α) substrates and a known player that causes chemoresistance[20,21]. In contrast, overexpression (OE) of iRhom1 in CT26 cells (Supplementary Fig. 4) led to increased levels of p-MAPK14 and p-HSP27 (Fig. 1g). We have further shown that treatment with CPT-SAHA (Fig. 1h) or DOX (Supplementary Fig. 2B) led to increased expression of p-HSP27 and p-MAPK14 in WT cells but not in iRhom1 KO cells (Supplementary Fig. 2). Re-expression of iRhom1 in iRhom1 KO cells rescued the chemoresistance (Supplementary Fig. 2d). Overexpression of MAPK14 in iRhom1−/− cells also partially attenuated the sensitivity of the cells to CPT-SAHA or DOX (Supplementary Fig. 2A). Figure 1i shows that iRhom1 OE in WT CT26 cells led to further increase in chemoresistance but this effect was significantly abolished by MAPK14 KD. Figure 1J shows that inhibition of phosphorylation of HSP27 by Ivermectin (IVM) treatment led to an overall significant synergistic effect with a chemotherapeutic agent in iRhom1 WT 4T1 and CT26 cells. However, this synergistic effect was significantly attenuated in iRhom1 KO cell lines, suggesting that MAPK14-HSP27 axis likely plays a role in iRhom1-mediated drug resistance. KD of iRhom2 also led to sensitization to chemotherapy drugs in CT26 or 4T1 cells (Supplementary Fig. 5), but to a lesser extent compared to iRhom1 KD. However, iRhom2 KD showed no impact on the expression levels of p-MAPK14 and p-HSP27 in CT26 or 4T1 cells (Supplementary Fig. 5). In addition, iRhom1 KO or KD did not cause significant changes in the protein expression levels of iRhom2 (Supplementary Fig. 6).

Searching of iRhom1-interactive partners in proteomic database[22,23] identified MAPK14 as one potential candidate that is regulated by iRhom1 through protein-protein interaction (PPI). The interaction of iRhom1 with MAPK14 or p-MAPK14 was further confirmed in a pull-down assay (Fig. 1k, Supplementary Fig. 7). MKK3/6 is an upstream kinase to phosphorylate MAPK14. IRhom1 KO did not affect the phosphorylation of MKK3/6 (Supplementary Fig. 8A). However, the interaction between pMKK3/6 and MAPK14 was disrupted in iRhom1 KO cells, suggesting that iRhom1 serves as a scaffold for MAPK14 phosphorylation (Supplementary Fig. 8B). MAPK14/HSP27 axis is implicated in both oncogenesis and chemoresistance[24,25]. Our data suggest that iRhom1 inhibits the drug response likely through regulating the iRhom1/MAPK14/HSP27 pathway. No interaction was observed between MAPK14 and iRhom2 (Supplementary Fig. 9).

### Role of iRhom1 in regulating immune response

It has been reported that iRhom1 expression level is negatively correlated with CD8+ T cell infiltration in patients with BC[14]. Interestingly, the expression levels of iRhom1 are lower in the cancer subtypes that are candidates for anti-PD1 immunotherapy such as TNBC or microsatellite instability-high (MSI-H) CRC compared to other subtypes

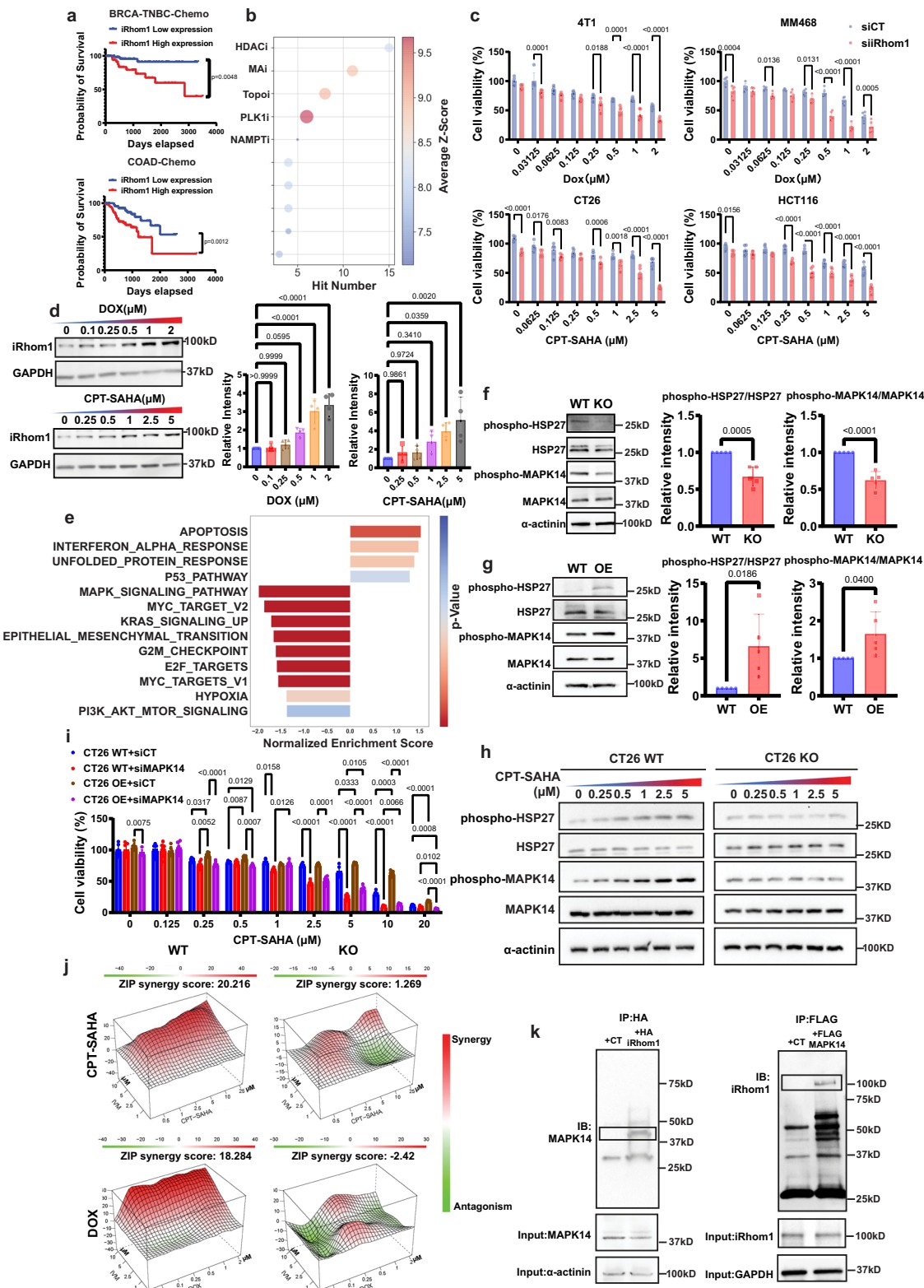

(Fig. 2a), suggesting a likely role of iRhom1 in antitumor immunity. As an initial step to gain insights into the role of iRhom1 in modulating immune responses, we examined the in vivo tumor growth rate of iRhom1 WT vs KO tumor cells. Consistent with slower proliferation in culture, iRhom1 KO CT26 cells formed smaller tumors compared to WT cells in immunodeficient mice (Fig. 2b). Interestingly, tumor formation was completely abolished in immunocompetent mice inoculated with iRhom1 KO CT26 cells (Fig. 2b). Re-expression of iRhom1 in iRhom1 KO cells (Rescue) led to regained tumor growth in vivo

(Fig. 2b). The incomplete rescue of tumor growth in iRhom1-reexpressed cells is likely attributed to the increased immunogenicity of CRISPR-engineered cells that involved the use of a lentiviral vector[26–28]. Similar results were observed in iRhom1 KO 4T1.2 cells (Supplementary Fig. 10). The above data suggest that the immuno-modulating effect of iRhom1 likely plays a more important role in controlling tumor growth in vivo. This is supported by RNAseq data showing upregulation of several immune pathways including enhanced interferon response and enhanced antigen presentation in

**Fig. 1 | IRhom1 is involved in chemoresistance. a** Correlation of iRhom1 gene expression levels with survivals in BRCA-TNBC or COAD patients receiving chemotherapy. For BRCA-TNBC-Chemo: n(Low expression) = 52, n (High expression) = 33; For COAD-Chemo: n (Low expression)=89, n(High expression) = 56. **b** CTRP-based prediction of the correlation between iRhom1 gene expression levels and the responses to different types of anticancer drugs. HDACi: histone deacetylase inhibitor, MAi: microtube assembly inhibitor, Topoi: topoisomerase inhibitor, PLK1i: polo-like kinase 1 inhibitor, NAMPTi: nicotinamide phosphoribosyltransferase inhibitor. **c** The impact of siRNA-mediated iRhom1 knockdown on the cytotoxicity of DOX or CPT-SAHA in several cancer cell lines. n = 6 independent samples. **d** Changes in protein levels of iRhom1 following treatment with DOX or CPT-SAHA. n = 4 independent experiments and data were quantified by densitometry. **e** RNA-seq analysis of gene enrichment in various signaling pathways in iRhom1 KO CT26 cells compared to WT cells. **f** Changes in the protein levels of MAPK14-HSP27 axis in iRhom1 KO cells compared to WT cells. n = 5 independent experiments and data were quantified by densitometry. **g** Changes in the protein levels of MAPK14-HSP27 axis in iRhom1 OE cells compared to WT cells. n = 5 independent experiments and data were quantified by densitometry.

**h** Changes in the protein levels of MAPK14-HSP27 axis after treatment with various concentrations of CPT-SAHA in iRhom1 KO cells and WT cells. n = 3 independent experiments and data were quantified by densitometry. **i** MTT cytotoxicity assay of treatments with various concentrations of CPT-SAHA in CT26 WT or CT26 iRhom1 OE cells, with or without knockdown of MAPK14 respectively. n = 6 independent samples. **j** Synergy between Ivermectin (IVM) and chemotherapy drug (CPT-SAHA or DOX). **k** Immunoprecipitation of HA-iRhom1 with anti-HA beads led to pulldown of p-38α (MAPK14) in HA-iRhom1 expressing 293 T cells and reciprocal immunoprecipitation of FLAG-MAPK14 with anti-FLAG beads led to pulldown of iRhom1 in FLAG-MAPK14 expressing 293 T cells. Data are presented as mean ± s.e.m. in (**c, i**) and mean ± s.d. in (**d, f, g**). Statistical analysis was performed by log rank test for comparison in (**a**), two-tailed Student's t-test for comparison in (**c, f, g**), and one-way ANOVA with Tukey's post hoc test for comparison in (**d** and **i**). Data are representative of two independent experiments in (**e**), **i–k** three independent experiments in (**c, h**), four independent experiments in (**d**), and five independent experiments in (**f, g**). Source data are provided as a Source Data file for (**c, d, f, g, h, i, j, k**).

the KO cells (Fig. 2c). To gain the mechanistic insights, we examined the impact of iRhom1 KO on antigen (Ag) presentation. B16-OVA cells express chicken OVA as surrogate tumor Ag (e.g., OVA-derived peptide SIINFEKL bound to MHC class I H-2Kb) and have been widely used to study Ag presentation[29]. We generated iRhom1$^{-/-}$ B16-OVA as described above. As shown in Fig. 2d, the presentation of SIINFEKL/MHC-I complex was significantly enhanced in iRhom1$^{-/-}$ cells. Importantly, the enhanced Ag presentation was significantly attenuated when iRhom1 expression was reconstituted in KO cells via transfection with an iRhom1 plasmid. Expression of iRhom2 showed no impact on the Ag presentation in iRhom1 KO cells (Supplementary Fig. 11). To examine the biological consequence of improved Ag presentation in iRhom1 KO cells, a cytotoxicity assay was conducted in which OVA-specific CD8$^+$ T cells (OT-1 T cells) were co-cultured with WT B16-OVA or iRhom1 KO B16-OVA cells. Increasing the OT-1 CD8 + T (effector cells) to B16-OVA (target cells) (E/T) ratio was associated with increased cytotoxicity towards both WT B16-OVA and iRhom1$^{-/-}$ B16-OVA cells (Fig. 2e). It was also apparent that OT-1 T cells exerted significantly more cytotoxicity on iRhom1 KO B16-OVA cells than on B16-OVA cells (Fig. 2e, Supplementary Fig. 12). In addition, the cytotoxicity on both cells was significantly attenuated by cotreatment with H2K-specific Ab. These data suggest that KO of iRhom1 in B16-OVA led to a significant increase in H2K-mediated presentation of OVA antigen to OT-1 T cells, resulting in enhanced cytotoxicity.

To look for potential candidate(s) in Ag processing and presentation (APP) machinery that is regulated by iRhom1, we analyzed the proteomics data of TNBC patients in Clinical Proteomic Tumor Analysis Consortium (CPTAC), focusing on the major components involved in APP pathway. Interestingly, among 7 molecules examined, ERAP1 stands out as the only one whose protein expression levels were significantly and inversely correlated with the expression levels of iRhom1 (Fig. 2f). ERAP1 plays an important role in producing the peptide Ag presented by MHC-I molecules[30]. It is involved in trimming the SIINFEKL precursors to become the final SIINFEKL peptide that subsequently forms complex with MHCI molecules[30,31]. ERAP1 is also reported to be a sheddase like ADAM17[32,33]. We hypothesized that ERAP1 may be similarly subjected to iRhom1-mediated regulation via PPI, which may play an important role in iRhom1-mediated immune response. Supplementary Fig. 13 shows that the enhanced antigen presentation in iRhom1 KO cells was abolished by siRNA-mediated ERAP1 KD. More importantly, iRhom1 KO showed no effect on the presentation of OVA peptide in B16 cells overexpressing the mature SIINFEKL peptide that does not need to be further processed by ERAP1, suggesting a key role of ERAP1 in iRhom1 KO-mediated enhancement in Ag presentation (Supplementary Fig. 14A).

KO of iRhom1 led to increased levels of ERAP1 (Fig. 2g), but not other APP-related proteins (Supplementary Fig. 14). In contrast,

iRhom1 OE resulted in downregulation of ERAP1, further validating the negative regulation of ERAP1 by iRhom1 (Fig. 2g). Figure 2h shows that an HA tag-specific antibody (Ab) effectively pulled down ERAP1 in 293 T cells transfected with HA-iRhom1 plasmid. Reciprocally, iRhom1 was pulled down by GFP-ERAP1, suggesting a likely PPI between iRhom1 and ERAP1. Endoplasmic-reticulum-associated protein degradation (ERAD) represents a critical pathway in regulating homeostasis and protein degradation in ER. IRhom1 has been reported to be involved in ERAD as a chaperone in regulating protein turnover[34,35]. Indeed, treatment with Eeyarestatin 1, an ERAD inhibitor, led to increased level of ERAP1 protein in iRhom1 WT cells and OE cells but not in iRhom1 KO cells (Fig. 2i). Furthermore, KO of iRhom1 resulted in increased protein stability of ERAP1 as shown in a cycloheximide chase assay (Fig. 2j, k). Taken together, the above data provides evidence that iRhom1 inhibits antigen presentation at least partly through regulating the ERAD-mediated degradation of ERAP1 in ER. KD of iRhom2 showed no impact on either the protein expression levels of ERAP1 or the efficiency of Ag presentation (Supplementary Fig. 15).

## Development of a PCL-CP-based nanocarrier for co-delivery of pre-siiRhom and a chemotherapy agent

Our data thus far as well as those from others suggest that iRhom1 may represent an attractive therapeutic target for cancer, especially in combination with other treatment such as chemotherapy. However, no small molecule inhibitors are currently available for iRhom1. Therefore, we developed a therapy based on a combination of siiRhom1 and chemotherapy drugs using a polymeric nanocarrier PEG-Chitosan-Lipid (PCL). A bioengineered iRhom1 pre-siRNA (pre-siiRhom) was used to achieve iRhom1 knockdown. Pre-siRNA is generated by fermentation and is biotransformed into mature siRNA upon intracellular delivery (Supplementary Fig. 16). Folded within cells and without chemical modification, the pre-siRNA better captures the safety profile of natural RNAs and has potential to scale up[36,37]. The PCL nanocarrier was designed to be equipped with several unique features (Fig. 3a): 1) A lipid motif was introduced to facilitate interactions with cell membrane and improve transfection[38] while also helps to improve the loading of hydrophobic/lipidic drugs such as DOX or CPT-SAHA into the lipophilic core; 2) Chitosan was chosen as the backbone to improve biodegradability; 3) CS (chondroitin sulfate, 10–30 K)/PEG2K-CS were used to coat the PCL micelles coloaded with DOX (or CPT-SAHA)/pre-siRNA to generate PCL-CP NPs with neutral or slightly anionic surface. CS, as a natural ligand of CD44, was also included to mediate active targeting of tumor cells and tumor endothelial cells (ECs) as CD44 is overexpressed in both types of cells. PEG2K-CS was included to minimize the "nonspecific" uptake by RES. DOX or CPT-SAHA could be effectively loaded into PCL micelles through hydrophobic interaction with lipid motif in PCL (Fig. 3b). PCL/DOX (CPT-

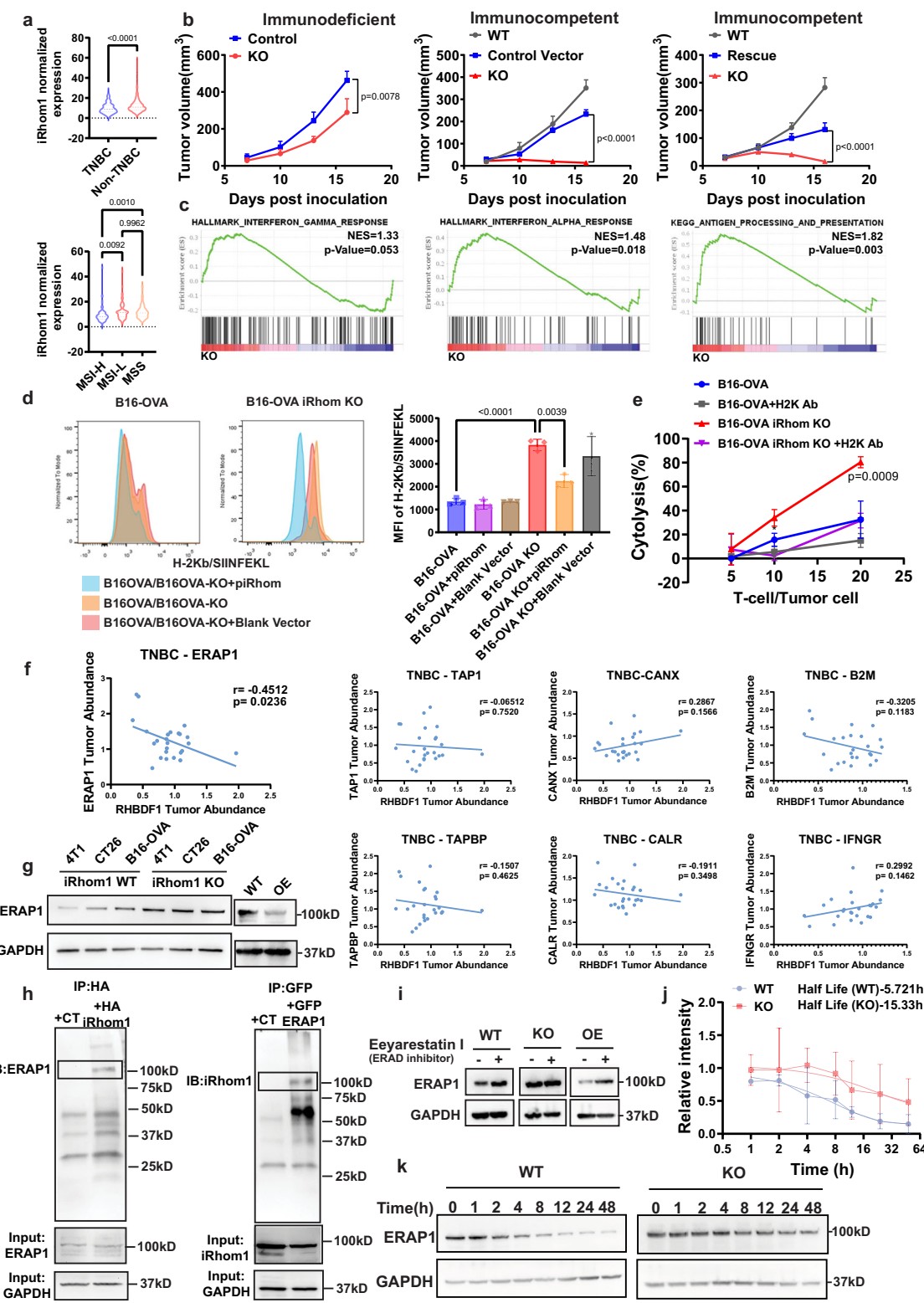

SAHA) readily formed complexes with pre-siRNA at various N/P ratios through charge-charge interaction between cationic amine group of PCL and anionic pre-siRNA (Fig. 3c). At a N/P ratio of 10/1, DOX/pre-siRNA-coloaded micelles were ~110 nm in size, smaller than that of PCL micelles loaded with DOX alone (170–190 nm) (Fig. 3c), suggesting that pre-siRNA could physically wrap around and stabilize the micelles. Subsequent coating with anionic CS/PEG-CS on the cationic surface of PCL/DOX(CPT-SAHA)/pre-siRNA micelles led to formation of PCL-CP NPs that gradually became anionic with increasing amounts of CS/PEG-

CS (Fig. 3d, e). Figure 3f shows a cryo-EM image of PCL-CP NPs at a N (nitrogen of PCL polymer)/P (phosphate of pre-siRNA)/S (sulfate of CS)/S (sulfate of PEG-CS) ratio of 10/1/1/0.5. Pre-siRNA was effectively loaded into NPs. Lack of fluorescence signal of pre-siRNA in the sample of pre-siRNA-loaded PCL or PCL-CP suggests the formation of tight PCL polymer/pre-siRNA complex, resulting in the exclusion of ethidium bromide from pre-siRNA (Fig. 3g). The final construct of PCL-CP/DOX/pre-siRNA exhibited low critical micelle concentration (CMC) (Supplementary Fig. 17), suggesting good stability after dilution in

**Fig. 2 | IRhom1 negatively regulates antigen presentation and antitumor immune response. a** TCGA analysis of iRhom1 gene expression levels in various cancer subtypes. TNBC: triple negative breast cancer; non-TNBC: breast cancer patients other than TNBC patients; MSI-H: microsatellite instability-high; MSI-L: microsatellite instability-low; MSS: microsatellite stable. $n$ (TNBC) = 115, n(non-TNBC) = 653, n(MSI-H) = 78, n(MSI-L) = 76, $n$(MSS) = 267. **b** Tumor growth curves of WT, iRhom1 KO, iRhom1 rescue and vector control CT26 cells in immunodeficient ($n$ = 4 mice) or immunocompetent mice ($n$ = 5 mice). **c** Activation of immune-related signaling pathways in iRhom1$^{-/-}$ CT26 cells (RNAseq analysis). **d** Flow analysis of H2Kb/SIINFEKL presentation on B16-OVA and iRhom1 KO B16-OVA (B16-OVA KO) cells, with or without re-expression of iRhom1 (+piRhom). $n$ = 3 independent samples. **e** Cytolysis of WT or iRhom1 KO B16-OVA cells co-cultured with CD8$^+$ T cells isolated from spleens of OT-I mice, with or without blockade by H2K antibody (H2K Ab). $n$ = 3 independent samples. **f** The correlation of protein expression levels between iRhom1 and various APP-related proteins in the tumor samples of TNBC patients. $n$ = 25. **g** Changes in ERAP1 protein levels in various cancer cell lines following iRhom1 KO and OE. **h** Co-Immunoprecipitation of HA-iRhom1 with anti-HA antibody led to pulldown of ERAP1 in HA-iRhom1 expressing 293 T cells and reciprocal immunoprecipitation of GFP-ERAP1 with anti-GFP beads led to pulldown of iRhom1 in GFP-ERAP1 expressing 293 T cells. **i** Changes in ERAP1 protein levels in iRhom1 WT, KO and OE cells after treatment with Eeyarestatin I (4.5 μM). **j** The stability ($t_{1/2}$) of ERAP1 in iRhom1 WT and KO cells determined by cycloheximide chase experiment. The experiment was repeated three times independently and data were quantified by densitometry. **k** Representative WB of ERAP1 protein levels in cycloheximide chase experiments. Data are presented as mean ± s.e.m. in (**b, d, e, j**). Statistical analysis was performed by two-tailed Student's $t$ test for comparison in (**a** (upper panel), **b** (immunodeficient)), and one-way ANOVA with Tukey's post hoc test for comparison in (**a** (lower panel), **b** (immunocompetent), **d** and **e**). Data are representative of two independent experiments in (**b, c, d, e, g–i**) and three independent experiments in (**j, k**). Source data are provided as a Source Data file for (**b, d, e, g, h, i, j, k**).

blood after i.v. administration. PCL-CP NPs showed slow kinetics of release of DOX or CPT-SAHA in PBS and the drug release was slightly accelerated in serum. Consistent with its effect in reducing the size of the NPs, pre-siRNA also slowed down the release of DOX in either PBS or serum (Fig. 3h). Interestingly, chitosan modified with PEG and lipid through a Schiff base remained sensitive to digestion by chitosanase and DOX (CPT-SAHA)/pre-siRNA-coloaded PCL-CP NPs rapidly became disassembled in the presence of chitosanase (Fig. 3i). In contrast, modification of chitosan via the commonly used amide bond[39] led to a drastic decrease in sensitivity to chitosanase. Therefore, our PCL-CP NPs retain the biodegradability of chitosan, which is critical for in vivo therapeutic application.

## PCL-CP carrier shows efficient tumor targeting and a favorable pharmacokinetics profile

Biodistribution of the PCL-CP NPs was first evaluated in an CT26 mouse tumor model by near infrared fluorescence (NIRF) imaging. DiR-loaded, Cy5.5-labeled PCL-CP NPs at an optimal N/P/S/S ratio of 10:1:1:0.5 showed high and concentrated signals in tumors for both DiR and Cy5.5 compared to other major organs, indicating that PCL-CP NPs were stable in blood and highly effective in tumor targeting (Fig. 4a). The pharmacokinetics and tissue distribution were further investigated for DOX and pre-siRNA by HPLC-fluorescence detection and q-PCR, respectively. DOX loaded in PCL-CP NPs showed significant increases over free DOX in both AUC and $t_{1/2}$, which demonstrates the excellent stability and long circulation time of the NPs in blood (Fig. 4b). Figure 4c, d shows that incorporation of DOX into the NPs led to a significant improvement in tumor accumulation with ~5.5 % of injected dose (ID) found in the tumors at 24 h following i.v. administration. Meanwhile, the distribution of DOX formulated in the NPs was decreased in the normal organs compared to free DOX. Similar results were shown for the PK and tissue distribution of pre-siRNA (Fig. 4e). The amounts of pre-siRNA accumulated in the tumors were ~7.21% of ID. Figure 4f shows that delivery of luciferase pre-siRNA (pre-siLuc) via PCL-CP NPs led to significant KD of the target gene expression as shown by drastically decreased luminescence in CT26-luc tumors.

## Role of CD44 in tumor endothelial cells (ECs) and tumor cells in the tumor accumulation and penetration of PCL-CP NPs

The effectiveness of tumor-targeting by our PCL-CP NPs is likely attributed to both EPR and CS/CD44-mediated active targeting. Our recent study suggested an important role of targeting of tumor ECs through CD44 in the overall tumor targeting[40]. To further elucidate the respective roles of CD44-mediated transcytosis in tumor ECs and tumor cells, we also generated CD44$^{-/-}$ HUVEC and CT26 cells, respectively (Supplementary Fig. 18). HUVECs cultured in the presence of growth factors express a relatively high level of CD44 and

have been used to model tumor ECs. We first established WT or CD44$^{-/-}$ tumors in either WT or CD44$^{-/-}$ mice and then investigated the tissue distribution of the NPs via both imaging and HPLC analysis. Figure 5a, b shows that KO of CD44 in mice led to a drastic reduction of Cy5.5 signals in both tumors and liver regardless of the CD44 status in the tumor cells. KO of CD44 in the tumor cells also caused deceases of the Cy5.5 signals in tumors in both WT and CD44$^{-/-}$ mice but to a much lesser extent. Fluorescence microscopic examination of tumor sections shows widespread distribution of Cy5.5 signals in the WT tumors grown in WT mice. The Cy5.5 signals in CD44$^{-/-}$ tumors grown in WT mice were largely confined to areas adjacent to blood vessels (CD31$^+$). KO of CD44 in mice led to drastic decreases in the Cy5.5 signals in the tumors, particularly the CD44$^{-/-}$ tumors (Fig. 5c). Similar results were shown in quantitative analysis of DOX distribution (Fig. 5d, e). Figure 5g, i shows the results of a transwell assay with both WT and CD44$^{-/-}$ HUVECs. Various Cy5.5-labeled NPs were added to the upper chamber and the samples were collected from the lower chamber at various time intervals (Fig. 5f). Significantly greater amounts of fluorescence signals were found in the lower chamber with the NPs coated with CS. The amounts of the signals were significantly decreased when the cells were co-treated with chlorpromazine, an endocytosis inhibitor, suggesting that the CS-decorated NPs are capable of crossing the EC through an active process of transcytosis. It should be noted that this process was essentially abolished in a transwell with CD44$^{-/-}$ HUVECs (Fig. 5I), suggesting a critical role of CS/CD44 interaction in the cellular uptake and transcytosis of CS-coated NPs. Similar results were observed in a transwell study with WT and CD44$^{-/-}$ CT26 cells (Fig. 5h, j). Figure 5k shows the results of NPs penetration in an CT26 tumorsphere. It is apparent that CS-coated but not non-coated NPs effectively penetrated and reached the core of the tumorsphere. Again, this process was significantly attenuated by an endocytosis inhibitor. Furthermore, the CS-mediated NP penetration was only seen in WT but not CD44$^{-/-}$ CT26 tumorsphere (Fig. 5l), suggesting a role of CS/CD44-mediated transcytosis in tumor penetration.

CD44-mediated tumor targeting has been studied for decades but the underlying mechanism remains elusive, especially the respective role of CD44 in tumor ECs and tumor cells. Most studies seem to suggest a mechanism that involves EPR followed by targeting to CD44 on tumor cells[41]. Our data suggest that the tumor endothelial CD44-mediated internalization and transcytosis play a more important role than either EPR or tumor cell CD44-mediated targeting in the total amounts of NPs accumulated in the tumor tissues (Fig. 5m). Following extravasation, the internalization and transcytosis that are mediated by the tumor cell CD44 contribute to the penetration of the CS-coated NPs in tumor tissues (Fig. 5m). This information may be highly significant for the future design of more effective tumor-targeting NPs.

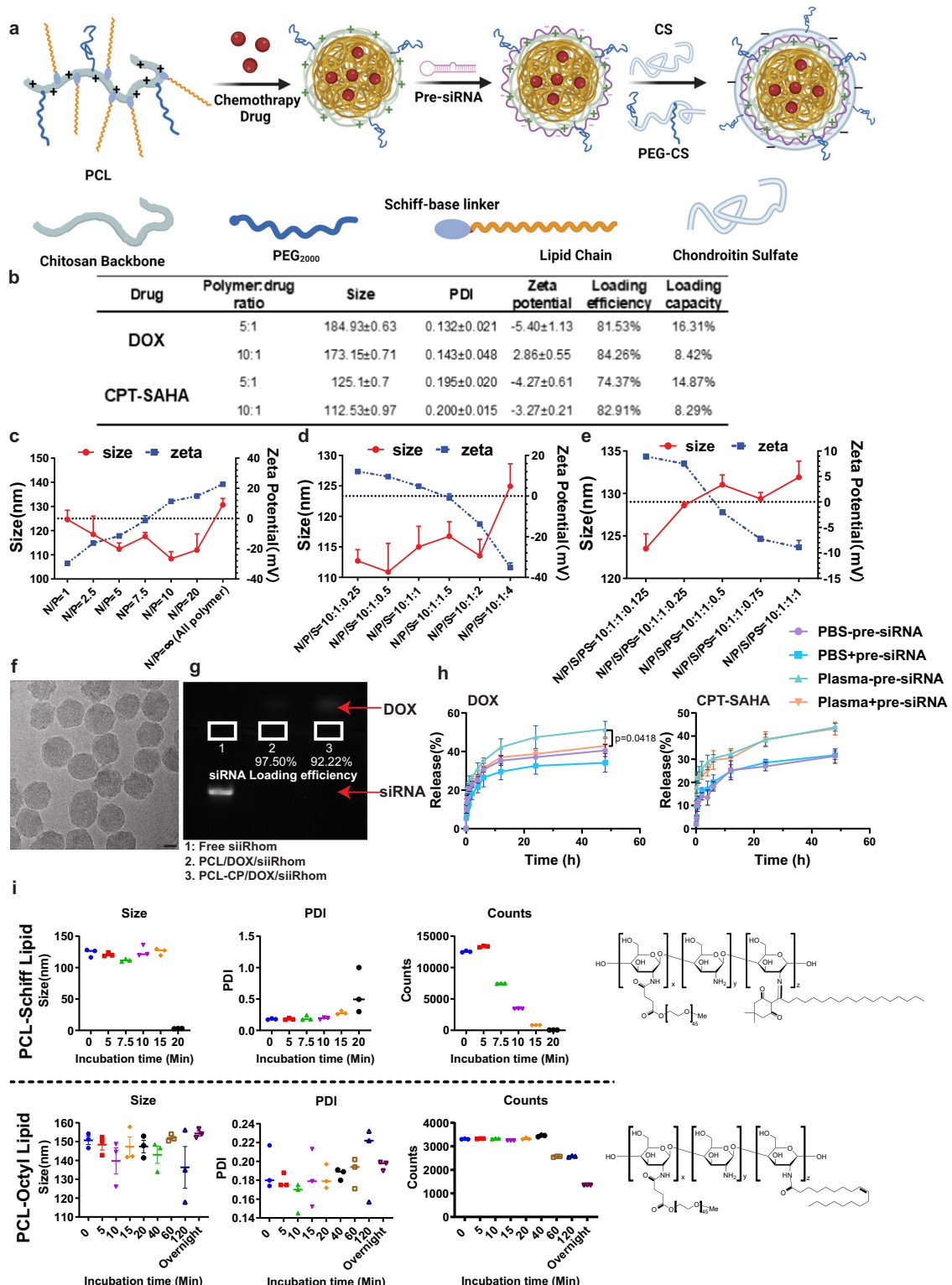

## Inhibition of iRhom1 further improves CD44-dependent tumor targeting by decreasing CD44 cleavage on the cell membrane

The extracellular domain that is responsible for CD44 ligand binding is constitutively cleaved at membrane-proximal region by metalloprotease in cancer cells[42]. However, its implication in CD44-mediated tumor targeting via nanomedicine has never been studied. IRhom1 is reported to regulate the trafficking of ADAM17, one of the major metalloproteases that are involved in CD44 cleavage[43,44]. To investigate whether iRhom1 KO affects CD44-mediated tumor targeting, cellular uptake of PCL NPs with or without CS coating was examined in

WT and iRhom1 KO or KD tumor cells. Figure 6a shows that iRhom1 KO or KD further improved CS/CD44-mediated cellular uptake. Similarly, iRhom1 KO further improved the CS/CD44-mediated transcytosis (Fig. 6b). NIRF imaging shows more signals of the NPs in iRhom1 KO tumors compared to WT tumors (Fig. 6c, d). Similar results were shown with quantitative analysis of DOX distribution in tumors (Fig. 6e). Figure 6f, g shows that the CD44 extracellular region was upregulated in iRhom1[−/−] CT26 cells compared to WT CT26 cells. Western blotting further showed the upregulation of the full length CD44 (CD44-full), along with downregulation of the cleaved membrane-associated CD44

**Fig. 3 | Biophysical characterizations of DOX (CPT-SAHA)/pre-siRNA-coloaded PCL-CP NPs. a** A schematic diagram of the protocol for the preparation of DOX (CPT-SAHA)/pre-siRNA-coloaded PCL-CP NPs. The figure is created with BioRender.com. **b** Biophysical characterization of DOX (CPT-SAHA)-loaded PCL micelles at various carrier/drug ratios (w/w). $n = 3$ independent samples. **c** Sizes and zeta potentials of PCL/DOX (CPT-SAHA)/pre-siRNA complexes at various N/P ratios. $n = 3$ independent samples. **d** Sizes and zeta potentials of DOX/pre-siRNA-coloaded PCL-C NPs (coated with CS alone) at various N/P/S ratios. $n = 3$ independent samples. **e** Sizes and zeta potentials of DOX/pre-siRNA-coloaded PCL-CP NPs (coated with a mixture of CS and PEG-CS) at various N/P/S(CS)/S(PEG-CS) ratios. $n = 3$ independent samples. **f** Cryo-EM characterization of DOX/pre-siRNA-coloaded PCL-CP NPs prepared at a carrier/drug ratio of 10:1 and a N/P/S/S ratio of 10:1:1:0.5. Bar: 20 nm. **g** Gel retardation assay of DOX/pre-siRNA-coloaded PCL-CP NPs. Carrier/drug ratio = 10:1, N/P/S/S = 10:1:1:0.5. **h** In vitro release of DOX or CPT-SAHA from PCL-CP NPs in PBS or plasma. $n = 3$ independent samples. Data are presented as mean ± s.e.m. *$p < 0.05$. **i** Chitosanase-mediated degradation of PCL derivatized with lipid with a Schiff-base or amide linker. $n = 3$ independent samples. Data are presented as mean ± s.e.m. in (**c**, **d**, **e**, **h**). Statistical analysis was performed by one-way ANOVA with Tukey's post hoc test for comparison in (**h**). Data are representative of two independent experiments in (**f**–**h**) and three independent experiments in (**b**–**e**, **i**). Source data are provided as a Source Data file for (**c**, **d**, **e**, **g**, **h**, **i**).

(CD44-EXT) in iRhom KO or KD cells, suggesting the inhibition of CD44 cleavage in iRhom1 KO or KD cells (Supplementary Fig. 19, Fig. 6h). As iRhom1 regulates ADAM17 activity in CD44 cleavage, we pretreated WT and iRhom KO or KD cells with TMI, an ADAM17 inhibitor. WT cells treated with ADAM17 inhibitor showed similar amounts of CD44-full compared to iRhom1 KO or KD cells, while ADAM17 inhibitor barely improved the CD44-full amount in iRhom1 KO or KD cells (Fig. 6h). Rescue of iRhom1 led to significant CD44 cleavage and this effect can be blocked by ADAM17 inhibitor (Fig. 6i). Figure 6j shows that cells treated with ADAM17 inhibitor mimicked the phenotype of iRhom1 KO or KD cells in cellular uptake. In addition, rescue of iRhom1 abolished the enhanced uptake (Fig. 6k). KD of iRhom2 exhibited similar phenotype as iRhom1 KO or KD (Supplementary Fig. 19B–D), which is not surprising considering the critical role of iRhom2 in mediating the maturation of ADAM17 as well. The above data suggests that iRhom1 can regulate CD44-mediated tumor targeting and transcytosis by affecting CD44 cleavage through iRhom1-ADAM17-CD44 axis.

### Co-delivery of pre-siiRhom and chemotherapy agent led to improved anti-tumor efficacy and enhanced antitumor immunity

Figure 7a shows the antitumor activity of different treatments in 4T1 orthotopic TNBC breast tumor model. NPs loaded with pre-siiRhom or DOX alone showed modest antitumor activity. Codelivery of pre-siiRhom and DOX via PCL-CP NPs led to significant improvement in therapeutic efficacy. IHC staining of tumor tissues showed upregulation of p-p38α and p-HSP27 following DOX treatment. This induction was significantly attenuated when DOX was co-delivered with pre-siiRhom (Supplementary Fig. 20), consistent with in vitro data (Fig. 1h). Similar results were also demonstrated with the NPs co-loaded with pre-siiRhom and CPT-SAHA in CT26 CRC model (Fig. 7d). All treatments were well tolerated as evident from similar changes in body weights over time compared to the control group (Fig. 7b, e). In addition, no obvious changes were seen in histology (Supplementary Fig. 21) and blood chemistry (Supplementary Fig. 22).

KD of iRhom1 via the NPs led to a significant improvement in tumor immune microenvironment as evident from increases in both the total numbers of CD8+ T cells and the numbers of functional (IFNγ+ or GzmB+) CD8+ T cells. Inclusion of DOX had no significant impact on these profiles (Fig. 7c). Interestingly, CPT-SAHA alone showed modest effect in upregulating the total numbers of CD4+ T cells, CD8+ T cells and the numbers of functional (IFNγ+ or GzmB+) CD8+ T cells (Fig. 7f, Supplementary Fig. 23). Combination of iRhom1 KD and CPT-SAHA led to further improvement in tumor microenvironment. Flow analysis of isolated tumor cells showed significant upregulation of MHC1 following treatment with pre-siiRhom or CPT-SAHA, especially the combination of both via NPs (Fig. 7g). Western blot analysis of tumor tissues showed upregulation of ERAP1 following treatment with pre-siiRhom, alone or in combination with DOX (Supplementary Fig. 24), suggesting that iRhom1 KD may similarly improve antitumor immune response through upregulation of ERAP1 in vivo.

Figure 7f shows that there was significant upregulation of PD-1 expression in CD8+ T-cells treated with CPT-SAHA, alone or together with pre-siiRhom, suggesting a potential of combining with anti-PD-1 to further improve the cancer treatment. Indeed, combination of CPT-SAHA/pre-siiRhom/PCL-CP and anti-PD1 antibody (aPD-1) led to significant improvement in the antitumor activity as demonstrated by significant prolongation of survival time. In addition, 2 out of the 8 mice in the combination group showed complete tumor eradication 7 days following the 5th treatment (Fig. 7h).

## Discussion

Using several cancer models, we have extended previous works on the potential role of iRhom1 in oncogenesis and drug response. In addition, we have shown that iRhom1 inhibits antitumor immune response by negatively regulating the stability of ERAP1 and the ERAP1-mediated Ag processing and presentation. In addition, chemotherapy-induced iRhom1 expression may contribute to immune escape through the iRhom1-ERAP1 axis, suggesting a mechanistic link between chemoresistance and immunoresistance.

Both iRhom1 and iRhom2 have long been known to regulate EGFR activity, a critical oncogenic signaling in multiple types of cancer, through regulating ADAM17 activity[11]. A role of iRhom1 and ADAM17 in oncogenesis and tumor progression has been well documented in BC[15,45]. A recent study by Freeman and colleagues shows that oncogenic KRAS mutants target the cytoplasmic domain of iRhom2 to induce ADAM17-dependent shedding and the release of ERBB ligands[46]. In addition, this mechanism is conserved in lung cancer cells, where iRhom activity is required for tumor growth in vivo[46]. Despite the substantial overlaps in tissue distribution and activity, iRhom1 and iRhom2 do have differences in their physiological functions[35]. Our preliminary data showed that iRhom2 KD had no impact on the expression level of phosphorylated MAPK14 despite the fact that iRhom2 KD also caused increased drug sensitivity. In addition, iRhom2 does not appear to be involved in ERAP1-mediated Ag presentation either by itself or following iRhom1 KO. More studies are needed to address the role of iRhom2 in regulation of drug sensitivity and cancer immune response as well. It would also be interesting to study the impact of simultaneous KO of iRhom1 and iRhom2 in the context of cancer treatment.

MAPK14/HSP27 axis has been reported to be involved in drug resistance to cisplatin, 5-fluorouracil, taxanes and DOX in BC, CRC and lung cancer[24,47–51]. However, the underlying mechanisms of how MAPK14/HSP27 is regulated remain to be further elucidated. IRhom1 KO or KD resulted in significantly decreased phosphorylation of both MAPK14 and HSP27, which was correlated with increased sensitivity to DOX and CPT-SAHA in 4T1 and CT26 cells, respectively. Importantly such changes were significantly attenuated when iRhom1 expression was reconstituted via transient transfection. In addition, OE of iRhom1 led to decreased drug sensitivity in both 4T1 and CT26 cells. Interestingly, treatment with DOX or CPT-SAHA led to a further increase in iRhom1 expression, suggesting a likely role of iRhom1 in regulating both intrinsic and acquired resistance. The detailed mechanism of how iRhom1 regulates the level of p-MAPK14 is not clear at present. Our preliminary data show that iRhom1 KO does not affect the phosphorylation of the upstream kinase MKK3/6 but disrupts the interaction of

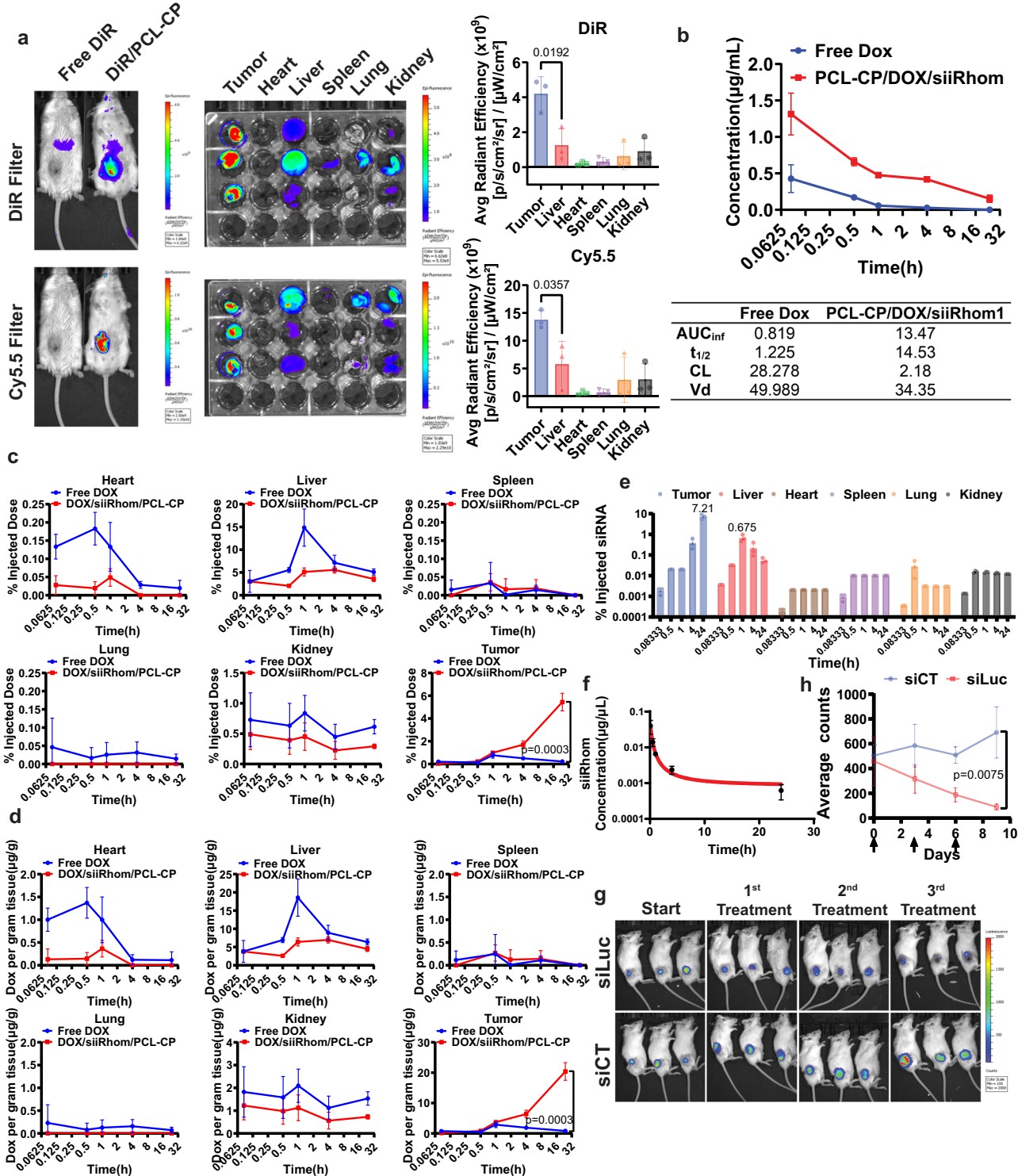

**Fig. 4 | PCL-CP NPs show efficient tumor targeting with limited liver accumulation. a** NIRF whole body imaging and ex vivo imaging of CT26 tumor-bearing mice at 24 h following i.v. administration of DiR-loaded, Cy5.5-labled PCL-CP NPs. $n = 3$ animals. **b** Plasma PK profile of free DOX or DOX loaded in DOX/pre-siRNA-coloaded PCL-CP NPs after i.v. administration. Non-compartment model was applied to generate the major pharmacokinetic profiles. AUCinf: area under the plasma concentration-time curve from time 0 to infinity; $t_{1/2}$ half-life; CL clearance; Vd volume of distribution. $n = 3$ biologically independent samples. **c** Biodistribution of free DOX or DOX loaded in DOX/pre-siRNA-coloaded PCL-CP NPs after i.v. administration (presented as percentage of injected dose). $n = 3$ biologically independent samples. **d** Biodistribution of free DOX or DOX loaded in DOX/siRhom/PCL-CP NPs after i.v. administration (presented as DOX concentration normalized by tissue weight). $n = 3$ biologically independent samples. The biodistribution profile (**e**) and the plasma PK (**f**) of siRNA loaded in DOX/pre-siRNA-coloaded PCL-CP NPs after i.v. administration. $n = 3$ biologically independent samples. **g** Mice bearing CT26-luc tumors received i.v. administration of luciferase siRNA-loaded PCL-CP NPs at a dose of 2 mg siRNA/kg once every 3 days and the mice were subjected to whole body bioluminescence imaging the next day after each treatment. **h** The efficiency of knockdown after each treatment was also analyzed quantitatively. $n = 3$ animals. Data are presented as mean ± s.e.m. in (**a**, **b**, **c**, **d**, **e**, **h**). Statistical analysis was performed by one-way ANOVA with Tukey's post hoc test for comparison in (**a**), and two-tailed Student's $t$ test for comparison in (**c**, **d**, **h**). Data are representative of two independent experiments in (**a**–**g**). Source data are provided as a Source Data file for (**c**, **d**, **e**, **f**, **h**).

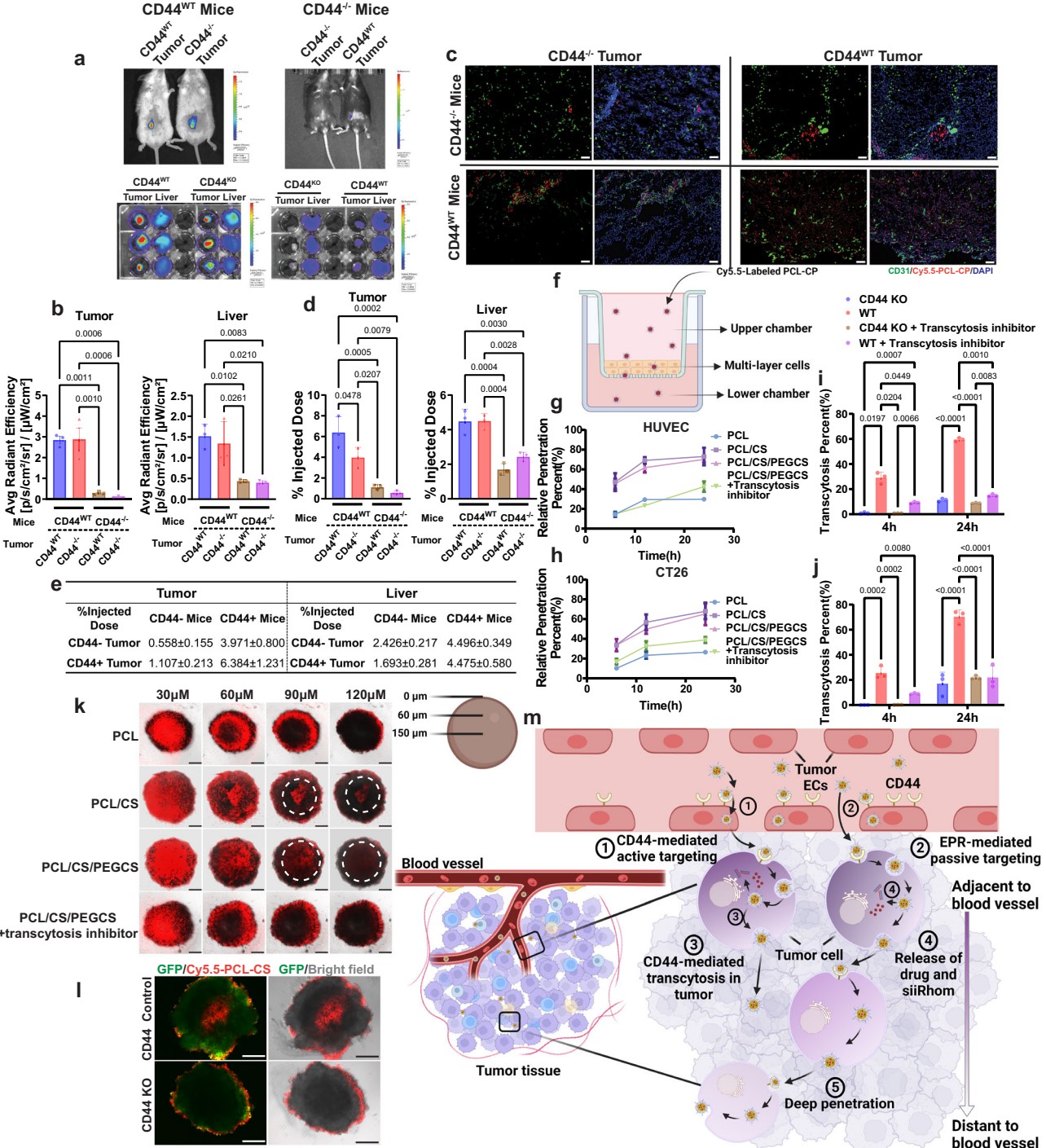

**Fig. 5 | PCL-CP NPs target and penetrate tumor through CD44 mediated transcytosis. a** NIRF whole body imaging, ex vivo imaging of CD44 WT or KO tumor-bearing CD44 WT or KO mice at 24 h following i.v. administration of DiR-loaded, Cy5.5-labled PCL-CP NPs. **b** Quantification of ex vivo imaging data from (**a**). n = 3 animals. **c** Fluorescence images of frozen tumor core sections from CD44 WT or KO tumor grown on CD44 WT or KO mice at 24 h after treatment with Cy5.5-labled PCL-CP NPs. CD31 was stained with FITC-labeled antibody to show the endothelial cells. Bar = 50 μm. **d** Biodistribution of DOX loaded in DOX/pre-siRNA-coloaded PCL-CP NPs in CD44 WT or KO tumor-bearing CD44 WT or KO mice at 24 h following i.v. administration. n = 3 independent samples. **e** Quantitative analysis of biodistribution of DOX in (**d**). n = 3 independent samples. **f** Illustration of transwell study. The figure is created with BioRender.com. Transwell assay of transmigration of Cy5.5 labeled PCL NPs, PCL-C NPs or PCL-CP NPs (with or without pretreatment with a transcytosis inhibitor) across HUVEC cells (**g**), CT26 cells (**h**), CD44 KO HUVEC cells (**i**), or CD44 KO CT26 cells (**j**). Fluorescence intensity of the

medium in the lower chamber was measured to calculate the percentage of transmigration at indicated time points. n = 3 independent experiments. **k** Confocal z-stack images of CT26 tumor cell spheroids after 18 h incubation with Cy5.5-labeled PCL NPs, PCL-C NPs or PCL-CP NPs (with or without pretreatment with a transcytosis inhibitor). White circles highlight the efficient penetration of CS-coated NPs. Bar = 150 μm. **l** Confocal z-stack images of the middle sections of CD44 WT or KO CT26-GFP spheroids after 18 h incubation with Cy5.5-labeled PCL-CP NPs. Bar=200 μm. **m** Proposed role of the CD44 in tumor ECs and tumor cells in mediating tumor targeting and penetration of CS-coated NPs. The figure is created with BioRender.com. Data are presented as mean ± s.e.m. in (**b, d, g, h, i, j**). Statistical analysis was performed by one-way ANOVA with Tukey's post hoc test for comparison in (**b, d, i, j**). Data are representative of two independent experiments in (**a**–**e, k, l**) and three independent experiments in (**g**–**j**). Source data are provided as a Source Data file for (**b, d, e, g, h, i, j**).

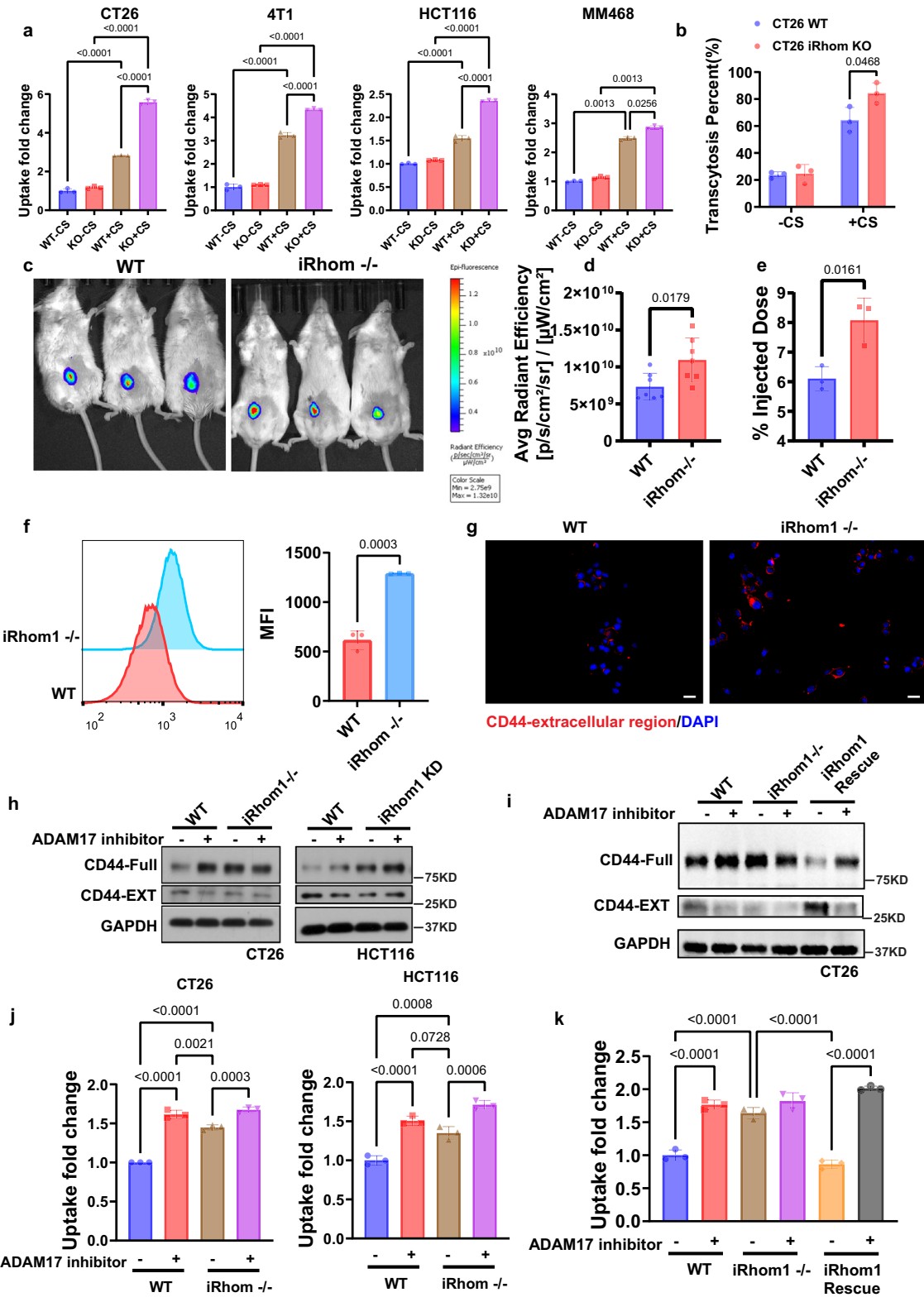

pMKK3/6 and MAPK14. It is possible that iRhom1 serves as a scaffold to facilitate the phosphorylation of MAPK14 or improve the stability of p-MAPK14. It should be noted that iRhom1 KO also led to upregulation of other signaling pathways such as AKT that may also play an important role in cancer cell survival and drug resistance[52]. However, the iRhom1/MAPK14/HSP27 is regulated, at least in part, through a mechanism that is independent of AKT signaling (Supplementary Fig. 25). More studies are needed to further define the underlying mechanism of iRhom1-mediated drug resistance.

IRhom1 KO also led to activation of several immune pathways including antigen processing and presentation (APP) pathway (Fig. 2c). Interestingly, despite upregulation of several APP-related genes at transcriptional level in RNAseq and qRT-PCR, only ERAP1 showed significant upregulation at protein level following iRhom1 KO (Fig. 2g, Supplementary Fig. 11B). Regulation of ERAP1 protein level by iRhom1 was further confirmed by the data that OE of iRhom1 led to decreased protein level of ERAP1 (Fig. 2g). The PPI between iRhom1 and ERAP1 was also confirmed by pull-down assays (Fig. 2h). IRhom1 appears to

**Fig. 6 | Inhibition of iRhom1 further promoted the CD44-mediated tumor targeting through reduced CD44 cleavage. a** Cellular uptake of Cy5.5-PCL NPs with or without CS coating in various WT or iRhom1 KO cell lines. $n = 3$ independent samples. **b** Transwell assay of transmigration of Cy5.5-PCL NPs with or without CS coating across WT or iRhom1 KO CT26 cells. n = 3 independent samples. **c** NIRF whole body imaging of WT or iRhom1 KO tumor-bearing mice following i.v. administration of DOX-loaded, Cy5.5-PCL-CP NPs and **d** the corresponding quantification. n = 7 animals. **e** Quantitative analysis of biodistribution of DOX in WT or iRhom1 KO tumor-bearing mice (WT). $n = 3$ independent samples. **f** CD44 expression levels measured by flow cytometry by staining with an antibody recognizing the CD44 external region. $n = 3$ independent samples. **g** Fluorescence images of CD44 external region expression in WT or iRhom1 KO CT26 cells. Bar = 20 μm. **h** Changes in protein levels of full-length CD44 (CD44-Full) and the cleaved, CD44 membrane-bound fragment (CD44-EXT) in WT or iRhom1 knockout/knockdown cells, with or without ADAM17 inhibitor pretreatment. **i** Changes in protein levels of CD44-Full and the CD44-EXT in WT, iRhom1 knockout and iRhom1 rescue cells, with or without ADAM17 inhibitor treatment. **j** Cellular uptake of Cy5.5 labeled PCL-CS NPs in WT or iRhom1 knockout/knockdown cells with or without ADAM17 inhibitor pretreatment. $n = 3$ independent samples. **k** Cellular uptake of Cy5.5 labeled PCL-CS NPs in WT, iRhom1 knockout/knockdown or iRhom1 rescue cells with or without ADAM17 inhibitor pretreatment. $n = 3$ independent samples. Data are presented as mean ± s.e.m. in (**a, b, d, e, f, j, k**). Statistical analysis was performed by one-way ANOVA with Tukey's post hoc test for comparison in (**a, j, k**). and two-tailed Student's t-test for comparison in (**b, d, e, f**). Data are representative of two independent experiments in (**c–e, g–i**) and three independent experiments in (**a, b, f, j, k**). Source data are provided as a Source Data file for (**a, b, d, e, f, h, i, j, k**).

decrease ERAP1 stability through facilitating its degradation via the endoplasmic reticulum-associated degradation (ERAD) pathway (Fig. 2j, k). ERAP1 has been well studied and shown to play an important role in Ag processing, such as in trimming the OVA SIINFEKL precursors to become the final SIINFEKL peptide. The significance of iRhom1/ERAP1 interaction was demonstrated by the data that iRhom1 KO or KD led to significant improvement in the presentation of SIIN-FEKL antigen, resulting in significantly enhanced lysis of OVA-B16 cells in a coculture assay with OT-1 T cells. Although iRhom1 likely affects antitumor immunity directly or indirectly through multiple mechanisms, our data strongly support the notion that iRhom1 inhibits the cytotoxic T-cell response at least partly by reducing the stability of ERAP1 protein and the ERAP1-mediated antigen processing and presentation. Our preliminary data show that KD of iRhom2 had no significant impact on the protein expression level of ERAP1. Whether and how iRhom2 regulates antitumor immunity requires more study in the future.

We noticed the presence of high MW aggregates in WB following overexpression of iRhom1 (Supplementary Fig. 4). It is unclear if such aggregates do exist inside cells, which may cause non-physiological response and complicate our data interpretation. Nonetheless, our conclusions on the roles of iRhom1 in chemo-immune-resistance are supported by data from experiments using different approaches.

Numerous reports have been published on CD44-mediated tumor targeting using hyaluronic acid (HA) or CS-decorated NPs. One major limitation with this targeting strategy is the expression of CD44 on liver sinusoidal endothelial cells (LSECs) that remove most of the NPs in blood due to their abundance. However, the level of CD44 on LSECs is significantly lower than that on tumor cells or tumor ECs. Therefore, decoration of the NPs with "optimal" amounts of PEG shall lead to drastic reduction in the interaction of the NPs with LSECs without significantly compromising their productive interaction with tumor ECs and/or tumor cells. We have demonstrated the success of this strategy initially with PMBOP-CP NPs[40] and now with the new biodegradable PCL-CP NPs (Fig. 4a): the injected NPs were largely concentrated at tumor site along with decreased uptake by liver. We have further systematically studied the mechanism of tumor targeting and penetration using 2D, 3D, and mouse tumor models generated with WT and CD44$^{-/-}$ tumor cells, and WT and CD44$^{-/-}$ mice. Our data suggest that accumulation of the NPs at tumor site is largely achieved through targeting of CD44 on tumor ECs. Following intracellular delivery into tumor ECs, parts of the NPs are released into the cytosol from endosome/lysosomes while significant amounts of NPs reach the tumor cells through transcytosis. At the same time, small amounts of NPs reach the tumor cells through EPR. Once extravasation, the above process will continue through layers of tumor cells, leading to both intracellular delivery of cargos and deep penetration of the NPs in the tumor tissues. The release of cargos (drug/siRNA) or the entire PCL-CP NPs from endosome may result from protonation of the primary amines of the glucosamine residues as the endosome matures and becomes more acidic, leading to a high charge density and membrane destabilization[53,54]. The presence of lipid chain in PCL polymer shall further facilitate the interaction of the NPs with the endosome membrane. In addition, enzymatic degradation of chitosan further induces the escape from endosomes/lysosomes, which is driven by intravesicular osmotic swelling[55]. Interestingly, KO/KD of either iRhom1 or iRhom2 decreases CD44 cleavage and further improves the CD44-mediated tumor targeting of NPs likely through decreasing the activity of ADAM17. This is consistent with literature that ADAM17 mediates the cleavage of CD44 extracellular domain (ectodomain) in various types of cells including tumor cells[44,56]. It should be noted that ADAM17 inhibitors are currently being evaluated as new therapeutics for treatment of various types of cancers[57–59]. Therefore, ADAM17 inhibitors and CD44-targeting NPs may be combined to achieve synergistic antitumor activity.

Targeted delivery of iRhom1 pre-siRNA alone led to a modest antitumor activity along with increased numbers of CD8$^+$ T cells. Codelivery of iRhom1 pre-siRNA with DOX or CPT-SAHA led to further improvement in both the overall therapeutic efficacy and tumor immune microenvironment. The superior antitumor activity of the combination therapy is likely attributed to several mechanisms including the direct growth inhibiting activity of iRhom1 KD, the increased sensitivity of tumor cells to chemotherapy agents, and activated antitumor immune response. Importantly, this approach can be combined with anti-PD1 antibody to further improve the therapeutic outcome. Targeting iRhom1 in combination with chemotherapy may represent an effective immunochemotherapy for the treatment of various types of cancers including breast and colon cancers.

## Methods
### Clinical data analysis
GEPIA2 web tool was used for the analysis of iRhom1 expression in different cancer types and the correlation between iRhom1 expression and survival outcome[60].

For the analysis of specific chemotherapy cohort, we collected clinical information and the transcriptome data of 1174 patients of BC from TCGA-BRCA, and 515 CRC patients from the TCGA-COAD. Triple negative breast cancer patient cohort (BRCA-TNBC) was selected based on the IHC information of BC patients. The cohort of patients that received chemotherapy was identified based on the clinical information to generate BRCA-TNBC-chemotherapy cohort ($n = 85$) and COAD-chemotherapy cohort ($n = 145$) respectively. The analyses in this study were mostly restricted to primary tumor samples. The transcriptome data from GDC were annotated based on human reference genome GRCh38 and were quantified as the gene-level expression in FPKM. The cut-off of expression is 50 percent quantile.

For the analysis of the correlation of the expression levels of iRhom1 and APP-related molecules in human tumor tissues, a CPTAC dataset that contains mass spectrometry-based proteomics of 122 treatment-naive primary breast cancers was used and analyzed[61].

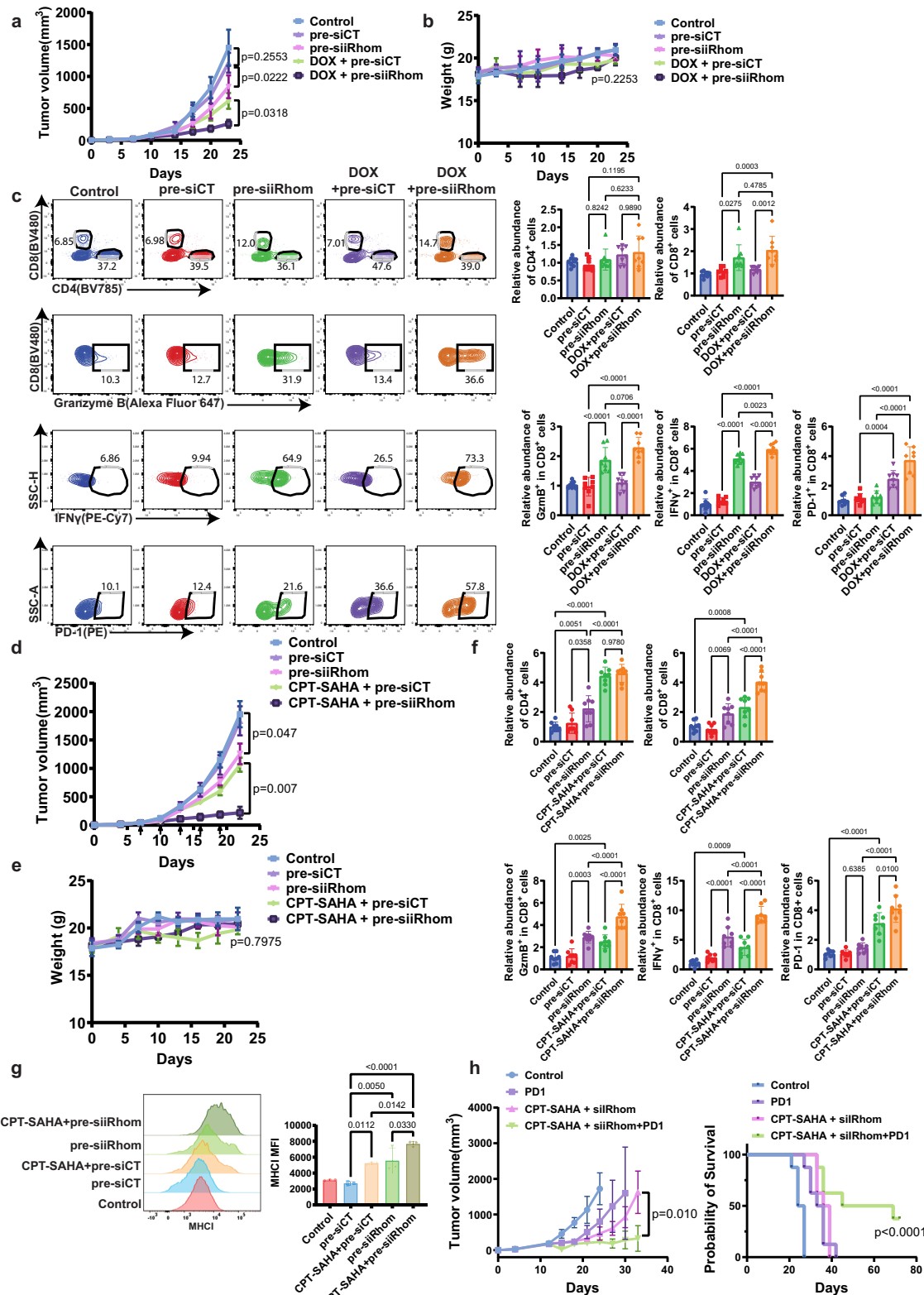

## Clustering analysis of drug response

Drug response data of 545 agents tested across 722 cancer cell lines are downloaded from the CTRP[62,63]. Two hundred and eighty-two (282) agents that showed sensitivity with low RHBDF1 expression were clustered based on their mechanism of action. Category of mechanism of action was defined by the protein targets or activity of agents based on the Informer Set of CTRP[18]. One hundred and sixty-two (162) drug targets were clustered together with average correlation Z-score and

total number of molecules in each cluster (see. Supplementary Data 1 and 2).

## RNA-seq analysis

RNA seq of iRhom$^{-/-}$ CT26 cell sample and control sample was performed at the Health Sciences Sequencing Core at Children's Hospital of Pittsburgh. Raw sequence data was analyzed by HISAT-Stringtie workflow as described in previously published protocol[64-66] to

**Fig. 7 | Treatment with DOX (CPT-SAHA)/pre-siiRhom PCL-CP NPs led to enhanced therapeutic efficacy and improved TIME.** Changes in tumor volumes (**a**) and body weights (**b**) in mice bearing 4T1 orthotopic tumors receiving various treatments. $n = 5$ animals. **c** Representative flow cytometric analysis and the quantification of the relative abundance of CD8[+] T-cells, CD4[+] T-cells, CD8[+] IFNγ[+] T-cells, CD8[+] Granzyme B[+] T-cells and CD8[+] PD1[+] T-cells in 4T1 tumor tissues after various treatments. $n = 8$ biologically independent samples. Changes in tumor volumes (**d**) and body weights (**e**) in mice bearing CT26 tumors receiving various treatments. $n = 5$ animals. **f** Flow cytometric analysis and the quantification of the relative abundance of CD8[+] T-cells, CD4[+] T-cells, CD8 [+] IFNγ[+] T-cells, CD8[+]

Granzyme B[+] T-cells and CD8[+] PD1[+] T-cells in CT26 tumor tissues after various treatments. $n = 8$ biologically independent samples. **g** Histogram and the quantitative analysis of MHCI expression in CT26 tumor tissues after various treatments. $n = 3$ biologically independent samples. **h** Tumor growth inhibition and survival of CT26 tumor-bearing mice receiving various treatments. $n = 8$ animals. Data are presented as mean ± s.e.m. in all panels. Statistical analysis was performed by one-way ANOVA with Tukey's post hoc test for comparison in (**a**, **b**, **c**, **d**, **e**, **f**, **g**, **h** (left)) and log rank test for comparison in (**h**) (right). Data are representative of two independent experiments in (**a–h**). Source data are provided as a Source Data file for (**a–h**).

generate transcript level gene expression. Then gene set enrichment analysis (GSEA)[67] was performed to identify the treatment-associated alteration in functional pathways.

## Reagent
Dulbecco's Modified Eagle's Medium (DMEM), RPMI 1640 Medium, trypsin-EDTA solution, 3-(4,5dimethylthiazol-2-yl)-2,5-diphenyl tetrazolium bromide (MTT), triethylamine, D-Luciferin, doxorubicin, camptothecin, ivermectin and SAHA were purchased from Sigma-Aldrich (MO, U.S.A). Antibodies for Western blot, IHC, immunostaining and flow cytometry were listed in Supplementary Table 1.

## Cell lines and animals
CT26 murine CRC cell lines (CRL-2638), HCT116 human CRC cell line (CCL-247), 4T1 murine BC cell line (CRL-2539) and MDA-MB-468 human BC cell line (HTB-132) were obtained from ATCC (Manassas, VA). MC38 and MC38-Luc cell lines were kindly given by Dr. Zongsheng Guo (University of Pittsburgh Cancer Institute, University of Pittsburgh, Pittsburgh, PA 15261, USA). MC38 cell lines were originally obtained from Kerafast (MA, USA). CT26-Luc cell line was kindly given by Dr. Jianqin Lu (University of Arizona, College of Pharmacy). B16F10 and B16F10-OVA were obtained from Dr. Da Yang's lab. iRhom1 KO CT26 cell line, iRhom1 KO 4T1.2 cell line, iRhom1 KO B16-OVA cell line, iRhom1 KO B16 cell line, iRhom1 rescue CT26 cell line, iRhom1 overexpression CT26 cell line, B16-SIINFEKL cell line and CD44 KO CT26 cell line were produced by following general lentiviral/retroviral infection protocol as detailed in lentiviral infection section below. They were cultured in DMEM medium (4T1, MDA-MB468, B16, B16-OVA) or RPMI1640 medium (CT26, HCT116) supplemented with 10% FBS and 1% penicillin/streptomycin at 37 °C in a humidified atmosphere with 5% $CO_2$.

Female Balb/c mice (4–6 weeks) were purchased from Charles River (CA, U.S.A). Female C57BL/6 mice (4–6 weeks), B6.129(Cg)-Cd44tm1Hbg/J (CD44[−/−]) mice (4–6 weeks), C57BL/6-Tg(TcraTcrb)1100Mjb/J (OT-1) mice (4–6 weeks), Nu/J mice (4–6 weeks) were purchased from Jackson Lab (CT, U.S.A). All animals were housed under pathogen-free conditions according to AAALAC (Association for Assessment and Accreditation of Laboratory Animal Care) guidelines. All animal-related experiments were performed in full compliance with institutional guidelines and approved by the Animal Use and Care Administrative Advisory Committee at the University of Pittsburgh under Protocol #: 21099779. Mice were housed at an ambient temperature of 22 °C (22–24 °C) and humidity of 45%, with a 14/10 day/night cycle (on at 6:00, off at 20:00), and allowed access to food ad libitum.

## Vector, RNA interference, and lentiviral infection
RHBDF1 cDNA ORF Clone (Cat. MG5A2039-U) was obtained from Sino Biological (PA, U.S.A). Flag tagged MAPK14 Mouse ORF Clone (Cat.MR227429) was purchased from Origene (MD, USA). LentiCRISPR V2 Plasmid (#52961) and PresentER-SIINFEKL (GFP) (#102944) were purchased from Addgene (MA, USA). CD44-All-in-one lentiviral sgRNA-CRISPR-Cas9 plasmid was purchased from Horizon Discovery Ltd. (Cambridge, UK). pLVX-HA-Hygro[68], psPAX2, pMD2.G were kindly

provided by Dr. D. Yang (University of Pittsburgh). The LentiCRISPR-sgRHBDF1 was generated by following the published protocol[69]. The pLVX-HA-iRhom-Hygro was generated by the insertion of RHBDF1 cDNA ORF into pLVX-HA-Hygro vector. The ERAP1 siRNA (Cat. sc-44435) was purchased from Santa Cruz Biotechnology (TX. USA). IRhom1 human siRNA (Cat. SR312010) and mouse siRNA (Cat. SR427578) were purchased from Origene. IRhom2 siRNA (Human #140833; Mouse #170854) were purchased from Thermo Fisher. pLenti-C-mGFP-P2A-rhbdf1-Puro (Cat. MR210964L4) and pLenti-C-mGFP-P2A-rhbdf2-Puro (Cat. MR210845L4V) plasmids for iRhom1/iRhom2 expression was purchased from Origene. The sequences of customized oligonucleotides were listed in Supplementary Table 2.

IRhom1 KO cell lines were generated by using CRISPR technology. Cells were infected with the lentivirus packaged by LentiCRISPR-sgRHBDF1 expression plasmid encoding puromycin resistance. The sgRNA sequences of RHBDF1 were previously described in GeCKO v2 Mouse CRISPR Knockout Pooled Library [69] for mRHBDF1. The successfully knocked out cells were selected by single clone culture followed by Western blot analysis for the lack of RHBDF1 proteins.

IRhom1 control vector cell line was generated by using CRISPR technology. Cells were treated with a control lentiviral vector with Cas9 coding sequence but without the specific guiding sequences (LentiCRISPR plasmid without sgRNA sequence). The successfully knocked out cells were selected by single clone culture followed by Western blot analysis for the characterization of RHBDF1 proteins expression.

IRhom1 rescue cell line was generated by stable re-expression of pLenti-C-mGFP-P2A-rhbdf1-Puro in iRhom1 KO cell line. The successful re-expression/overexpression cells were selected by cell sorting of EGFP+ population. Cells were further confirmed by Western blot analysis for both GFP and iRhom1 expression.

IRhom1 overexpression cell line was generated by stable expression of pLenti-C-mGFP-P2A-rhbdf1-Puro plasmid in iRhom1 WT cell lines. The successful re-expression/overexpression cells were selected by cell sorting of EGFP+ population. Cells were further confirmed by Western blot analysis for both GFP and iRhom1 expression.

B16 WT or B16 iRhom1 KO cells expressing SIINFEKL peptide were generated by stable expression of PresentER-SIINFEKL (GFP) plasmid. The successful expression cells were selected by cell sorting of GFP+ population. Cells were further confirmed by flow cytometry analysis of H2Kb-SIINFEKL signal.

CD44−/− cells were also generated by using CRISPR technology. Cells were infected with the lentivirus packaged by CD44-All-in-one lentiviral sgRNA-CRISPR-Cas9 plasmid encoding EGFP and puromycin resistance. The successfully knocked out cells were selected by cell sorting of EGFP+ and CD44[−/−] population. Cells were further confirmed by Western blot analysis for the lack of CD44 proteins.

For plasmid transfection and iRhom1-presiRNA-based in vitro RNA interference, cells were transfected with plasmid using Lipofectamine™ 3000 (ThermoFisher, #L3000015) following the manufacturer's instructions. For siRNA-based in vitro RNA interference, cells were transfected with 40 nM specific siRNA, or a control siRNA using Lipofectamine RNAiMAX (ThermoFisher, #13778150) per the manufacturer's instructions.

## Preparation of rhomboid pre-siRNA

Production of rhomboid pre-siRNA (pre-siiRhom) and control scaffold tRNA (LSA) was conducted as described before[70–72]. Briefly, HST08 bacterial cells were transformed with Bio-Rhomboid-siRNA plasmid or the tRNA-plasmid. Following the isolation of total RNA from bacteria, recombinant RNA was purified with an NGC Quest 10 Plus Chromatography system consisting of an ENrichTM Q 10×100 column (Bio-Rad, Hercules, CA). RNA concentration was determined with a Nano-drop 2000 Spectrophotometer (Thermo Fisher Scientific), and RNA purity was examined by high-performance liquid chromatography as described before[73]. Endotoxin levels were determined by using Limulus Amebocyte Lysate Pyrogent-5000 kinetic assay (Lonza, Walkersville, MD), following the manufacturer's instructions.

## Tumor models

Subcutaneous (s.c.) CRC tumor model was established by injecting CT26 ($5 \times 10^5$ per mouse) cells into the right abdomen of Balb/c mice. Orthotopic breast tumor model will be established by inoculating 4T1 cells ($1 \times 10^5$ cells/mouse) into the mammary gland of a BALB/c or nude mouse. Subcutaneous (s.c.) CD44$^{-/-}$ tumor model was established by injecting CD44$^{-/-}$ CT26 ($5 \times 10^5$ per mouse) cells into the right abdomen of Balb/c mice or CD44$^{-/-}$ MC38 ($5 \times 10^5$ per mouse) cells into the right abdomen of B6.129(Cg)-Cd44tm1Hbg/J (CD44$^{-/-}$) mice. S.C. luciferase-expressing CRC model was established by injecting CT26-Luc ($5 \times 10^5$ per mouse) cells into the right flank of Balb/c mice. The iRhom KO tumor model was established by inoculation more iRhom1$^{-/-}$ CT26 cells ($5 \times 10^6$ per mouse) together with Matrigel into the right abdomen of Balb/c mice.

## Western blot

The total protein was extracted from indicated cells with radio-immunoprecipitation assay (RIPA) lysis buffer supplemented with 1× protease inhibitor (Sigma-Aldrich) and phosphatase inhibitors (Thermo Fisher Scientific) for 15 min. The protein concentration was determined by a BCA Protein Assay Kit (Thermo Fisher Scientific). Afterward, the cell lysates were supplemented with 5× SDS loading buffer, denatured at 98 °C for 10 min, resolved on SDS–polyacrylamide gel electrophoresis, and transferred onto polyvinylidene difluoride membranes (Bio-Rad). The membrane was then incubated with appropriate primary antibodies overnight at 4 °C, followed by incubation with horseradish peroxidase–conjugated secondary antibody for another 1 hour at room temperature. Signal was visualized by enhanced chemiluminescence substrate (Thermo Fisher Scientific) and exposed by films with the AX700LE film processor (Alphatek) or iBright™ FL1500 Imaging System.

## Co-immunoprecipitation

Cellular lysates were prepared by incubating the cells in IP lysis buffer (Thermo Fisher Scientific) in the presence of protease inhibitor cocktail (Sigma-Aldrich) and phosphatase inhibitors (Thermo Fisher Scientific) for 20 min at 4 °C with vortexing every 5 min, followed by centrifugation at 15,000 g for 10 min at 4 °C. For immunoprecipitation, about 1 mg of protein was incubated with anti-HA magnetic beads (Thermo Fisher Scientific), Anti-DYKDDDDK magnetic beads (Thermo Fisher Scientific) or Anti-GFP magnetic beads (ChromoTek). Protein was pulled down according to manufacturer's protocol. The precipitated proteins were eluted from the beads by resuspending the beads in 2× SDS-PAGE loading buffer and boiling for 10 min. The boiled immune complexes were subjected to SDS-PAGE, followed by immunoblotting with appropriate antibodies.

## In vitro T cell killing assay

CD8$^+$ T cells were isolated from spleens of OT-1 TCR transgenic mice using a CD8a$^+$ T Cell Isolation Kit (Miltenyi Biotec) with a MidiMACS separator. Purified OT-1 T cells were then suspended in RPMI 1640 medium containing 10% FBS, anti-mouse CD3 (clone: 2C11, 4 µg/ml), and anti-mouse CD28 (clone: 37.51, 4 µg/ml). After 48 hours of stimulation, activated OT-1 T cells were seeded into a new plate with B16-OVA cells tumor cells at various E:T ratios in fresh medium for 24 h of further culture. Then cytotoxicity detection kit (Sigma-Aldrich) was used to measure the cytolysis rate elicited by CD8$^+$ T cells against different tumor cells based on producer's protocol. Under the same conditions, B16-OVA (WT or iRhom1-KO) cells were cocultured with CD8$^+$ T cells in sterile 12-well culture plates at 1:20 ratio for 18 h, and images were taken under the phase contrast mode using a BZ-X710 Fluorescence Microscope (Keyence, Itasca, IL, USA).

## Polymer synthesis and chemical characterization

**Synthesis of CPT-SAHA.** Camptothecin (1 g, 2.87 mmol) and 1,4,5-oxadithiepane-2,7-dione (2 g, 12.33 mmol) were dissolved in pyridine (30 mL). The reaction was kept at 40 °C for 48 h. After the reaction, pyridine was removed by rotator evaporator. The residue was mixed with cold ether to form precipitation. The precipitate was washed by HCl (1 mol/L) to obtain CPT-SS-COOH (0.48 g).

CPT-SS-COOH (0.51 g, 1 mmol), SAHA (0.26 g 1 mmol), DCC (0.41 g, 2 mmol), and DMAP (0.024 g, 0.2 mmol) were dissolved in DMF (20 mL), and the reaction was kept at room temperature for 48 h. After the reaction, the precipitates were removed by filtration. The filtrate was mixed with cold ether and the precipitate was collected and purified by silica gel column chromatography. The yield of CPT-SAHA was ~20% (0.146 g). The structure was characterized by NMR (Supplementary Fig. 26). 1H NMR (400 MHz, DMSO-$d6$): $\delta$ 10.35 (s, 1H), 9.85 (s, 1H), 8.68 (s, 1H), 8.16 (d, $J = 8.5$ Hz, 1H), 8.11 (d, $J = 8.12$, 1H), 7.86 (t, $J = 7.4$ Hz, 1H), 7.71 (t, $J = 7.8$ Hz, 1H), 7.58 (d, $J = 8$ Hz, 2H), 7.35 (s, 1H), 7.27 (t, $J = 7.8$ Hz, 3H), 7.01 (t, $J = 7.4$ Hz, 1H), 5.44 (s, 1H), 5.27 (s, 1H), 3.67 (s, 4H), 2.28 (t, $J = 7.4$ Hz, 3H), 1.88–1.96 (m, 5H), 1.47–1.57 (m, 6H), 1.26–1.29 (m, 6H), 0.90 (t, $J = 7.2$ Hz, 3H).

**Synthesis of PEG-chitosan.** Low molecule weight chitosan (340 mg, Sigma Aldrich, Cat. 448869) was dissolved in 5 mL DMF. PEG2K-NHS ester (420 mg, JenKem Technology, M-SCM-2000) and 1 mL triethylamine were added to the DMF mixture. The mixture was shaken at room temperature for 48 h. After the reaction, all the solution was transferred into a dialysis bag (MWCO: 12k–14k) and dialyzed against water. After 1 day dialysis, the solution was lyophilized to generate PEG-Chitosan.

**Synthesis of 2-stearoylcyclohexane-1,3-dione (Ketone lipid).** The synthesis was performed following a previously published route[74]. Dimedone (1.05 mol), stearic acid (1 mol), and DMAP (1.5 mol) were dissolved in dichloromethane ([dimedone] = 1.0 M). A separate solution of DCC (1.2 mol) in dichloromethane ([DCC] = 1.0 M) was added slowly at room temperature to the reaction mixture. The reaction mixture was shaken at room temperature for 4 h. Then, the white N, N'-dicyclohexylurea precipitate was filtered off and washed with dichloromethane until colorless. The dichloromethane filtrate was combined and washed with 3% HCl until the pH of the aqueous phase was <3. The organic phase was separated, dried over Na$_2$SO$_4$, filtered and the solvent was removed under vacuum. The crude, pale yellow solid was recrystallized from EtOAc/Hex to yield colorless crystals. The structure was characterized by NMR (Supplementary Fig. 27). 1H NMR (400 MHz, CDCl3): $\delta$ 3.02 (t, $J = 7.6$ Hz, 2H), 2.54 (s, 2H), 2.35 (s, 2H), 1.60 (m, 2H), 1.25 (m, 30 H), 1.08 (s, 6H), 0.88 (t, $J = 6.8$ Hz, 3H).

**Synthesis of PEG-chitosan-lipid (Schiff link).** PEG-chitosan (370 mg) was dissolved in 5 mL DMF followed by addition of 162 mg of 2-stearoylcyclohexane-1,3-dione (Schiff lipid). The mixture was shaken at room temperature overnight. Cold ether was then added to the reaction mixture. The precipitates were washed with cold ether 3 times. The solid was collected and dried under vacuum. The structure was

characterized by NMR. $^1$H-NMR (400 MHz, DMSO-$d6$): methyl group of lipid tail ($\delta$ 0.85, t, $J$ = 3.24 Hz, 3H); PEG's terminal methoxy group ($\delta$ 3.24, s, 3H); Di-methyl group of lipid ring ($\delta$ 0.98, s, 6H). PEG: lipid = 1:4, calculated based on the integration ratio between methyl group of lipid tail and PEG's terminal methoxy group. The final polymer has ~9.5 PEG units, ~35 lipid units per 100 units of chitosan (Supplementary Fig. 28).

**Synthesis of PEG-chitosan-lipid (amide link).** PEG-chitosan (370 mg) was dissolved in 5 mL DMF, followed by addition of 120 mg stearoyl chloride (Sigma-Aldrich) and 0.5 mL TEA. The mixture was shaken at room temperature overnight. Cold ether was then added to the reaction mixture. The precipitates were washed with cold ether 3 times. The solid was collected and dried under vacuum. The structure was characterized by NMR. $^1$H-NMR (400 MHz, DMSO-$d6$): methyl group of lipid tail ($\delta$ 0.85, t, $J$ = 3.24 Hz, 3H); PEG's terminal methoxy group ($\delta$ 3.24, s, 3H); olefinic hydrogen of oleyl acid motif ($\delta$ 5.32, m, 2H). PEG: lipid = 1:4, calculated based on the integration ratio between methyl group of lipid tail and PEG's terminal methoxy group. The final polymer has ~9.5 PEG units, ~38 lipid units per 100 units of chitosan (Supplementary Fig. 29).

**Preparation and characterization of drug-loaded micelles**
Blank micelles, drug (DOX or CPT-SAHA)-loaded micelles or drug/pre-siiRhom co-loaded micelles were prepared via a dialysis method. Briefly, drug solution (10 mg/mL in DMSO) was mixed with PCL polymer (5 mg/mL in DMSO) at various carrier/drug weight ratios. The mixture was transferred to a dialysis bag (MWCO:3000) and dialyzed against nano water overnight to generate the drug-loaded micelles. The micelles were concentrated by centrifugation using Vivaspin® tube (MWCO:3000) (Sartorious, Germany). Pre-siiRhom diluted with nano water was then mixed with drug-loaded micelles to form PCL/drug/pre-siiRhom complexes. Pre-siiRhom complexation was examined by gel retardation. Subsequent incubation with CS/CS-PEG of various ratios led to the formation of CS/CS-PEG-decorated, drug/pre-siiRhom co-loaded PCL-CP NPs. The amounts of DOX or CPT-SAHA was measured by fluorometer and the amounts of pre-siiRhom was measured by Qubit 4 fluorometer (Thermo Fisher Scientific) according to manufacturer's protocol. Particle sizes and polydispersity were measured by DLS (Nano-ZS 90, Malvern Instruments, Malvern) and the morphology was examined by cryo-electron microscopy (cryo-EM).

**CryoEM methods**
Samples were first checked with negative-stain electron microscopy by applying 3 μL to a freshly glow-discharged continuous carbon on a copper grid and staining with a 1% uranyl acetate solution. Grids were inserted into a Tecnai TF20 electron microscope (Thermofisher Scientific, Waltham, Massachusetts, USA) equipped with a field emission gun and imaged on an XF416 CMOS camera (TVIPS GmbH, Gilching, Germany) to visualize nanoparticle uniformity and concentration. Cryo-grids were prepared by pipetting 3 μL of sample on a Protochips C-flat CF-2/1-3CU-T grid (Protochips, Morrisville, North Carolina, USA) that had been glow discharged at 25 mA for 30 s using an Emitech KX100 glow discharger. Grids were mounted in a Thermofisher Vitrobot Mk 4 with relative humidity of 95%, blotted for 3 s with a force setting of 4, and plunged into a 40/60 mixture of liquid ethane/propane[75] that was cooled by a bath of liquid nitrogen. Grids were transferred onto a Gatan 910 3-grid cryoholder (Gatan, Inc, Pleasanton, California, USA) and into the TF20 microscope maintaining a temperature no higher than −175 °C throughout. The microscope was operated at 200 kV and contrast was enhanced with a 100 μm objective aperture. Cryo-electron micrographs were collected at a nominal 150,000× magnification on the TVIPS XF416 CMOS camera with a pixel size of 0.74 Å at the sample. Low dose methods were used to avoid electron beam damage and images were acquired with TVIPS EMplified software using movie mode for drift correction.

**Measurement of critical micelle concentration (CMC)**
The CMC of PCL-CP/DOX/pre-siRNA was determined via a DLS-based method[76].

**Drug release kinetics**
The in vitro drug release from drug/pre-siiRhom-coloaded PCL-CP NPs was conducted by dialysis method. Briefly, 1 mL of PCL-CP NPs containing 0.2 mg of DOX or CPT-SAHA in PBS buffer or serum were placed in a clamped dialysis bag and immersed in 25 mL of 0.1 M PBS solution containing 0.5% (w/v) Tween 80. The experiment was performed in triplicate in an incubation shaker at 37 °C with gentle shaking (100 rpm). At selected time intervals, 10 μL of PCL-CP/drug/pre-siiRhom solution in the dialysis bag and 5 mL medium outside the dialysis bag were collected while same amount of fresh medium was added for replenishment. The DOX or CPT-SAHA concentration was determined by fluorometer.

**Chitosanase-mediated degradation of PCL micelles**
PCL (Schiff or amide link) micelles in 40 mM sodium acetate (pH = 6) were prepared via dialysis method. Chitosanase (1 mg/mL, Sigma-Aldrich, Cat. 220477) was then added into the micelle solution to a final concentration of 2 μM. The mixture was incubated at 37 °C with gentle shaking. The size and count number were examined by DLS at different times following the incubation.

**In vitro cytotoxicity assay**
Cytotoxicity was performed by MTT assay on 4T1 (murine triple negative breast cancer cell line), MDA-MB-468 (human triple negative breast cancer cell line), CT-26 (murine colorectal cancer cell line), and HCT116 (human colorectal cancer cell line). Cells received various treatments including pre-siiRhom alone, DOX or CPT-SAHA alone, or drug and pre-siiRhom combination at various drug and siRNA concentrations for 48 h. For combination treatment, cells were transfected with pre-siRNA using Lipofectamine 3000 (Invitrogen) for 24 h followed by drug treatment for another 48 h. MTT assay was then performed following a published protocol[77]. $IC_{50}$ was calculated through the Quest Graph™ IC50 Calculator[78]. Synergy was calculated through the SynergyFinder[79].

**Pharmacokinetics and in vivo biodistribution**
The plasma PK profiles of both DOX and pre-siiRhom after treatment with PCL-CP/DOX/pre-siiRhom were determined by HPLC-FLR and RT-PCR respectively. Groups of three female Balb/c mice bearing 4T1 orthotopic breast tumors (~400 mm$^3$) received tail vein injections of free DOX.HCl, free pre- siiRhom or PCL-CP/DOX/pre-siiRhom at a DOX dose of 5 mg/kg and a pre-siiRhom dose of 1.5 mg/kg, respectively. Blood samples were collected into tubes containing ETDA at designated time points (5 min, 30 min, 1 h, 4 h, and 24 h post injection). The samples were centrifuged at 15,000 $g$ for 10 min at 4 °C and 200 μL of plasma was collected. For DOX measurement, 100 μL plasma was mixed with 50 μL daunorubicin (1 μg/mL, internal control) methanol solution, 250 μL 12 mM phosphoric acid and additional 600 μL methanol, and the mixture was vortexed for 5 min. The samples were centrifuged at 15,000 $g$ for 10 min at 4 °C and 500 μL of the clear supernatant was collected and injected into HPLC for DOX measurement using a fluorescence detector. The elution condition of DOX is 0.1% TFA: methanol: acetonitrile (50:25:25) with the excitation at 480 nm and emission at 580 nm.

For pre-siiRhom measurement, RNA was purified from 30 μL plasma using miRNeasy Serum/Plasma kit (Qiagen, MD, U.S.A). The pre-siiRhom was measured by qRT-PCR as previously published[80]. Briefly, a standard curve was generated by spiking known amounts of

pre-siiRhom followed by purification and qRT-PCR amplification as described above. Primer sequences were listed in Supplementary Table 2. The copy number of pre-siiRhom in each sample was determined by applying the Ct value to the standard curve. Noncompartmental pharmacokinetic analysis was executed by PK R package[81].

For biodistribution study, tumor-bearing mice received free DOX.HCl, pre-siiRhom alone or PCL-CP/DOX/pre-siiRhom as described above. Tumors and major organs (heart, liver, spleen, lung, and kidney) were collected at designated time points (5 min, 30 min, 1 h, 4 h and 24 h post injection). Tumors and organs were weighted and PBS was added at 5 mL per gram tissue. Tissues were homogenized in PBS buffer. For DOX measurement, 100 μL of homogenized tissue were mixed with 50 μL daunorubicin methanol solution (1 μg/mL), 250 μL 12 mM phosphoric acid and additional 600 μL methanol. The mixture was vortexed for 5 min. The samples were centrifuged at 15,000 $g$ for 10 min at 4 °C and 500 μL of the clear supernatant was collected and injected into HPLC for DOX analysis as described above. For pre-siiRhom measurement, RNA was first extracted from 200 μL of homogenized tissue using Trizol reagent (Invitrogen, NY, U.S.A) following the protocol of the manufacturer. The RNA was further purified with PureLink™ miRNA Isolation Kit (Invitrogen, NY, U.S.A). Then pre-siiRhom was similarly determined as described above.

## Cellular uptake
For cellular uptake study, WT or CD44$^{-/-}$ cells were incubated with Cy5.5 labeled PCL-CP NPs, respectively, for 6 h. Then, the culture medium was discarded, and cells were washed with cold saline 3 times and analyzed by flow cytometry.

## In vitro transwell assay
For all transwell-based transcytosis assays, 0.4 μm pore size filter transwell inserts (Corning, product no. 3470) were used in 6 well plates. An in vitro multilayer cell model was established by seeding the cells onto the apical side of a transwell insert. Growth media volumes were calculated to be 500 μl and 2000 μl for apical and basolateral chambers respectively. Seeded cells were cultured for 3 days to form cell layers[82]. The lower chamber was then added with blank DMEM medium, and the upper chamber was added with DMEM medium containing Cy5.5-labeled PCL, PCL/CS or PCL-CP NPs (500 μg/mL), with or without pre-treatment with transcytosis inhibitor chlorpromazine (6 μg/ml) for 2 h. At 8 h after addition of NPs, the fluorescence intensity of the medium in the lower chamber was determined.

In a separate study, HUVEC cells were seeded onto the 0.4 μm diameter microporous membrane of a cell culture insert, followed by incubation in growth factor-containing ECM medium for 2 days to cover the membrane surface. The lower chamber was added with ECM medium, and the upper chamber was added with ECM medium containing Cy5.5-labeled PCL, PCL/CS or PCL-CP NPs (500 μg/mL), with or without pre-treatment with transcytosis inhibitor chlorpromazine (6 μg/ml) for 2 h. At 8 h after addition of NPs, the fluorescence intensity of the medium in the lower chamber was determined.

## Cell spheroid penetration
WT or CD44$^{-/-}$ CT26 (4T1) cells were seeded in a Nunclon Sphera 96 well U-bottom plate (Thermo Fisher) at a density of 10000 cells per well with 6 μg/ml of collagen I to form a single spheroid per well[83]. After 96 h incubation, dense spheroids were formed, which were confirmed by microscopic examination. The cell spheroids were then incubated with Cy5.5-labeled PCL, PCL/CS or PCL-CP NPs, with or without pre-treatment with chlorpromazine (6 μg/ml) for 2 h. After 18 h incubation with NPs, the spheroids were gently rinsed by saline 3 times. The penetration ability was observed by a confocal laser scanning microscope with Z stack scanning (CLSM, FluoView 3000, Olympus) at 30 μm intervals from the bottom to the middle of the spheroids.

## NP accumulation and penetration in the tumor tissues
The tumor targeting effect of PCL-CP NPs was evaluated in both WT/CD44$^{-/-}$ CT26 tumors (s.c.) established in Balb/c mice and WT/CD44$^{-/-}$ MC38 tumors (s.c.) established in B6.129(Cg)-Cd44tm1Hbg/J (CD44$^{-/-}$) mice. Hydrophobic fluorescence dye DiR was loaded into the Cy5.5-labeled PCL-CP carrier at a wt/wt ratio of 20:1 and intravenously injected into the mice for real-time imaging at the DiR dosage of 0.5 mg/kg. After 24 h, the mice were imaged by an IVIS 200 system with excitation at 730 nm and ICG filter to detect the DiR signal, and excitation at 640 nm and Cy5.5 filter to detect the Cy5.5 signal. The tumor and various organs were then excised for ex vivo imaging following our previous protocol[84]. The tumor was then frozen sectioned, and stained with DAPI to label the cell nucleus and the antibody for CD31 to label the vascular endothelial cell. The fluorescence signals in the tumor sections were examined under Keyence BZ-X800 fluorescence microscope.

## In vivo gene knockdown
Luciferase pre-siRNA- (pre-siLuc) or pre-siCT-loaded PCL-CP NPs were intravenously injected into CT26-Luc tumor-bearing mice at a dose of 1.5 mg pre-siRNA/kg. The efficiency of gene knockdown was measured three times by whole body bioluminescence imaging on the 2nd day following the 1st, 2nd, and 3rd injection of the NPs once every 3 days, respectively. The exposure time was set at 60 s for every experiment. Mice were anesthetized according to protocol prior to imaging.

## Therapeutic treatment
Therapeutic efficacy was evaluated in both murine breast and colon cancer models. Treatments were started when the tumor reached about 50 mm$^3$. For colorectal tumor model, groups of five mice were treated with saline, PCL-CP/pre-siCT, PCL-CP/pre-siiRhom, PCL-CP/CPT-SAHA and PCL-CP/CPT-SAHA/pre-siiRhom respectively. The dose of CPT-SAHA was 20 mg/kg, and the dose of pre-siRNA was 1.5 mg/kg. For breast tumor model, groups of 5 mice were treated with saline, PCL-CP/pre-siCT, PCL-CP/pre-siiRhom, PCL-CP/DOX and PCL-CP/DOX/pre-siiRhom respectively. The dose of DOX was 5 mg/kg and the dose of pre-siRNA was 1.5 mg/kg. Treatments were given once every 3 days for five times by tail vein injection. Tumor volumes were monitored and calculated according to the formula: $(L*W^2)/2$ ($L$ and $W$ are the long and short diameters). Body weights were also followed throughout the entire treatment period. Mice were followed until death or were killed if the tumor size reached 2000 mm$^3$, the maximal tumor size permitted by the Animal Use and Care Administrative Advisory Committee at the University of Pittsburgh. In some cases, this limit has been exceeded the last day of measurement and the mice were immediately euthanized. After completing the in vivo experiment, tumor tissues were collected, fixed in 10% formaldehyde, and then embedded in paraffin. The paraffin-embedded tumor tissues were sectioned into slices at 4 μm using an HM 325 Rotary Microtome for further immunohistochemistry staining.

Survival study with aPD-1 combination was conducted with CT26 tumor model. When the tumor volume reached about 200 mm$^3$, mice were randomly grouped ($n = 8$) and treated with saline, aPD-1, PCL-CP/CPT-SAHA/pre-siiRhom, aPD-1 + PCL-CP/CPT-SAHA/pre-siiRhom. Mice were followed for about 4 months until death or were killed if the tumor size reached 2000 mm$^3$, the maximal tumor size permitted by the Animal Use and Care Administrative Advisory Committee at the University of Pittsburgh. In some cases, this limit has been exceeded the last day of measurement and the mice were immediately euthanized.

## Immunohistochemistry
For immunostaining, the tumor tissue sections were deparaffinized in xylene and hydrated in descending grades of ethyl alcohol. Sections were unmasked with a boiling 0.1 M sodium citrate buffer and

incubated with 0.3% (v/v) hydrogen peroxide to inactivate endogenous peroxidase activity. Then, the sections were washed twice in distilled water and incubated with diluted normal blocking serum for 1 h. After that, the sections were incubated with primary antibody diluted in blocking buffer at 4 °C overnight and washed with TBST for three times prior to incubation with secondary antibody. Then the sections were washed with TBST and treated with Vectastain Elite ABC reagent. The sections were incubated with DAB substrate at room temperature for 15 s. Finally, counterstaining was conducted with hematoxylin for imaging under a BZ-X710 Fluorescence Microscope (Keyence, Itasca, IL, USA).

### Toxicity

Blood samples were collected after therapeutic study. Complete blood count (CBC) was performed via HemaVet 950FS Auto Blood Analyzer. In addition, serum samples were prepared and the serum levels of alanine aminotransferase (ALT), aspartate aminotransferase (AST) and creatinine were evaluated following manufacturer's protocols as indicators of hepatic and renal function.

Tissues were also processed for H&E staining to test if there is histology change following various treatments[85].

### Analysis of tumor-infiltrating immune cells

The immune cell populations in tumors after various treatments were measured by flow cytometry following previous protocol[86] (Supplementary Fig. 30). Briefly, one day after the last treatment, cell suspensions from the spleens or tumors were prepared, and red blood cells lysed. Single cell suspensions were incubated with respective antibodies. Zombie dye was used to discriminate viable and dead cells. Infiltration of various immune cells ($CD4^+$, $CD8^+$, Treg) in tumor tissues, the production of lymphocyte effector molecules (such as IFN-γ and granzyme B) in immune cells, and the surface levels of MHC-I on tumor cells were determined by multi-color flow cytometric analysis.

### Statistics and reproducibility

Statistical analysis was performed with two-tailed Student's $t$ test for comparison between two groups, one-way analysis of variance (ANOVA) with Tukey's post hoc test for comparison between multiple groups, and log-rank (Mantel-Cox) test for survival analysis as indicated in figure legend. Results were considered statistically significant if $P < 0.05$. Prism 10.1.0 (GraphPad Software) was used for data analysis and graph plotting.

Data are representative of two independent experiments in 1e, 1i, 1j, 1k, 2b, 2c, 2d, 2e, 2g-2i, 3f-3h, 4a-4g, 5a-5e, 5c, 5k, 5l, 6c-6e, 6g-6i, 7a-7h. Data are representative of three independent experiments in 1c, 1h, 2j, 2k, 3b-3e,3i, 5g-5j, 6a, 6b, 6f, 6j, 6k. Data are representative of four independent experiments in 1d. Data are representative of five independent experiments in 1f, 1g. Data are representative of two independent experiments in Supplementary Figs. 2a, d, 3a, b, 4a, b, 5a, c, 7, 8a, b, 9, 10a, b, 11, 12, 13a, b, 14a, b, 15a, b, 17a, b, 18a–d, 19a–d, 20, 21a, b, 22a–o, 24, 25, three independent experiments in Supplementary Figs. 2b, c, 5b, 6a, 14c, and four independent experiments in Supplementary Fig. 6b.

### Reporting summary

Further information on research design is available in the Nature Portfolio Reporting Summary linked to this article.

### Data availability

Data from the CTRP and Informer set can be accessed through the Cancer Therapeutics Response Portal (https://portals.broadinstitute.org/ctrp.v2.1/). Data from the TCGA can be accessed through the GDC data portal (https://portal.gdc.cancer.gov/). Data from the CPTAC can be accessed through the proteomics data commons (https://proteomic.datacommons.cancer.gov/pdc/) and published data[61]. The bulk messenger RNA-seq data mapped to the mouse genome (GRCm38: https://www.ncbi.nlm.nih.gov/assembly/GCF_000001635.20/) are available in the NCBI to Gene Expression Omnibus with accession number GSE225818. The remaining data are available within the Article, Supplementary Information or Source Data file. Source data are provided with this paper.

### Code availability

The code for processing TCGA analysis and CPTAC analysis is submitted to Zenodo[87] (https://doi.org/10.5281/zenodo.10372933).

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

## Acknowledgements

This work was supported by the fund from National Institute of Health grants R01CA219399, R01CA223788, R01CA278608, R01CA270623 (to S.L.), R01CA239716 (to B.L. and S.L.), The David and Betty Brenneman Scholar Fund (to S.L.) and 1S10OD030254-01A1 (to R.B.G.). This research was also supported in part by the University of Pittsburgh Center for Research Computing, RRID:SCR_022735, through the resources provided. Specifically, this work used the HTC cluster, which is supported by NIH award number S10OD028483. We thank Dr. Yong Wan for his advice on the use of CPTAC proteomic database in protein–protein correlation analysis.

## Author contributions

Z.L. and So.L. administered the project. Z.L., Y.H., N.B., B.L., and So.L. conceived and designed the experiments. Z.L., Y.H., N.B., Y.C., H.H., Y.W., Z.Z., Sh.L., C.-Y.C., Z.W., and J.C., performed the experiments. Z.L., Y.H., N.B., Y.C., Y.W., Z.W., J.S., D.Y., J.C., and So.L. analyzed the data. A.-M.Y., Q.W., D.Y., B.L., and L.-Y.L. provided resources, materials, advice, and assistance. Z.L., and So.L. wrote the manuscript. All authors edited the manuscript.

## Competing interests

The authors declare no competing interests.
