## [Peer Review File · Nature Communications]

Inhibition of iRhom1 by CD44-targeting Nanocarrier for Improved Cancer ImmunotherapyREVIEWER COMMENTS

Reviewer #1 (Remarks to the Author): with expertise in iRhom biology

This manuscript addresses the role of iRhom1 as a potential target in the context of cancer immunotherapy. iRhoms 1 and 2 are known to be crucial regulators of ADAM17-dependent EGFR signaling as well as other ADAM17-dependent functions. Most studies to date have focused in iRhom2, which controls the function of ADAM17 in immune cells and thereby regulates the release of pro-inflammatory molecules such as TNF α and the IL-6R. Less is known about iRhom1, except that it is crucial for the function of ADAM17 in the absence of iRhom2. Therefore, it would in principle be interesting to learn more about the functions of iRhom1 and the underlying mechanisms.

Here, the authors first analyze a possible role of iRhom1 in chemoresistance in BRCA-TNBC and provide data to suggest that this effect depends on an activation of MAPK14 (p38- α)/Hsp27 signaling by iRhom1. Then they explore a role of iRhom1 as a negative regulator of antigen presentation via an interaction with the ER-associated protease 1 (ERAP1). These findings are used as a rationale to develop nanoparticles carrying pre-siRNA against iRhom1 and DOX, which are shown to target tumor tissue in a heterotopic mouse tumor model in a manner that depends on CD44. This targeting was further corroborated using CD44 $^{-/-}$ or mice and targeting to tumors was shown to be enhanced by inactivation of iRhom1, which resulted in increased CD44 levels. Overall, the treatment with nanoparticles containing pre-siRhom and DOX improved the therapeutic efficacy and the improved toxicity profile. Based on these results, the authors propose that targeting iRhom1 and delivering DOX with nanoparticles would be beneficial for treating various cancer types.

Overall, this manuscript represents a tour de force which proposes several new mechanisms underlying iRhom1 function, then describes newly developed nanoparticles for combinatorial treatment and various models to test these compounds. This is a lot of new information, but the main critique is that the new interactions and functions of iRhom1 reported here were explored in a rather superficial manner that do not necessarily support the authors mechanistic interpretations. iRhom1 is reported as an ER-associated protein, even though this view has changed substantially since the original papers cited by the

authors were published. There is also no discussion of iRhom2, which usually can support the functions of ADAM17 in the absence of iRhom1. The very detailed models of novel interactions and functions of iRhom1 in Figure 1 and 2 K are not fully supported by the rather superficial analysis of the new interacting partners and are therefore major reasons for concern. For example, the authors don't consider that iRhom1 has seven membrane spanning domains and is an integral membrane protein in these models. Thus, although some of the data shown is consistent with a role for iRhom1 in regulating ERAP1, these results are by no means convincing to this reviewer. There is an abundance of data linking iRhom1 and iRhom2 to ADAM17 and EGFR signaling, so to propose interaction and regulation of a different protease, ERAP1, by a protein that is no longer considered an ER chaperone, would require a thorough analysis with many more controls. This manuscript also lacks a discussion that would put the results in the context of the known literature, but instead it only has a brief conclusion. The translational work is certainly intriguing, but in this reviewer's view, there are too many concerns about the mechanistic aspects of the characterization of iRhom1 presented in the first few figures to recommend publication of this manuscript Nature Communications.

As to specifics:

1) In the intro, iRhom1 and 2 are reported to be predominantly localized in the ER, based on a 2011 paper in Cell. There are several more recent papers that clearly show a role for both iRhoms as regulators of ADAM17 on the cell surface. These papers should be mentioned and cited in the intro.

2) References 6 – 9 are not appropriate to support the statement that iRhoms control the activity of ADAM17 and the release of TNF α and several EGF-receptor ligands. Key primary references should be cited here.

3) Why did the authors decide to only analyze iRhom1? Given that both iRhom1 and iRhom2 are crucial for the regulation of ADAM17, they should also analyze iRhom2 and compare the findings.

4) Please provide a better explanation for how the Cancer Therapeutics Response Portal

categorizes mechanisms of action.

5) The data in Figure 1, panel C is confusing and not particularly convincing. Two different concentrations of DOX or CPT-SAHA are shown in blue and pink bars, yet the first two controls (Control and pre-siiRhom) presumably have no DOX or CPT-SAHA added. Nevertheless, the pre-siiRhom appears to lower cell viability on its own, although this is not mentioned or analyzed (this effect is most pronounced in the HCT116 cells). The combined effect is therefore difficult to interpret. Moreover, the concentration range is too narrow, the authors should explore the effect of pre-siiRhom1 over a wider range of DOX and CPT-SAHA concentrations, and include a pre-siiRhom2 as well.

6) The Western blot for iRhom1 in panel D shows μM instead of $\mu\text{g}/\text{ml}$, as in C, so a direct comparison is difficult without converting these numbers. At least panel D shows a reasonable range of concentrations, which should be used to repeat the experiments shown in C. How often were the Western blots repeated? Please include quantification of the levels of iRhom1 in the repeat experiments in the supplements. Panel D and many other panels showing Western blots lack molecular weight markers.

7) In Supplementary figure S2, which is an important figure, several key controls are missing, so it is difficult to interpret these results. In Panel A, the WT 4T1 and CT26 cells with and without Flag-MAPK-14 should be shown and the results from at least 3 experiments quantified separately. The levels of p-MAPK14 in panel B in the CT26Ko are quite variable and do not correlate well with the CPT-SAHA levels, making this difficult to interpret. In panel C, the effect of re-expression of MAPK14 appears to be similar on WT and KO cells, what does this mean? Baseline untreated cells should be included, as should be a control of KO cells re-expressing iRhom1. Finally, similar experiments should be performed with iRhom2 ko cells to show that the effect is specific for iRhom1. What conditions do the statistical comparisons refer to, and the asterisks?

8) The Western blots in Figure 1H are difficult to interpret, with apparent transfer problems in some of the KO HSP27 lanes. These results should be quantified from samples that are performed at least in triplicate.

9) In Figure 1I, the untreated iRhom1 ko has 50% viability compared to the wild type cells, which calls into question the other results shown in this figure. Presumably this figure is not correctly labeled, but as currently shown, it is difficult to interpret.

10) The co-immunoprecipitation results in Figure 1J are also far from conclusive and lack crucial controls. The iRhom1 is not shown, the IP should also be performed with anti-MAPK14 and anti-pMAPK14 and probed for iRhom1, and similar experiments should be performed with iRhom2.

11) The observed increased phosphorylation of pMAPK14 in the presence of iRhom1 should be explored further with gain and loss of function experiments. The authors should keep in mind that iRhom1 is an integral membrane spanning protein with 7 transmembrane domains, so overexpression can easily lead to aggregation artifacts. Therefore it is important to show the entire blot, including the high molecular weight regions, of the iRhom1 and iRhom2 expression for these experiments.

12) In Figure 1 K, the iRhom1 is shown as a soluble cytoplasmic protein, even though it is an integral membrane protein that most likely interacts with ADAM17 throughout the secretory pathway. The model also shows aspects that are not addressed in this manuscript and is thus not properly supported by the results. In this reviewer's opinion, panel K should therefore be removed.

13) In figure 2, the KO cell line should be rescued with iRhom1 to show that this reverses the changes in proliferation observed in the KO. The colors used for the graphs for the different genotypes should be the same.

14) Figure 2D indeed contains rescue experiments, but should also include iRhom2 rescue and perhaps a different multi-membrane-spanning protein, to show that the effect is specific for iRhom1.

15) Figure 2F is not interpretable in its present form and must be improved.

16) The biochemical data on ERAP in panels H is of poor quality and lacking controls, similar to Figure 1J. Here, the IP should also be performed with ERAP, the overexpressed iRhom1 should be shown, and iRhom2 and perhaps another multi-membrane spanning protein should be used as a control.

17) It is not clear that the effects of the KO in panel G and of Eyrastatin in panel I are caused by the proposed direct interaction between iRhom1 and ERAP or by indirect effects due to the inactivation of iRhom1. The effects of Eyrastatin should also be examined in the KO cells.

18) The model in panel K shows iRhom1 floating in the cytoplasm, even though it is a membrane protein. Recent studies have shown a tight complex between iRhom2 and ADAM17 that is recapitulated by AlphaFold, so iRhom1 most likely also interacts with ADAM17 in a similar manner (although this remains to be shown). The data in this manuscript do not support the elaborate model shown in panel K. Is there any additional evidence for a functionally relevant interaction between iRhom1 and ERAP, for example in genetic studies in mice? What about iRhom2?

19) Additional controls, including iRhom2 ko cells and rescue of the iRhom1 cells with iRhom1 need to be included in the experiments shown in Figure 6. This includes the Western blots in panels H and J and their quantification in panel I.

20) The authors should include a discussion section in which their conclusions and interpretations are presented in the context of what is known about the functions of the iRhoms and ADAM17 and with caveats and alternative interpretations.

Reviewer #2 (Remarks to the Author): with expertise in RNA therapeutics, biomaterials

In this manuscript, the authors identified iRhom1 (RHBD1) as a target to overcome chemo-immune resistance in cancer treatment. The results showed that iRhom1 decreased chemotherapy (e.g., DOX and CPT-SAHA) sensitivity by regulating the MAPK14-pHSP27 signal pathway. They also found that iRhom1 inhibited the cytotoxic T-cell response by reducing the stability of ERAP1 and the ERAP1-mediated antigen processing and

presentation. Based on those findings the authors then developed a biodegradable nanocarrier (PCL-CP NPs) for co-delivery of pre-siRNA of IRhom1 and chemotherapy agent (DOX or CPT-SAHA) for improvement of cancer treatment. With high efficiency in tumor targeting and penetration through both the EPR effect and CD44-mediated transcytosis, the PCL-CP NPs led to significantly enhanced antitumor efficacy and activated tumor immune microenvironment in breast (4T1) and colon (CT 26) cancer models. Overall, this study identified an interesting target that is involved in chemo-immune resistance and then developed a pre-siRNA/drug co-delivery NP strategy to silence this target for improving chemo-immune therapy in cancer therapy. This is a comprehensive work and the results are promising. Below are a few issues that need to be further addressed.

1. Compared with the results in Figure 1 F and Figure 1 G, the p-AKT pathway seems to be more correlated with the KO of IRhom1, but the authors chose p-MAPK14-Phsp27 as the signal axis as the mechanism to explain the chemotherapy sensitivity.
2. The authors need to explain why the start point tumor volumes are different in Figure 2B.
3. The figure legend in Figure 2D needs to be double checked (three groups are all IRhom1 KO?).
4. Figure 3G is the result of siRNA gel retardation by the PCL-CPs, which may not clearly prove the protection of NPs by RNase. Also, the siRNA band in the free siRNA group existed, which means the free siRNA also could be somehow resistant to degradation by RNase.
5. In Figure 5L, the CT26 tumor cell spheroids were established by tumor cells with no blood vessels within, which may not strongly support the CD44-mediated ECs transcytosis.
6. The authors may need to further explain why there was no significant difference in Figure 6D.
7. All the abbreviations should be defined when they first appear (such as E/T ratio...).

Reviewer #3 (Remarks to the Author): with expertise in cancer immunology, ERAP1

The authors report that iRhom1 makes cells resistant to chemotherapy by regulating the MAPK14-pHSP27 axis. They found that the downmodulation of iRhom1 reduces the establishment of ERAP1. They develop a biodegradable nanocarrier to transport drugs and pre-siRNA of iRhom1 into tumour cells and endothelial cells via EPR and CD44-mediated transcytosis. Inhibition of iRhom1 together with administration of chemotherapeutic agents caused a control of tumour growth and an increased immune response mediated by CD8+ T cells.

The work is very interesting. However, there are many points to be addressed.

Major

The font used for the figures is illegible without a magnifying. I advise authors to remake figures using readable fonts. Sometimes it is better to put less but readable than a lot but incomprehensible.

Fig 1I - Page 6 line 7. What is described in the text does not reflect what is shown in the figure. The figure shows that ivermectin (which inhibits phosphorylation of HSP27) inhibits the proliferation of WT, but not KO cells. Silencing iRhom1 and ivermectin treatment, in fact, have the same consequence: i.e. decreased phosphorylation of HSP27. Rather than talking about 'abolished sensitisation effect', I would say that this figure proves that the antiproliferative effect of iRhom1 KO is mediated by MAPK14 inhibition.

Fig 1J - the WB of the pulldown is unclear and should be repeated.

Page 6 - Line 15. The authors state that "Our data suggest that iRhom1 inhibits the drug response likely through regulating the iRhom1/MAPK14/pHSP27 pathway. As much as it is quite clear that iRhom1 silencing has an effect on the MAPK14/pHSP27 signalling pathway, it is not so clear that iRhom1 inhibits drug response. This assertion is not sufficiently supported by data because, while the authors demonstrated in figure 1C that the lines are sensitised by siRhom1 to Doxorubicin and CPT-SAHA, no experiments in the opposite direction were performed. In detail:

- does iRhom1 overexpression induce resistance in a cell line sensitive to these drugs?
- does iRhom1 overexpression induce resistance in a cell line KO for MAPK14?

- If I silence iRhom1 in a cell line with a downstream mutation in the pathway (e.g. MAPK14 constitutively active), do I still induce drug sensitivity?

The mechanism by which iRhom1 would act on MAPK14 activity is not well characterised. It is unclear whether iRhom1 stabilises, phosphorylates or scaffolds, mediating the interaction between MKK and MAPK14 and facilitating their activation.

If the authors silence iRhom1, does MKK still bind MAPK14?

Fig S3 - The authors must include the cells transfected with the control plasmid. In addition, there is only one WT lysate of which the origin (whether 4T1 or CT26) is not indicated. The normaliser is saturated.

Fig 2 - The authors do not evaluate the expression of the murine MHC class I alleles (H2-Kb/H2-Db). It is possible that the effect of the increased SIINFEKL presentation in B16OVA cells is actually due to a generic increase in antigen presentation. This hypothesis is supported by what is shown in Figure 2C, i.e. as described by them on page 7 "RNAseq data showing upregulation of several immune pathways including enhanced interferon response and enhanced antigen presentation in the KO cells".

Indeed, the experiment shown in Figure S4 does not demonstrate an involvement of ERAP1 in the pro-tumour activity carried out by iRhom1, but rather the (already known) role of ERAP1 in the presentation of a highly specific antigen such as SIINFEKL.

To demonstrate that iRhom1 silencing upregulates ERAP1, the authors would have to assess via WB the expression of the other components of the APP pathway and the presentation of at least one ERAP1-independent antigen. One possibility could be to overexpress the SIINFEKL peptide, which does not need to be cut by ERAP1 to be processed, and to show that in this case iRhom1 silencing does not cause increased presentation.

Infact is not clear how the authors decided to study ERAP1.

Fig 2B - in the text data on immunodeficient mice are discussed before data of immunocompetent mice. The author should invert the graphs

Fig 2E - Why are B16-OVA killed by OT-1 T lymphocytes at a ratio of E:T 20:1 only 25%?

Fig 2G the normalizer is in saturation

Overall, Figure 2I does not prove what they reported in the text:

“Taken together, the above data provides the first evidence that iRhom1 inhibits antigen presentation at least partly through regulating the ERAD-mediated degradation of ERAP1 in the ER”.

In detail, the WB in which the ERAD inhibitor is used demonstrates that upon inhibition of ERAD there is an increase in ERAP1, but this necessarily shows that this is due to its non-degradation. The authors should show that under these circumstances there is no increase in gene transcription and that there is an increase in protein stability instead. Even with this information, the authors could still not argue that iRhom1 mediates ERAP1 degradation through ERAD.

To do so, in fact, they would have to demonstrate that:

- 1) over-expressing iRhom2 increases the degradation of ERAP1
- 2) overexpressing iRhom2 in the presence of an ERAD inhibitor does not lead to degradation of ERAP1
- 3) silencing iRhom1 results in increased protein stability of ERAP1.

Considering that ERAP1 is under the control of IFN signalling, and that the authors have already shown that silencing iRhom1 induces increased transcription of iRhom1- and APP-associated genes, it is much more likely that the increase in ERAP1 is due to this and is concomitant with the increase in the other APP genes.

Given that p38 (MAPK4) is also downstream of IFN γ signalling in several contexts, it is possible that the effects observed on both p38 and ERAP1 are actually dependent on IFN γ signalling and not on direct iRhom1 activity on these proteins. It would be interesting to see how much of the phenotype observed in iRhom1 KO cells is maintained in the presence of an inhibitor of IFN signalling or in a cell line with this mutated signalling pathway.

Fig 2 J: the dot distribution does not show such a strong correlation.

Fig 7. Is statistical power analysis sufficient with 3 mice? if it is not sufficient it is necessary to repeat the experiment with an adequate number of mice, $n > 7$.

Minors

I would not mention figure S2 if they do not describe figure 1 first, it causes confusion.

Fig 1A - Page 4 line 21. The acronyms do not coincide. In the fig. the acronyms BRCA-TNBC or COAD are used, while in the text BC or CRC are used. It would be more understandable if this were standardised.

Fig 1C - The authors should better explain in the text why they chose DOX and CPT-SAHA for their treatments.

Fig 2B - Keep the same colours for the straight lines in the plots, see how in the lower panel the colours are reversed from the upper one between CTR and KO. It would also be appropriate for the colour chosen for WT to be more different and for the WT curve to also be included in the panel regarding growth in immunodeficient mice.

Reviewer #4 (Remarks to the Author): with expertise in nanomedicine, cancer

In this manuscript, the authors identified, IRhom1(RHBDF1) as a target involved in chemo-immune-resistance that can induce cancer cell proliferation along with the inhibition of immune cell infiltration. Authors described a chitosan based biodegradable nanocarrier for effective in codelivery of iRhom pre-siRNA (pre-siiRhom) and chemotherapeutic drugs with chondroitin sulfate as targeting moiety towards CD44. Treatment with the nano-construct led to an efficient antitumour immune response together with the chemo sensitization ability and explained the plausible immune activation mechanism. The authors have approached the problem in a systematic way which is supported by a sufficient number of experimental models. Even though, the work has well demonstrated with enough supportive data, the result and discussion part needs improvement in terms of description of the experiments conducted and the flow of writing.

After reviewing the manuscript, I felt that the manuscript can be accepted for publication in Nature communications after revision as suggested below:

Some specific comments are as follows:

1. Authors should have given the effect of iRhom KO in the proliferation rate of in vitro cultured cells.
2. It will be more convincing to check the expression level of iRhom1 in WT B16OVA before

explaining about the Ag presentation experiment (Figure 2)

3. Authors should describe the exact interaction mechanism taking place during the micelle formation.

4. In Page no.6, line no 7-9 has to be re-written as it seems to be confusing

5. It is advisable to change the colour code of line graph of either KO or WT in Figure 2B for easy identification.

6. Authors can provide better quality bright field images in Figure 2F. The morphology of the cells is not clearly distinguishable in the provided image.

7. The scale bar provided for the CryoEM is 20 nm, which approximates for the particle size of 50-60 nm, doesn't match with the size of micelle explained in the manuscript.

8. In Figure 3G, the construct siRNA bands are not clear the in the gel and also the authors can provide the explanation for figure 3G in the text.

9. What is the effect of PCL concentration on the micelle size.

10. Authors should provide the critical micelle formation concentration.

11. DLS data of the final construct should be provided.

12. It is advisable to provide the number of PEG per 100 units of chitosan in the construct

13. Authors should explain the drug release mechanism inside the cells upon endocytosis in detail.

14. Since the CD44 is also expressed on the surface of the endothelial cells, how the authors can explain the off-target effect and cargo release restricted to tumor cells?

15. As per Figure S9, no significant change is visible in the WB data of ERAP in comparison with the in vitro data.

16. In Table S1, the entries in the drug target column are incomplete.

17. In Fig 4E change the Y axis unit "ug to μg ".

18. Authors should provide the scale bar for Figure 8B.

19. The authors should provide the full form of the abbreviations used in the text.

20. The writing style can be improved so that, there can be a flow for the experiments performed.

21. The authors should check for the grammatical errors.

We would like to thank all reviewers' precious time for reviewing this manuscript and the insightful comments. The following detail changes that have been made in response to each specific comment of the reviewers:

Response to Reviewer #1:

This manuscript addresses the role of iRhom1 as a potential target in the context of cancer immunotherapy. iRhoms 1 and 2 are known to be crucial regulators of ADAM17-dependent EGFR signaling as well as other ADAM17-dependent functions. Most studies to date have focused in iRhom2, which controls the function of ADAM17 in immune cells and thereby regulates the release of pro-inflammatory molecules such as TNF α and the IL-6R. Less is known about iRhom1, except that it is crucial for the function of ADAM17 in the absence of iRhom2. Therefore, it would in principle be interesting to learn more about the functions of iRhom1 and the underlying mechanisms.

Here, the authors first analyze a possible role of iRhom1 in chemoresistance in BRCA-TNBC and provide data to suggest that this effect depends on an activation of MAPK14 (p38-a)/Hsp27 signaling by iRhom1. Then they explore a role of iRhom1 as a negative regulator of antigen presentation via an interaction with the ER-associated protease 1 (ERAP1). These findings are used as a rationale to develop nanoparticles carrying pre-siRNA against iRhom1 and DOX, which are shown to target tumor tissue in a heterotopic mouse tumor model in a manner that depends on CD44. This targeting was further corroborated using CD44^{-/-} or mice and targeting to tumors was shown to be enhanced by inactivation of iRhom1, which resulted in increased CD44 levels. Overall, the treatment with nanoparticles containing pre-siiRhom and DOX improved the therapeutic efficacy and the improved toxicity profile. Based on these results, the authors propose that targeting iRhom1 and delivering DOX with nanoparticles would be beneficial for treating various cancer types.

Overall, this manuscript represents a tour de force which proposes several new mechanisms underlying iRhom1 function, then describes newly developed nanoparticles for combinatorial treatment and various models to test these compounds. This is a lot of new information, but the main critique is that the new interactions and functions of iRhom1 reported here were explored in a rather superficial manner that do not necessarily support the authors mechanistic interpretations. iRhom1 is reported as an ER-associated protein, even though this view has changed substantially since the original papers cited by the authors were published. There is also no discussion of iRhom2, which usually can support the functions of ADAM17 in the absence of iRhom1. The very detailed models of novel interactions and functions of iRhom1 in Figure 1 and 2 K are not fully supported by the rather superficial analysis of the new interacting partners and are therefore major reasons for concern. For example, the authors don't consider that iRhom1 has seven membrane spanning domains and is an integral membrane protein in these models. Thus, although some of the data shown is consistent with a role for iRhom1 in regulating ERAP1, these results are by no means convincing to this reviewer. There is an abundance of data linking iRhom1 and iRhom2 to ADAM17 and EGFR signaling, so to propose interaction and regulation of a different

protease, ERAP1, by a protein that is no longer considered an ER chaperone, would require a thorough analysis with many more controls. This manuscript also lacks a discussion that would put the results in the context of the known literature, but instead it only has a brief conclusion. The translational work is certainly intriguing, but in this reviewer's view, there are too many concerns about the mechanistic aspects of the characterization of iRhom1 presented in the first few figures to recommend publication of this manuscript Nature Communications.

As to specifics:

- Q1:** In the intro, iRhom1 and 2 are reported to be predominantly localized in the ER, based on a 2011 paper in Cell. There are several more recent papers that clearly show a role for both iRhoms as regulators of ADAM17 on the cell surface. These papers should be mentioned and cited in the intro.
- A1:** First of all, we would like to thank reviewer 1 for the overall positive comments and nice words about our work. We would like to thank the reviewer for the suggestion and have included the suggested papers in the revised manuscript (**page 3, line 17-18**).
- Q2:** References 6 – 9 are not appropriate to support the statement that iRhoms control the activity of ADAM17 and the release of TNF α and several EGF-receptor ligands. Key primary references should be cited here.
- A2:** We would like to thank the reviewer for the suggestion and have cited the key primary references on the roles of iRhoms in controlling the activity of ADAM17 and the release of TNF α and several EGF-receptor ligands (**Reference 6 ,9, 10** in revised manuscript).
- Q3:** Why did the authors decide to only analyze iRhom1? Given that both iRhom1 and iRhom2 are crucial for the regulation of ADAM17, they should also analyze iRhom2 and compare the findings.
- A3:** This study represents collaborative research of Dr. Song Li' lab and Dr. Luyuan Li's group. Dr. Luyuan Li has a long-standing interest in studying the role of iRhom1 in oncogenesis and tumor immune microenvironment. The other reason that we focus on iRhom1 is that iRhom1 appears to show stronger correlation with survival and drug response in BC and CRC compared to iRhom2 (**Fig. S24**) despite that both are crucial for the regulation of ADAM17. Per the suggestion of the reviewer, we have conducted limited experiments examining the biological consequences of iRhom2 KD. Indeed, our data show that iRhom1 but not iRhom2 plays a role in MAPK14/HSP27-mediated drug resistance and ERAP1-mediated Ag presentation in 4T1 and CT26 models (**Figs. S4, S8, and S12**). However, it cannot be ruled out that iRhom2 may be involved in other cell types or contributes to chemo-immune-resistance through other mechanisms. This has been discussed in the revised manuscript (**page 19, line 12-20**). We feel that

more comprehensive study on the role of iRhom2 is beyond the scope of this work and will be pursued in the future.

- Q4:** Please provide a better explanation for how the Cancer Therapeutics Response Portal categorizes mechanisms of action.
- A4:** The Cancer Therapeutics Response Portal categorizes mechanisms of action based on the protein target or activity of an agent. The detailed information is available in the Informer Set of CTRP. We have also included this information in the main text (**page 5, line 7-8; page 24, line 17-18**)
- Q5:** The data in Figure 1, panel C is confusing and not particularly convincing. Two different concentrations of DOX or CPT-SAHA are shown in blue and pink bars, yet the first two controls (Control and pre-siiRhom) presumably have no DOX or CPT-SAHA added. Nevertheless, the pre-siiRhom appears to lower cell viability on its own, although this is not mentioned or analyzed (this effect is most pronounced in the HCT116 cells). The combined effect is therefore difficult to interpret. Moreover, the concentration range is too narrow, the authors should explore the effect of pre-siiRhom1 over a wider range of DOX and CPT-SAHA concentrations and include a pre-siiRhom2 as well.
- A5:** Thank you for your suggestion. Panel C has now been updated with a wider range of DOX and CPT-SAHA concentrations. We have also included siiRhom1 and siiRhom2 data in **Fig. 1C** and **Fig. S4**, respectively. In addition, we described that iRhom1 KO can contribute to slowdown of cell proliferation (**Fig. S3B**), which explains the fact that cells treated with pre-siiRhom alone show slightly lower cell viability.
- Q6:** The Western blot for iRhom1 in panel D shows μM instead of $\mu\text{g/ml}$, as in C, so a direct comparison is difficult without converting these numbers. At least panel D shows a reasonable range of concentrations, which should be used to repeat the experiments shown in C. How often were the Western blots repeated? Please include quantification of the levels of iRhom1 in the repeat experiments in the supplements. Panel D and many other panels showing Western blots lack molecular weight markers.
- A6:** Thank you for the suggestions. All drug concentrations are now shown as μM . All Westerns are repeated 3 times or more. We have also provided quantitative data for **Fig. 1D** based on densitometry. In addition, we have included molecular weight markers in all Western blots.
- Q7:** In Supplementary figure S2, which is an important figure, several key controls are missing, so it is difficult to interpret these results. In Panel A, the WT 4T1 and CT26 cells with and without Flag-MAPK-14 should be shown and the results from at least 3 experiments quantified separately. The levels of p-MAPK14 in panel B in the CT26Ko are quite variable and do not correlate well with the CPT-SAHA levels, making this difficult to interpret. In panel C, the effect of re-expression of

MAPK14 appears to be similar on WT and KO cells, what does this mean? Baseline untreated cells should be included, as should be a control of KO cells re-expressing iRhom1. Finally, similar experiments should be performed with iRhom2 ko cells to show that the effect is specific for iRhom1. What conditions do the statistical comparisons refer to, and the asterisks?

A7: Thank you for the suggestion. The role of MAPK14/HSP27 axis in drug resistance has been well documented. This study is mainly focused on how iRhom1 regulates this axis and the drug sensitivity. In our previous version, we showed that iRhom1 KO led to decreased phosphorylation of MAPK14 and HSP27 and increased drug sensitivity. Per the reviewer's suggestion, we have performed iRhom1 overexpression experiment. Our new data shows that iRhom1 OE leads to increased phosphorylation of MAPK14 and HSP27, and reduced drug sensitivity. In addition, the decreased drug sensitivity in iRhom1 OE cells is partially reversed by MAPK14 KD (**Fig. 1I**). Therefore, we feel that there is no need to further conduct MAPK14 OE experiment. So, we have decided to delete the previous Fig. S2A.

We apologize for the poor quality of Western for Fig. S2B. We have provided a new western of improved quality.

Regarding Panel C (**Fig. S2A** in the revised manuscript), although MAPK14 OE increases the IC50s in both WT and iRhom1 KO cells, the IC50s for KO cells are significantly lower than those in WT cells, suggesting only partial rescue effect. This might be attributed to regulation of MAPK14 phosphorylation in an iRhom1-independent manner.

Finally, we didn't observe the significant sensitization effect of iRhom2 KD (**Fig. S4B**). In addition, we didn't see significant downregulation of pMAPK14 and pHSP27 after iRhom2 KD (**Fig. S4C**). We feel that there is no need to repeat similar experiments as conducted with iRhom1 KO cells.

Q8: The Western blots in Figure 1H are difficult to interpret, with apparent transfer problems in some of the KO HSP27 lanes. These results should be quantified from samples that are performed at least in triplicate.

A8: We apologize for the poor quality of Western in our previous version. We have provided an improved Western which is included in the revised manuscript as **Fig. 1F**.

Q9: In Figure 1I, the untreated iRhom1 ko has 50% viability compared to the wild type cells, which calls into question the other results shown in this figure. Presumably this figure is not correctly labeled, but as currently shown, it is difficult to interpret.

A9: We apologize for the confusion. As described in the figure legend, **Fig. 1I** is a cotreatment with chemotherapy drug and ivermectin. The dose of chemotherapy drug is fixed at 2.5 μ M for CPT-SAHA or 0.25 μ M for DOX. That is the reason that

iRhom1 KO has 50% viability compared to 79% for the wild type cells since iRhom1 KO cells are more sensitive to chemotherapy drug. We have elaborated this in the figure legend to **Fig. 1J** in the revised manuscript.

- Q10:** The co-immunoprecipitation results in Figure 1J are also far from conclusive and lack crucial controls. The iRhom1 is not shown, the IP should also be performed with anti-MAPK14 and anti-pMAPK14 and probed for iRhom1, and similar experiments should be performed with iRhom2.
- A10:** Thank you for the comments. We agree that a reciprocal pulldown is necessary. We have repeated the immunoprecipitation of both forward pulldown and reverse pulldown for iRhom1 and MAPK14 (**Fig. 1K**). However, due to the low abundance of pMAPK14 and the antibody quality, we were unable to pulldown proteins via anti-pMAPK14 and then probe for iRhom1. We have also performed pulldown of MAPK14 and then probe for iRhom2 (**Fig. S8**). However, no significant interaction is detected. Therefore, we didn't perform iRhom2 pulldown.
- Q11:** The observed increased phosphorylation of pMAPK14 in the presence of iRhom1 should be explored further with gain and loss of function experiments. The authors should keep in mind that iRhom1 is an integral membrane spanning protein with 7 transmembrane domains, so overexpression can easily lead to aggregation artifacts. Therefore, it is important to show the entire blot, including the high molecular weight regions, of the iRhom1 and iRhom2 expression for these experiments.
- A11:** We very much appreciate the reviewer's insightful suggestions. We have performed the overexpression of iRhom1 and examined its impact on phosphorylation of MAPK14 and HSP27 (**Fig. 1G**). Indeed, aggregation is shown in high molecular weight regions, along with the non-aggregated form following overexpression of GFP-iRhom1 (**Fig. S5**). Nonetheless, overexpression of GFP-iRhom1 leads to increased phosphorylation of both MAPK14 and HSP27 (**Fig. 1G**). Since iRhom2 KD shows no effect on pMAPK14 and pHSP27 (**Fig. S4C**), we didn't perform iRhom2 overexpression study.
- Q12:** In Figure 1 K, the iRhom1 is shown as a soluble cytoplasmic protein, even though it is an integral membrane protein that most likely interacts with ADAM17 throughout the secretory pathway. The model also shows aspects that are not addressed in this manuscript and is thus not properly supported by the results. In this reviewer's opinion, panel K should therefore be removed.
- A12:** Thank you for the suggestion, we have removed the illustration in the revised manuscript.
- Q13:** In figure 2, the KO cell line should be rescued with iRhom1 to show that this reverses the changes in proliferation observed in the KO. The colors used for the graphs for the different genotypes should be the same.

- A13:** Thank you for the suggestion, we have provided a new **Fig. 2B** with the data of rescue experiment added. In addition, we now use the same color for each phenotype in different figures.
- Q14:** Figure 2D indeed contains rescue experiments, but should also include iRhom2 rescue and perhaps a different multi-membrane-spanning protein, to show that the effect is specific for iRhom1.
- A14:** Thank you for the suggestion. We have performed the iRhom2 KD and our data shows that it does not have significant impact on the presentation of MHCII-SIINFEKL (**Fig. S12**). Therefore, we did not further perform iRhom2 rescue experiment.
- Q15:** Figure 2F is not interpretable in its present form and must be improved.
- A15:** Thank you for the suggestion, we have removed the old Fig.2F from the revised manuscript.
- Q16:** The biochemical data on ERAP in panels H is of poor quality and lacking controls, similar to Figure 1J. Here, the IP should also be performed with ERAP, the overexpressed iRhom1 should be shown, and iRhom2 and perhaps another multi-membrane spanning protein should be used as a control.
- A16:** Thank you for the suggestion. We have performed GFP-tagged ERAP1 pulldown followed by probing for iRhom1 and iRhom2, respectively. It is apparent that ERAP1 shows interaction with iRhom1 (**Fig. 2H**) but not iRhom2 (**Fig. S8**). In addition, iRhom2 KD does not have impact on the protein expression level of ERAP1.
- Q17:** It is not clear that the effects of the KO in panel G and of Eyrastatin in panel I are caused by the proposed direct interaction between iRhom1 and ERAP or by indirect effects due to the inactivation of iRhom1. The effects of Eyrastatin should also be examined in the KO cells.
- A17:** Thank you for the suggestion. We have included the Eyrastatin treatment in both WT cells and KO cells as shown in **Fig. 2I**. ERAP1 is upregulated after Eyrastatin treatment in WT cells. However, this upregulation effect is abolished in KO cells. In addition, we have performed a similar study in overexpression cells. Overexpression of iRhom1 leads to further downregulation of ERAP1 level (**Fig. 2G**). Eyrastatin similarly causes upregulation of the ERAP1 expression in iRhom1 OE cells (**Fig. 2I**).
- Q18:** The model in panel K shows iRhom1 floating in the cytoplasm, even though it is a membrane protein. Recent studies have shown a tight complex between iRhom2 and ADAM17 that is recapitulated by AlphaFold, so iRhom1 most likely also interacts with ADAM17 in a similar manner (although this remains to be shown).

The data in this manuscript do not support the elaborate model shown in panel K. Is there any additional evidence for a functionally relevant interaction between iRhom1 and ERAP, for example in genetic studies in mice? What about iRhom2?

A18: Thank you for the suggestion, we have removed the proposed model from the revised manuscript. As iRhom1 homozygous KO (*rhbdf1^{-/-}*) mice die within two weeks after birth (Christova et al., 2013), we are not able to conduct cancer-related genetic studies in mice.

Q19: Additional controls, including iRhom2 ko cells and rescue of the iRhom1 cells with iRhom1 need to be included in the experiments shown in Figure 6. This includes the Western blots in panels H and J and their quantification in panel I.

A19: Thank you for the suggestion, we have included the data of iRhom2 KD (**Fig. S16**) and iRhom1 rescue (**Fig. 6I&K**) in the revised manuscript.

Q20: The authors should include a discussion section in which their conclusions and interpretations are presented in the context of what is known about the functions of the iRhoms and ADAM17 and with caveats and alternative interpretations.

A20: We would like to thank the reviewer for the suggestion and have included a Discussion section in the revised manuscript.

Response to Reviewer #2:

In this manuscript, the authors identified iRhom1 (RHBDP1) as a target to overcome chemo-immune resistance in cancer treatment. The results showed that iRhom1 decreased chemotherapy (e.g., DOX and CPT-SAHA) sensitivity by regulating the MAPK14-pHSP27 signal pathway. They also found that iRhom1 inhibited the cytotoxic T-cell response by reducing the stability of ERAP1 and the ERAP1-mediated antigen processing and presentation. Based on those findings the authors then developed a biodegradable nanocarrier (PCL-CP NPs) for co-delivery of pre-siRNA of iRhom1 and chemotherapy agent (DOX or CPT-SAHA) for improvement of cancer treatment. With high efficiency in tumor targeting and penetration through both the EPR effect and CD44-mediated transcytosis, the PCL-CP NPs led to significantly enhanced antitumor efficacy and activated tumor immune microenvironment in breast (4T1) and colon (CT 26) cancer models. Overall, this study identified an interesting target that is involved in chemo-immune resistance and then developed a pre-siRNA/drug co-delivery NP strategy to silence this target for improving chemo-immune therapy in cancer therapy. This is a comprehensive work and the results are promising. Below are a few issues that need to be further addressed.

Q1: Compared with the results in Figure 1 F and Figure 1 G, the p-AKT pathway seems to be more correlated with the KO of iRhom1, but the authors chose p-MAPK14-Phsp27 as the signal axis as the mechanism to explain the chemotherapy sensitivity.

- A1:** First of all, we would like to thank reviewer 2 for the overall positive comments and kind words about our work. We choose MAPK14 as the BioPlex Network suggests that MAPK14 is a candidate that likely interacts with iRhom1. In addition, MAPK14/HSP27 axis has been well established to play an important role in drug resistance in several types of cancer. We agree with the reviewers that other signaling pathways such as AKT may also be involved in drug resistance, which requires more studies in the future. We have discussed this in the revised manuscript (**page 20, line 13-16**).
- Q2:** The authors need to explain why the start point tumor volumes are different in Figure 2B.
- A2:** The data in previous figure were from two animal studies starting at different times due to that the control vector cells were not available at the time we inoculated WT and KO cells. We've repeated this experiment with the same tumor cell inoculation time (**Fig. 2B**).
- Q3:** The figure legend in Figure 2D needs to be double checked (three groups are all IRhom1 KO?).
- A3:** The first 3 groups were based on B16-OVA cells and the latter 3 groups were based on B16-OVA KO cells. We have made it more clear in the figure legend (**Fig. 2D**) in the revised manuscript.
- Q4:** Figure 3G is the result of siRNA gel retardation by the PCL-CPs, which may not clearly prove the protection of NPs by RNase. Also, the siRNA band in the free siRNA group existed, which means the free siRNA also could be somehow resistant to degradation by RNase.
- A4:** Thank you for carefully reviewing our manuscript. We apologize that **Fig. 3G** shows the data of gel retardation experiment and no RNase was used. We have made the changes accordingly.
- Q5:** In Figure 5L, the CT26 tumor cell spheroids were established by tumor cells with no blood vessels within, which may not strongly support the CD44-mediated ECs transcytosis.
- A5:** Thank you for your comments. The CD44-mediated ECs transcytosis was demonstrated in a transwell study using HUVEC (**Fig. 5F~J**). The tumor spheroid study was designed to demonstrate the tumor cell transcytosis-dependent tumor penetration **after extravasation**. So, we did not include endothelial cells in the spheroids. We have made it more clear in the revised manuscript.
- Q6:** The authors may need to further explain why there was no significant difference in Figure 6D.

A6: Thank you for your comments. Despite the obvious differences, lack of statistical significance in our previous data is likely due to the small sample size. We have repeated this experiment with a n = 6. With the increase in sample size, we now see clearly statistical significance (**Fig. 6D**).

Q7: All the abbreviations should be defined when they first appear (such as E/T ratio...).

A7: We would like to thank the reviewer for the suggestion and have provided full names for all abbreviations the 1st time they appear.

Response to Reviewer #3:

The authors report that iRhom1 makes cells resistant to chemotherapy by regulating the MAPH14-pHSP27 axis. They found that the downmodulation of iRhom1 reduces the establishment of ERAP1. They develop a biodegradable nanocarrier to transport drugs and pre-siRNA of iRhom1 into tumour cells and endothelial cells via EPR and CD44-mediated transcytosis. Inhibition of iRhom1 together with administration of chemotherapeutic agents caused a control of tumour growth and an increased immune response mediated by CD8+ T cells. The work is very interesting. However, there are many points to be addressed.

Major

Q1: The font used for the figures is illegible without a magnifying. I advise authors to remake figures using readable fonts. Sometimes it is better to put less but readable than a lot but incomprehensible.

A1: We would like to thank the reviewer for the kind words in the general comments and for the kind suggestion. We have tried our best in modifying all figures to improve the readability.

Q2: Fig 1I - Page 6 line 7. What is described in the text does not reflect what is shown in the figure. The figure shows that ivermectin (which inhibits phosphorylation of HSP27) inhibits the proliferation of WT, but not KO cells. Silencing iRhom1 and ivermectin treatment, in fact, have the same consequence: i.e. decreased phosphorylation of HSP27. Rather than talking about 'abolished sensitisation effect', I would say that this figure proves that the antiproliferative effect of iRhom1 KO is mediated by MAPK14 inhibition.

A2: Thank you for the suggestions. We've changed the statement accordingly per your suggestion. (**page 6, line 20-23, page 7, line 1-2**)

Q3: Fig 1J - the WB of the pulldown is unclear and should be repeated. Page 6 - Line 15. The authors state that "Our data suggest that iRhom1 inhibits

the drug response likely through regulating the iRhom1/MAPK14/pHSP27 pathway. As much as it is quite clear that iRhom1 silencing has an effect on the MAPK14/pHSP27 signaling pathway, it is not so clear that iRhom1 inhibits drug response. This assertion is not sufficiently supported by data because, while the authors demonstrated in figure 1C that the lines are sensitized by siRhom1 to Doxorubicin and CPT-SAHA, no experiments in the opposite direction were performed. In detail:

- does iRhom1 overexpression induce resistance in a cell line sensitive to these drugs?

- does iRhom1 overexpression induce resistance in a cell line KO for MAPK14?

- If I silence iRhom1 in a cell line with a downstream mutation in the pathway (e.g. MAPK14 constitutively active), do I still induce drug sensitivity?

The mechanism by which iRhom1 would act on MAPK14 activity is not well characterized. It is unclear whether iRhom1 stabilizes, phosphorylates or scaffolds, mediating the interaction between MKK and MAPK14 and facilitating their activation.

If the authors silence iRhom1, does MKK still bind MAPK14?

A3: Thank you for the comments and suggestions. The WB in **Fig. 1J** has been repeated and a WB of improved quality is provided as **Fig. 1K** in the revised manuscript.

Regarding the 2nd issue, our new data show that rescue of iRhom1 in iRhom KO cell line restores the resistance to chemotherapy drug and iRhom1 overexpression further increases the resistance to the chemotherapy drugs in WT cells. And this effect is attenuated by MAPK14 KD (**Fig. S2C, Fig. 1I**).

Unfortunately, we have not been successful in obtaining constitutively active MAPK14 construct from other investigators. We are also unable to generate this construct within a short period of time. We are sorry that we are not able to conduct the experiment suggested by the reviewer.

Regarding the mode of iRhom1 interaction with MAPK14, iRhom1 KO does not affect the phosphorylation of MKK-3/6 (**Fig. S7**). However, iRhom1 KO does disrupt the interaction of pMKK3/6 with MAPK14, suggesting that iRhom1 may serve as an important scaffold for MAPK14 phosphorylation by MKK (**Fig. S7**). It is also possible that iRhom1 interacts with and stabilizes pMAPK14. We have discussed this in the revised manuscript (**page 20, line 10-13**).

Q4: Fig S3 - The authors must include the cells transfected with the control plasmid. In addition, there is only one WT lysate of which the origin (whether 4T1 or CT26) is not indicated. The normaliser is saturated.

A4: Thank you for the suggestion. We've included a group of control plasmid. In addition, we have specified the cell origin of lysate (**Fig. S9**). We have also provided improved internal controls.

- Q5:** Fig 2 - The authors do not evaluate the expression of the murine MHC class I alleles (H2-Kb/H2-Db). It is possible that the effect of the increased SIINFEKL presentation in B16OVA cells is actually due to a generic increase in antigen presentation. This hypothesis is supported by what is shown in Figure 2C, i.e. as described by them on page 7 “RNAseq data showing upregulation of several immune pathways including enhanced interferon response and enhanced antigen presentation in the KO cells”.
Indeed, the experiment shown in Figure S4 does not demonstrate an involvement of ERAP1 in the pro-tumour activity carried out by iRhom1, but rather the (already known) role of ERAP1 in the presentation of a highly specific antigen such as SIINFEKL.
To demonstrate that iRhom1 silencing upregulates ERAP1, the authors would have to assess via WB the expression of the other components of the APP pathway and the presentation of at least one ERAP1-independent antigen. One possibility could be to overexpress the SIINFEKL peptide, which does not need to be cut by ERAP1 to be processed, and to show that in this case iRhom1 silencing does not cause increased presentation. In fact is not clear how the authors decided to study ERAP1.
- A5:** We would like to thank the reviewer for the insightful comments and suggestions. We focus on ERAP1 because ERAP1 stands out as the only one among several components in APP whose protein expression level is significantly and inversely correlated with the iRhom1 protein level (**Fig. 2G**). Although iRhom1 KO leads to increased mRNA levels of several APP-related genes (**Fig. S11C**), it shows no effect on these molecules at protein levels (**Fig. S11B**). More importantly, iRhom1 KO shows no effect on the presentation of OVA peptide in B16 cells overexpressing the mature SIINFEKL peptide that does not need to be further processed by ERAP1, suggesting a key role of ERAP1 in iRhom1 KO-mediated enhancement in Ag presentation (**Fig. S11A**). We are really grateful to the reviewer for the suggestion of this experiment, which we believe strongly supports a role of iRhom1/ERAP1 in regulating the immune response.
- Q6:** Fig 2B - in the text data on immunodeficient mice are discussed before data of immunocompetent mice. The author should invert the graphs.
- A6:** Thank you for the suggestion, we have made changes accordingly.
- Q7:** Fig 2E - Why are B16-OVA killed by OT-1 T lymphocytes at a ratio of E:T 20:1 only 25%?
- A7:** Thank you for the question. This is likely due to the intrinsic resistance of B16-OVA to OT-1 T cells-mediated cytotoxicity. Similar result was also shown in published literature (DOI: 10.1126/sciadv.aba5412). Importantly, iRhom1 KO led to increased sensitivity to OT-1 T cells-mediated cytotoxicity (**Fig. 2E**).
- Q8:** Fig 2G the normalizer is in saturation.

A8: We have replaced it with a new WB of improved quality (**Fig. 2G**).

Q9: Overall, Figure 2I does not prove what they reported in the text: “Taken together, the above data provides the first evidence that iRhom1 inhibits antigen presentation at least partly through regulating the ERAD-mediated degradation of ERAP1 in the ER”. In detail, the WB in which the ERAD inhibitor is used demonstrates that upon inhibition of ERAD there is an increase in ERAP1, but this necessarily shows that this is due to its non-degradation. The authors should show that under these circumstances there is no increase in gene transcription and that there is an increase in protein stability instead. Even with this information, the authors could still not argue that iRhom1 mediates ERAP1 degradation through ERAD.

To do so, in fact, they would have to demonstrate that:

- 1) over-expressing iRhom2 increases the degradation of ERAP1
- 2) overexpressing iRhom2 in the presence of an ERAD inhibitor does not lead to degradation of ERAP1
- 3) silencing iRhom1 results in increased protein stability of ERAP1.

Considering that ERAP1 is under the control of IFN signalling, and that the authors have already shown that silencing iRhom1 induces increased transcription of iRhom1- and APP-associated genes, it is much more likely that the increase in ERAP1 is due to this and is concomitant with the increase in the other APP genes.

Given that p38 (MAPK4) is also downstream of IFN γ signaling in several contexts, it is possible that the effects observed on both p38 and ERAP1 are actually dependent on IFN γ signalling and not on direct iRhom1 activity on these proteins. It would be interesting to see how much of the phenotype observed in iRhom1 KO cells is maintained in the presence of an inhibitor of IFN signaling or in a cell line with this mutated signaling pathway.

A9: We would like to thank the reviewer again for the insightful comments. Per the reviewer’s suggestions we have performed several new experiments that clearly demonstrate 1) iRhom1 OE leads to decreased ERAP1 (**Fig. 2G**); 2) Treatment of ERAD inhibitor in iRhom1 OE cells rescues the protein level of ERAP1 (**Fig. 2I**); 3) iRhom1 KO leads to increased ERAP1 (**Fig. 2G**); 4) By using cycloheximide chase experiment, we show that iRhom1 KO leads to improved stability of ERAP1 (**Fig. 2J&K**).

Although several APP-related genes are upregulated at mRNA levels following iRhom1 KO, it shows no impact on other components at protein level except on ERAP1, suggesting a mechanism that is independent of IFN signaling.

Therefore, we did not further examine the impact iRhom1 KO or OE in the presence of an inhibitor of IFN signaling.

Q10: Fig 2 J: the dot distribution does not show such a strong correlation.

A10: Our statistical analysis shows a p value of 0.0236. This is in contrast to a poor correlation with the protein levels of other APP-related molecules (**Fig. 2F** in the revised manuscript). The data in **Fig. 2F** was analyzed per the advice of Dr. Yon Wan at Emory who has recently utilized CPTAC-based bioinformatic analysis to study the function of another protein (DOI: 10.1126/sciadv.add6626).

Q11: Fig 7. Is statistical power analysis sufficient with 3 mice? if it is not sufficient it is necessary to repeat the experiment with an adequate number of mice, $n > 7$.

A11: Thank you for your suggestion, we have updated **Fig. 7** with 8 mice in each group.

Minors

Q12: I would not mention figure S2 if they do not describe figure 1 first, it causes confusion.

A12: Thank you for your suggestion and we have made changes accordingly.

Q13: Fig 1A - Page 4 line 21. The acronyms do not coincide. In the fig. the acronyms BRCA-TNBC or COAD are used, while in the text BC or CRC are used. It would be more understandable if this were standardised.

A13: Thank you for the suggestion. TCGA database appears to have its own acronyms, some of which are not frequently seen in the published literature. We have changed BCa to BC in the text to be closer to BRCA-TNBC. We prefer to keep CRC in the text. However, we would be happy to use BRCA-TNBC and COAD throughout the manuscript should the reviewer strongly suggest so.

Q14: Fig 1C - The authors should better explain in the text why they chose DOX and CPT-SAHA for their treatments.

A14: Thank you for the suggestions. DOX, CPT and SAHA are chosen in this study based on the following rationales: 1) Bioinformatic analysis of The Cancer Therapeutics Response Portal (CTRP) suggests that the response to these drugs is likely to be regulated by iRhom1; 2) DOX and CPT are used in the clinic for the treatment of BC and CRC, respectively; 3) Lipid derivatization of CPT with SAHA helps to facilitate the loading of CPT into our NPs. CPT and SAHA have also been reported to have a synergy in antitumor activity. We have elaborated this in the revised manuscript (**page 5, line 11-15**).

Q15: Fig 2B - Keep the same colours for the straight lines in the plots, see how in the lower panel the colours are reversed from the upper one between CTR and KO. It would also be appropriate for the colour chosen for WT to be more different and for the WT curve to also be included in the panel regarding growth in immunodeficient mice.

A15: We would like to thank the reviewer for the suggestion and have made changes accordingly in **Fig 2B** as we have done similarly in **Fig. 7H**.

Response to Reviewer #4:

In this manuscript, the authors identified, IRhom1(RHBDF1) as a target involved in chemo-immune-resistance that can induce cancer cell proliferation along with the inhibition of immune cell infiltration. Authors described a chitosan based biodegradable nanocarrier for effective in codelivery of iRhom pre-siRNA (pre-siiRhom) and chemotherapeutic drugs with chondroitin sulfate as targeting moiety towards CD44. Treatment with the nano-construct led to an efficient antitumour immune response together with the chemo sensitization ability and explained the plausible immune activation mechanism. The authors have approached the problem in a systematic way which is supported by a sufficient number of experimental models. Even though, the work has well demonstrated with enough supportive data, the result and discussion part needs improvement in terms of description of the experiments conducted and the flow of writing.

After reviewing the manuscript, I felt that the manuscript can be accepted for publication in Nature communications after revision as suggested below:

Some specific comments are as follows:

Q1: Authors should have given the effect of iRhom KO in the proliferation rate of *in vitro* cultured cells.

A1: First of all, we would like to thank the reviewer for the overall positive assessment of our work and for the kind and encouraging words. Per the reviewer's suggestion, we have provided the data of *in vitro* proliferation of iRhom KO cells (**Fig. S3B**).

Q2: It will be more convincing to check the expression level of iRhom1 in WT B16OVA before explaining about the Ag presentation experiment (Figure 2)

A2: Thank you for the suggestion. We have provided the data of the characterizations of WT and iRhom1 KO B16 OVA in **Fig. S9**.

Q3: Authors should describe the exact interaction mechanism taking place during the micelle formation.

A3: Thank you for the suggestion. The interaction between DOX/CPT-SAHA and PCL polymer is based on hydrophobic interaction. The interaction between pre-siRNA and PCL is through charge-charge interaction. The coating of CS/PEG-CS on PCL/DOX(CPT-SAHA)/pre-siRNA complex is also through charge-charge interaction to form final construct PCL-CP/DOX(CPT-SAHA)/pre-siRNA. We also

have included the above information in the revised manuscript. (**page 11, line 15-17, line 18-19, line 22-23**)

Q4: In Page no.6, line no 7-9 has to be re-written as it seems to be confusing.

A4: Thank you for the suggestions. We have re-written this sentence about Ivermectin inhibition. (**page 6, line 19-23, page 7, line 1-2**)

Q5: It is advisable to change the colour code of line graph of either KO or WT in Figure 2B for easy identification.

A5: Thank you for the suggestion, we have modified **Fig. 2B** accordingly.

Q6: Authors can provide better quality bright field images in Figure 2F. The morphology of the cells is not clearly distinguishable in the provided image.

A6: We apologize for the low quality of the bright field images; we have deleted these images in the revised manuscript.

Q7: The scale bar provided for the CryoEM is 20 nm, which approximates for the particle size of 50-60 nm, doesn't match with the size of micelle explained in the manuscript.

A7: Thank you for your comments. The discrepancy of size between cryoEM and DLS is due to the principle of measurement. In cryoEM, the size usually is compacted as it cannot clearly show the hydrodynamic layer outside the NP and mainly represents the size of NP core. However, DLS measures the hydrodynamic size and usually is larger than the size in EM due to the existence of the hydrodynamic layer.

Q8: In Figure 3G, the construct siRNA bands are not clear the in the gel and also the authors can provide the explanation for figure 3G in the text.

A8: Thank you for the suggestion. Ethidium bromide emits fluorescence following intercalation into nucleic acids such as pre-siRNA. Lack of fluorescence suggests the formation of tight PCL polymer/pre-siRNA complex, resulting in the exclusion of ethidium bromide from pre-siRNA. We have elaborated this in the revised manuscript (**page 12, line 4-7**).

Q9: What is the effect of PCL concentration on the micelle size.

A9: Thank you for your comments. We have performed a serial dilution of PCL-CP NPs to monitor the size (**Fig. S14B**). The size of the NPs remains stable at concentrations above ~0.001mg/mL. However, further dilution below 0.001mg/mL results in a drastic increase in size.

Q10: Authors should provide the critical micelle formation concentration.

- A10:** Thank you for the suggestion. We have performed a serial dilution of PCL-CP NPs to monitor the count number as a way to calculate critical micellar concentration (CMC). As **Fig. S14A** shows, the CMC is around 0.0027mg/mL.
- Q11:** DLS data of the final construct should be provided.
- A11:** Thank you for the suggestions. We have provided the DLS data of our final construct with a N/P/S (CS)/S (PEG-CS) ratio of 10:1:1:0.5 (**Fig. 2E**). The size is around 130nm and the zeta potential is around -2mv.
- Q12:** It is advisable to provide the number of PEG per 100 units of chitosan in the construct.
- A12:** Thank you for the suggestion. PCL has ~9.5 PEG units per 100 units of chitosan based on its NMR. We have provided this information in the method section of revised manuscript (**page 31, line 6**).
- Q13:** Authors should explain the drug release mechanism inside the cells upon endocytosis in detail.
- A13:** Thank you for the suggestions. We have included a detailed explanation of the release process along with transcytosis in the discussion section (**page 22, line 5-16**).
- Q14:** Since CD44 is also expressed on the surface of the endothelial cells, how the authors can explain the off-target effect and cargo release restricted to tumor cells?
- A14:** Numerous reports have been published on CD4-mediated tumor targeting using hyaluronic acid (HA) or CS-decorated NPs. As indicated by the reviewer, one major limitation with this targeting strategy is the expression of CD44 on liver sinusoidal endothelial cells (LSECs) that removes most of the NPs in the circulation due to their abundance. However, the level of CD44 on LSECs is significantly lower than that on tumor cells or tumor ECs. Therefore, decoration of the NPs with “optimal” amounts of PEG shall lead to drastic reduction in the interaction of the NPs with LSECs without significantly compromising their productive interaction with tumor ECs and/or tumor cells. We have demonstrated the success of this strategy initially with PMBOP-CP NPs (Chen, Y., et al., 2023) and now with the biodegradable PCL-CP NPs (**Fig. 4**): the injected NPs were largely concentrated at tumor site along with decreased uptake by liver. We have elaborated this in the Discussion section of the revised manuscript (**page 21, line 15-22**).
- Q15:** As per Figure S9, no significant change is visible in the WB data of ERAP in comparison with the in vitro data.

- A15:** Thank you for the comments. The non-significant change might be due to the heterogenous composition in tumor tissue. However, our NPs majorly target the tumor cells in tumor tissue. We've repeated the experiment and provided a Western of better quality (**Fig. S19**).
- Q16:** In Table S1, the entries in the drug target column are incomplete.
- A16:** Thank you for the comments. We double checked our Table S1 to make sure there are no incomplete cells under the 'Drug_target' column. **Table. S1** now has 162 entries for 'Drug_target' column in the revised manuscript.
- Q17:** In Fig 4E change the Y axis unit "ug to μg ".
- A17:** Thank you for the suggestion, we have changed the unit ug to μg .
- Q18:** Authors should provide the scale bar for Figure 8B.
- A18:** Thank you for the suggestion, we have added a bar in **Fig. 8B** (Bar = 200 μm) and revised the figure legend accordingly.
- Q19:** The authors should provide the full form of the abbreviations used in the text.
- A19:** Thank you for the suggestion, we have provided full names for all abbreviations the 1st time they appear.
- Q20:** The writing style can be improved so that, there can be a flow for the experiments performed.
- A20:** Per the suggestion of several reviewers, we have added a Discussion section in the revised manuscript. In addition, we have added sentences in the Result section to provide a better flow.
- Q21:** The authors should check for the grammatical errors.
- A21:** We have carefully edited our manuscript. We have also had our manuscript edited by a native English speaker.

REVIEWER COMMENTS

Reviewer #1 (Remarks to the Author):

Please see comments marker "Reply" for the review of this revised manuscript.

Response to Reviewer #1:

This manuscript addresses the role of iRhom1 as a potential target in the context of cancer immunotherapy. iRhoms 1 and 2 are known to be crucial regulators of ADAM17-dependent EGFR signaling as well as other ADAM17-dependent functions. Most studies to date have focused in iRhom2, which controls the function of ADAM17 in immune cells and thereby regulates the release of pro-inflammatory molecules such as TNF α and the IL-6R. Less is known about iRhom1, except that it is crucial for the function of ADAM17 in the absence of iRhom2. Therefore, it would in principle be interesting to learn more about the functions of iRhom1 and the underlying mechanisms.

Here, the authors first analyze a possible role of iRhom1 in chemoresistance in BRCA-TNBC and provide data to suggest that this effect depends on an activation of MAPK14 (p38- α)/Hsp27 signaling by iRhom1. Then they explore a role of iRhom1 as a negative regulator of antigen presentation via an interaction with the ER-associated protease 1 (ERAP1). These findings are used as a rationale to develop nanoparticles carrying pre-siRNA against iRhom1 and DOX, which are shown to target tumor tissue in a heterotopic mouse tumor model in a manner that depends on CD44. This targeting was further corroborated using CD44 $^{-/-}$ or mice and targeting to tumors was shown to be enhanced by inactivation of iRhom1, which resulted in increased CD44 levels. Overall, the treatment with nanoparticles containing pre-siRhom and DOX improved the therapeutic efficacy and the improved toxicity profile. Based on these results, the authors propose that targeting iRhom1 and delivering DOX with nanoparticles would be beneficial for treating various cancer types.

Overall, this manuscript represents a tour de force which proposes several new mechanisms underlying iRhom1 function, then describes newly developed nanoparticles for combinatorial treatment and various models to test these compounds. This is a lot of new

information, but the main critique is that the new interactions and functions of iRhom1 reported here were explored in a rather superficial manner that do not necessarily support the authors mechanistic interpretations. iRhom1 is reported as an ER-associated protein, even though this view has changed substantially since the original papers cited by the authors were published. There is also no discussion of iRhom2, which usually can support the functions of ADAM17 in the absence of iRhom1. The very detailed models of novel interactions and functions of iRhom1 in Figure 1 and 2 K are not fully supported by the rather superficial analysis of the new interacting partners and are therefore major reasons for concern. For example, the authors don't consider that iRhom1 has seven membrane spanning domains and is an integral membrane protein in these models. Thus, although some of the data shown is consistent with a role for iRhom1 in regulating ERAP1, these results are by no means convincing to this reviewer. There is an abundance of data linking iRhom1 and iRhom2 to ADAM17 and EGFR signaling, so to propose interaction and regulation of a different protease, ERAP1, by a protein that is no longer considered an ER chaperone, would require a thorough analysis with many more controls. This manuscript also lacks a discussion that would put the results in the context of the known literature, but instead it only has a brief conclusion. The translational work is certainly intriguing, but in this reviewer's view, there are too many concerns about the mechanistic aspects of the characterization of iRhom1 presented in the first few figures to recommend publication of this manuscript Nature Communications.

Reply: The authors have provided additional data to address the reviewers' concerns.

Unfortunately, the additional results and expanded discussion do not address this reviewer's main concerns about the manuscript that are also stated above:

- 1) the new interactions and functions of iRhom1 reported here were explored in a rather superficial manner that do not necessarily support the authors mechanistic interpretations
 - 2) there are too many concerns about the mechanistic aspects of the characterization of iRhom1 presented in the first few figures to recommend publication of this manuscript
- Nature Communications

In addition, several of the specific comments were not addressed in a satisfactory manner, as outlined below.

As to specifics:

Q1: In the intro, iRhom1 and 2 are reported to be predominantly localized in the ER, based on a 2011 paper in Cell. There are several more recent papers that clearly show a role for both iRhoms as regulators of ADAM17 on the cell surface. These papers should be mentioned and cited in the intro.

A1: First of all, we would like to thank reviewer 1 for the overall positive comments and nice words about our work. We would like to thank the reviewer for the suggestion and have included the suggested papers in the revised manuscript (page 3, line 17-18).

Reply: Reference 11 should also be included with references 6, 9 and 10 on line 21. Reference 11 should be removed from line 22. Reference 7 and 12 are the same.

Q2: References 6 – 9 are not appropriate to support the statement that iRhoms control the activity of ADAM17 and the release of TNFa and several EGF-receptor ligands. Key primary references should be cited here.

A2: We would like to thank the reviewer for the suggestion and have cited the key primary references on the roles of iRhoms in controlling the activity of ADAM17 and the release of TNFa and several EGF-receptor ligands (Reference 6 ,9, 10 in revised manuscript).

Reply: see above, reference 11 should be included here as well, and the primary Science papers by Adrain and McIlwain should be cited here as well.

Q3: Why did the authors decide to only analyze iRhom1? Given that both iRhom1 and iRhom2 are crucial for the regulation of ADAM17, they should also analyze iRhom2 and compare the findings.

A3: This study represents collaborative research of Dr. Song Li' lab and Dr. Luyuan Li's group. Dr. Luyuan Li has a long-standing interest in studying the role of iRhom1 in oncogenesis and

tumor immune microenvironment. The other reason that we focus on iRhom1 is that iRhom1 appears to show stronger correlation with survival and drug response in BC and CRC compared to iRhom2 (Fig. S24) despite that both are crucial for the regulation of ADAM17. Per the suggestion of the reviewer, we have conducted limited experiments examining the biological consequences of iRhom2 KD. Indeed, our data show that iRhom1 but not iRhom2 plays a role in MAPK14/HSP27-mediated drug resistance and ERAP1-mediated Ag presentation in 4T1 and CT26 models (Figs. S4, S8, and S12). However, it cannot be ruled out that iRhom2 may be involved in other cell types or contributes to chemo-immune-resistance through other mechanisms. This has been discussed in the revised manuscript (page 19, line 12-20). We feel that more comprehensive study on the role of iRhom2 is beyond the scope of this work and will be pursued in the future.

Reply: In the revised Figure S4B, the authors show a significant effect of iRhom2 at 5 μ M DOX in 4T1 cells, at 0.25, 0.5 and 5 μ M CPT-SAHA in CT26 cells and at 0.25, .05, 2.5 and 5 μ M CPT-SAHA in HCT-116 cells, so the statement that iRhom2 does not play a role in this model is not supported by these results.

Q4: Please provide a better explanation for how the Cancer Therapeutics Response Portal categorizes mechanisms of action.

A4: The Cancer Therapeutics Response Portal categorizes mechanisms of action based on the protein target or activity of an agent. The detailed information is available in the Informer Set of CTRP. We have also included this information in the main text (page 5, line 7-8; page 24, line 17-18)

Q5: The data in Figure 1, panel C is confusing and not particularly convincing. Two different concentrations of DOX or CPT-SAHA are shown in blue and pink bars, yet the first two controls (Control and pre-siiRhom) presumably have no DOX or CPT-SAHA added. Nevertheless, the pre-siiRhom appears to lower cell viability on its own, although this is not mentioned or analyzed (this effect is most pronounced in the HCT116 cells). The combined effect is therefore difficult to interpret. Moreover, the concentration range is too narrow, the authors should explore the effect of pre-siiRhom1 over a wider range of DOX and CPT-

SAHA concentrations and include a pre-siiRhom2 as well.

A5: Thank you for your suggestion. Panel C has now been updated with a wider range of DOX and CPT-SAHA concentrations. We have also included siiRhom1 and siiRhom2 data in Fig. 1C and Fig. S4, respectively. In addition, we described that iRhom1 KO can contribute to slowdown of cell proliferation (Fig. S3B), which explains the fact that cells treated with pre-siiRhom alone show slightly lower cell viability.

Q6: The Western blot for iRhom1 in panel D shows μM instead of $\mu\text{g/ml}$, as in C, so a direct comparison is difficult without converting these numbers. At least panel D shows a reasonable range of concentrations, which should be used to repeat the experiments shown in C. How often were the Western blots repeated? Please include quantification of the levels of iRhom1 in the repeat experiments in the supplements. Panel D and many other panels showing Western blots lack molecular weight markers.

A6: Thank you for the suggestions. All drug concentrations are now shown as μM . All Westerns are repeated 3 times or more. We have also provided quantitative data for Fig. 1D based on densitometry. In addition, we have included molecular weight markers in all Western blots.

Reply: the increase in iRhom1 protein levels in response to DOX and CPT-SAHA treatment is different in the revised figure 1D in that higher levels of DOX and lower levels of CPT-SAHA are required to elicit this increase, raising questions about the reproducibility of these concentration curves.

Q7: In Supplementary figure S2, which is an important figure, several key controls are missing, so it is difficult to interpret these results. In Panel A, the WT 4T1 and CT26 cells with and without Flag-MAPK-14 should be shown and the results from at least 3 experiments quantified separately. The levels of p-MAPK14 in panel B in the CT26Ko are quite variable and do not correlate well with the CPT-SAHA levels, making this difficult to interpret. In panel C, the effect of re-expression of MAPK14 appears to be similar on WT and KO cells, what does this mean? Baseline untreated cells should be included, as should be a control of KO cells re-expressing iRhom1. Finally, similar experiments should be performed with

iRhom2 ko cells to show that the effect is specific for iRhom1. What conditions do the statistical comparisons refer to, and the asterisks?

A7: Thank you for the suggestion. The role of MAPK14/HSP27 axis in drug resistance has been well documented. This study is mainly focused on how iRhom1 regulates this axis and the drug sensitivity. In our previous version, we showed that iRhom1 KO led to decreased phosphorylation of MAPK14 and HSP27 and increased drug sensitivity. Per the reviewer's suggestion, we have performed iRhom1 overexpression experiment. Our new data shows that iRhom1 OE leads to increased phosphorylation of MAPK14 and HSP27, and reduced drug sensitivity. In addition, the decreased drug sensitivity in iRhom1 OE cells is partially reversed by MAPK14 KD (Fig. 1I). Therefore, we feel that there is no need to further conduct MAPK14 OE experiment. So, we have decided to delete the previous Fig. S2A.

We apologize for the poor quality of Western for Fig. S2B. We have provided a new western of improved quality.

Reply: the revised Figure S2B is now a completely different experiment, with different cells (4T1 WT and KO vs CT26 WT and KO in the original figure) and treatment with 0 – 1 uM DOX instead of the original 0 - 10 uM CPT-SAHA. Moreover, the quality of the blot is not significantly improved, with only a weak increase in p-HSP27 bands that also have a different appearance (no longer a single band) and significant variability in the MAPK14 levels at different treatment concentrations in the WT cells. These experiments do not address the previous critique, but instead substitute a different experiment.

Regarding Panel C (Fig. S2A in the revised manuscript), although MAPK14 OE increases the IC50s in both WT and iRhom1 KO cells, the IC50s for KO cells are significantly lower than those in WT cells, suggesting only partial rescue effect. This might be attributed to regulation of MAPK14 phosphorylation in an iRhom1-independent manner.

Reply: the requested ko cells rescued with iRhom1 were also not included.

Finally, we didn't observe the significant sensitization effect of iRhom2 KD (Fig. S4B). In

addition, we didn't see significant downregulation of pMAPK14 and pHSP27 after iRhom2 KD (Fig. S4C). We feel that there is no need to repeat similar experiments as conducted with iRhom1 KO cells.

Q8: The Western blots in Figure 1H are difficult to interpret, with apparent transfer problems in some of the KO HSP27 lanes. These results should be quantified from samples that are performed at least in triplicate.

A8: We apologize for the poor quality of Western in our previous version. We have provided an improved Western which is included in the revised manuscript as Fig. 1F.

Q9: In Figure 1I, the untreated iRhom1 ko has 50% viability compared to the wild type cells, which calls into question the other results shown in this figure. Presumably this figure is not correctly labeled, but as currently shown, it is difficult to interpret.

A9: We apologize for the confusion. As described in the figure legend, Fig. 1I is a cotreatment with chemotherapy drug and ivermectin. The dose of chemotherapy drug is fixed at 2.5 μ M for CPT-SAHA or 0.25 μ M for DOX. That is the reason that iRhom1 KO has 50% viability compared to 79% for the wild type cells since iRhom1 KO cells are more sensitive to chemotherapy drug. We have elaborated this in the figure legend to Fig. 1J in the revised manuscript.

Reply: In this reviewer's opinion, this experiment remains problematic and difficult to interpret. The authors have now removed the panel in the original Fig. 2I showing 4T1 cells, presumably because it showed >80% viability in WT Ivermectin treated cells vs <50% in iR1KO cells, whereas the revised figure 1C now shows a smaller difference between 4T1 WT siCT and iRhom1 treated cells (see 2 μ M DOX). If the KO of iRhom1 in CT26 cells reduces the viability of the cells by 50% when treated with 2.5 μ M CPT-SAHA compared to the 21% reduction in WT controls, then this very different starting point makes it difficult to interpret the additional effect caused by increasing Ivermectin concentrations. The authors interpret these results to suggest that Ivermectin is specifically inhibiting HSP27 phosphorylation, but there could be other reasons why the KO cells don't respond. Their viability is already highly impaired and the maximal effect of Ivermectin on WT cells brings them to the level of the

KO cells, so this experiment can not necessarily be interpreted in support the authors interpretation.

Q10: The co-immunoprecipitation results in Figure 1J are also far from conclusive and lack crucial controls. The iRhom1 is not shown, the IP should also be performed with anti-MAPK14 and anti-pMAPK14 and probed for iRhom1, and similar experiments should be performed with iRhom2.

A10: Thank you for the comments. We agree that a reciprocal pulldown is necessary. We have repeated the immunoprecipitation of both forward pulldown and reverse pulldown for iRhom1 and MAPK14 (Fig. 1K). However, due to the low abundance of pMAPK14 and the antibody quality, we were unable to pulldown proteins via anti-pMAPK14 and then probe for iRhom1. We have also performed pulldown of MAPK14 and then probe for iRhom2 (Fig. S8). However, no significant interaction is detected. Therefore, we didn't perform iRhom2 pulldown.

Q11: The observed increased phosphorylation of pMAPK14 in the presence of iRhom1 should be explored further with gain and loss of function experiments. The authors should keep in mind that iRhom1 is an integral membrane spanning protein with 7 transmembrane domains, so overexpression can easily lead to aggregation artifacts. Therefore, it is important to show the entire blot, including the high molecular weight regions, of the iRhom1 and iRhom2 expression for these experiments.

A11: We very much appreciate the reviewer's insightful suggestions. We have performed the overexpression of iRhom1 and examined its impact on phosphorylation of MAPK14 and HSP27 (Fig. 1G). Indeed, aggregation is shown in high molecular weight regions, along with the non-aggregated form following overexpression of GFP-iRhom1 (Fig. S5). Nonetheless, overexpression of GFP-iRhom1 leads to increased phosphorylation of both MAPK14 and HSP27 (Fig. 1G). Since iRhom2 KD shows no effect on pMAPK14 and pHSP27 (Fig. S4C), we didn't perform iRhom2 overexpression study.

Reply: The included Western blots in Figure show a higher percentage of aggregated iRhom1

than monomeric iRhom1 (at least when probed with the anti-iRhom1 antibody). Since such aggregates are known to trigger the unfolded protein response, this could indirectly affect the phosphorylation of HSP27 or MAPK shown in the revised Figure 1G (the effect on MAPK14 on the Western blot is quite subtle). This experiment therefore also does not necessarily support the authors' interpretation.

Q12: In Figure 1 K, the iRhom1 is shown as a soluble cytoplasmic protein, even though it is an integral membrane protein that most likely interacts with ADAM17 throughout the secretory pathway. The model also shows aspects that are not addressed in this manuscript and is thus not properly supported by the results. In this reviewer's opinion, panel K should therefore be removed.

A12: Thank you for the suggestion, we have removed the illustration in the revised manuscript.

Q13: In figure 2, the KO cell line should be rescued with iRhom1 to show that this reverses the changes in proliferation observed in the KO. The colors used for the graphs for the different genotypes should be the same.

A13: Thank you for the suggestion, we have provided a new Fig. 2B with the data of rescue experiment added. In addition, we now use the same color for each phenotype in different figures.

Reply: The revised Figure 2B shows a similar rescue effect of the control vector as the iRhom1 rescue vector. This reviewer attempted to understand this experiment (in the previous version, Figure 2B also showed a similar experiment in immunocompetent mice, in which a control vector experiment was similar to WT cells, with an "ns" added between the bars). This seemed to imply that the WT cells had been treated with a control vector, as opposed to an siRhom1 vector, but either way, this experiment does not make sense as presented. Since a rescue of iRhom1 KO cells with iRhom1 or a control was requested and this is presumably shown in the revised Figure 2B, the conclusion is that the control vector has a similar effect as the iRhom1 rescue.

Q14: Figure 2D indeed contains rescue experiments, but should also include iRhom2 rescue and perhaps a different multi-membrane-spanning protein, to show that the effect is specific for iRhom1.

A14: Thank you for the suggestion. We have performed the iRhom2 KD and our data shows that it does not have significant impact on the presentation of MHCII-SIINFEKL (Fig. S12). Therefore, we did not further perform iRhom2 rescue experiment.

Reply: The authors did not perform the requested experiment, which was to include an iRhom2 rescue and perhaps a different multi-membrane-spanning protein, to show the effect is specific for iRhom1. The rescue experiments were to be performed in the iRhom1 ko cells.

Q15: Figure 2F is not interpretable in its present form and must be improved.

A15: Thank you for the suggestion, we have removed the old Fig.2F from the revised manuscript.

Q16: The biochemical data on ERAP in panels H is of poor quality and lacking controls, similar to Figure 1J. Here, the IP should also be performed with ERAP, the overexpressed iRhom1 should be shown, and iRhom2 and perhaps another multi-membrane spanning protein should be used as a control.

A16: Thank you for the suggestion. We have performed GFP-tagged ERAP1 pulldown followed by probing for iRhom1 and iRhom2, respectively. It is apparent that ERAP1 shows interaction with iRhom1 (Fig. 2H) but not iRhom2 (Fig. S8). In addition, iRhom2 KD does not have impact on the protein expression level of ERAP1.

Q17: It is not clear that the effects of the KO in panel G and of Eyrastatin in panel I are caused by the proposed direct interaction between iRhom1 and ERAP or by indirect effects due to the inactivation of iRhom1. The effects of Eyrastatin should also be examined in the KO cells.

A17: Thank you for the suggestion. We have included the Eyrastatin treatment in both WT cells and KO cells as shown in Fig. 2I. ERAP1 is upregulated after Eyrastatin treatment in WT cells. However, this upregulation effect is abolished in KO cells. In addition, we have performed a similar study in overexpression cells. Overexpression of iRhom1 leads to further downregulation of ERAP1 level (Fig. 2G). Eyrastatin similarly causes upregulation of the ERAP1 expression in iRhom1 OE cells (Fig. 2I).

Q18: The model in panel K shows iRhom1 floating in the cytoplasm, even though it is a membrane protein. Recent studies have shown a tight complex between iRhom2 and ADAM17 that is recapitulated by AlphaFold, so iRhom1 most likely also interacts with ADAM17 in a similar manner (although this remains to be shown). The data in this manuscript do not support the elaborate model shown in panel K. Is there any additional evidence for a functionally relevant interaction between iRhom1 and ERAP, for example in genetic studies in mice? What about iRhom2?

A18: Thank you for the suggestion, we have removed the proposed model from the revised manuscript. As iRhom1 homozygous KO (*rhbdf1*^{-/-}) mice die within two weeks after birth (Christova et al., 2013), we are not able to conduct cancer-related genetic studies in mice.

Q19: Additional controls, including iRhom2 ko cells and rescue of the iRhom1 cells with iRhom1 need to be included in the experiments shown in Figure 6. This includes the Western blots in panels H and J and their quantification in panel I.

A19: Thank you for the suggestion, we have included the data of iRhom2 KD (Fig. S16) and iRhom1 rescue (Fig. 6I&K) in the revised manuscript.

Q20: The authors should include a discussion section in which their conclusions and interpretations are presented in the context of what is known about the functions of the iRhoms and ADAM17 and with caveats and alternative interpretations.

A20: We would like to thank the reviewer for the suggestion and have included a Discussion section in the revised manuscript.

Reply: Overall, the discussion is improved. There is some discussion of the role of iRhoms in regulating ADAM17 and some caveats regarding alternative interpretations are included. However, the statement on page 19, lines 13 – 15, that “iRhom2 has more restricted distribution and exclusive functions in bone marrow and immune cells” is not correct, since the expression of iRhom1 and iRhom2 overlap in all tissues except in most immune cells, where little, if any iRhom1 is expressed, and in brain, where very little, if any iRhom2 is expressed in mice (except in microglial cells). In this reviewer’s opinion, the many different conclusions in this manuscript are not sufficiently supported by the results, so that publication in a high quality journal such as Nature Communications can not be recommended by this reviewer.

Reviewer #2 (Remarks to the Author):

This is a revised manuscript that reports the co-delivery of pre-siRNA of iRhom1 and chemotherapy agent (DOX or CPT-SAHA) for improvement of cancer immunochemotherapy. To the authors' credit, they have addressed most of the reviewer's critiques by providing additional information and new data. The following concern may need to be further clarified.

Based on the BioPlex Network, MAPK14 is a candidate that likely interacts with iRhom1, and the MAPK14/HSP27 axis could play a key role in the drug resistance in certain types of cancer. However, as the authors noted in reference 53, the PI3K/AKT may be another pivotal pathway involved in drug resistance which is also supported by the data in the previous submission (Fig. 1F and 1G). It would thus be more meaningful to further clarify the contribution of these biological pathways, rather than just removing the previous data in the revised manuscript.

[**Editorial note:** with respect to your reply to Reviewer #4 original requests, Reviewer #2 is suggesting to provide better quality bright field images for (original) Figure 2F rather than deleting these data]

Reviewer #3 (Remarks to the Author):

the authors replied to my requests

We would like to thank all reviewers for spending time reviewing our manuscript again and for the invaluable comments. We have conducted new experiments and revised our manuscript per the reviewers' comments. The following detail changes that have been made in response to each specific comment of the reviewers:

Response to Reviewer #1:

- Q1:** Reference 11 should also be included with references 6, 9 and 10 on line 21. Reference 11 should be removed from line 22. Reference 7 and 12 are the same.
- A1:** We appreciate the reviewer's suggestion and have revised accordingly (**reference 11** in R2).
- Q2:** See above, reference 11 should be included here as well, and the primary Science papers by Adrain and McIlwain should be cited here as well.
- A2:** We thank the reviewer for the suggestion and have revised accordingly (**reference 9 and 12** in R2).
- Q3:** In the revised Figure S4B, the authors show a significant effect of siRhomb2 at 5 uM DOX in 4T1 cells, at 0.25, 0.5 and 5 uM CPT-SAHA in CT26 cells and at 0.25, .05, 2.5 and 5 uM CPT-SAHA in HCT-116 cells, so the statement that iRhomb2 does not play a role in this model is not supported by these results.
- A3:** We agree with the reviewer about the interpretation of data in Fig. S4B and we apologize for the inaccurate statement. We have revised our data descriptions accordingly (**page 7, line 2-3** in R2).
- Q4:** The increase in iRhomb1 protein levels in response to DOX and CPT-SAHA treatment is different in the revised figure 1D in that higher levels of DOX and lower levels of CPT-SAHA are required to elicit this increase, raising questions about the reproducibility of these concentration curves.
- A4:** The Western is only semi-quantitative and is known to be subject to variations. The upregulation of iRhomb1 following drug treatment in the new data is largely consistent with that from the early data except for small variations in the dose-response. For this reason, we have repeated this Western 4 times and provided the quantitative data based on densitometry analysis to further support our conclusion (**Fig. 1D**).
- Q5:** The revised Figure S2B is now a completely different experiment, with different cells (4T1 WT and KO vs CT26 WT and KO in the original figure) and treatment with 0 – 1 uM DOX instead of the original 0 - 10 uM CPT-SAHA. Moreover, the quality of the blot is not significantly improved, with only a weak increase in p-HSP27 bands that also have a different appearance (no longer a single band) and significant variability in the MAPK14 levels at different treatment concentrations in the WT cells. These experiments do not address the previous critique, but instead substitute a different experiment.

A5: During the R1 revision, in addition to including new data to address the reviewers' comments, we also reorganized some of the data to improve the presentation of our manuscript. We apologized for having not mentioned this in the rebuttal of last submission (R1). Most of the studies in Figures 1 and 2 and related Suppl Figures were performed with both CT26 and 4T1 cells. In the R1 revision, we placed all CT26-related data (or data from both cell lines) in the main figures in order to be consistent with the order of presentations throughout the manuscript. The experiment on **Fig. 1H** and **Fig. S2B** was also conducted with both cell lines but we changed the order of presentation (CT26 vs 4T1 in main figure vs Suppl figure) in the revised (R1) manuscript. We apologize for the confusion.

Regarding the Western, although we are confident about the conclusion of upregulation of p-MAPK14 and p-HSP27, we fully respect the concern of the reviewer over the quality of WB band. We have repeated the Western with a newly purchased transfer well and fresh antibody. We feel that the quality of the new Western is significantly improved (**Fig. 1H & Fig. S2B in R2**). Importantly, the data of new Western are consistent with those of previous Western. We have also provided the quantitative data of densitometry analysis based on 3 repeats (**Fig. S2C**).

Q6: The requested ko cells rescued with iRhom1 were also not included.

A6: We apologize and have now conducted the rescue experiment. The data were included in the revised (R2) manuscript as **Fig. S2A**.

Q7: In this reviewer's opinion, this experiment remains problematic and difficult to interpret. The authors have now removed the panel in the original Fig. 2I showing 4T1 cells, presumably because it showed >80% viability in WT Ivermectin treated cells vs <50% in iR1KO cells, whereas the revised figure 1C now shows a smaller difference between 4T1 WT siCT and siiRhom1 treated cells (see 2 uM DOX). If the KO of iRhom1 in CT26 cells reduces the viability of the cells by 50% when treated with 2.5 uM CPT-SAHA compared to the 21% reduction in WT controls, then this very different starting point makes it difficult to interpret the additional effect caused by increasing Ivermectin concentrations. The authors interpret these results to suggest that Ivermectin is specifically inhibiting HSP27 phosphorylation, but there could be other reasons why the KO cells don't respond. Their viability is already highly impaired and the maximal effect of Ivermectin on WT cells brings them to the level of the KO cells, so this experiment can not necessarily be interpreted in support the authors interpretation.

A7: For the same reason mentioned in **A5**, we placed the data from CT26 cells in Main Figures (Fig. 1J in R1). We apologize for forgetting to put 4T1 data in Supplementary file in R1. The data from both CT26 and 4T1 cells from a new experiment, which was designed to better address the reviewer's question of the impact of Ivermectin (IVM) on chemotherapy response as detailed below, are now included in R2 as **Fig. 1J**. Accordingly, old **Fig. 1J** in R1 was deleted from R2 manuscript. The lower response in siRNA KD cells compared to KO cells is likely due to the less downregulation of iRhom1 expression in KD cells.

Regarding the 2nd question, we have conducted a new experiment with different concentrations of DOX and Ivermectin and used combination index (CI) to thoroughly evaluate the synergy between inhibition of p-HSP27 and chemotherapy. As shown in **Fig. 1J** in this R2 manuscript, inhibition of phosphorylation of HSP27 by IVM treatment led to an overall significant synergistic effect with a chemotherapeutic agent in WT 4T1 & CT26 cells. However, this synergistic effect was significantly attenuated in iRhom1 KO cell lines, suggesting that MAPK14-HSP27 axis likely plays a role in iRhom1-mediated drug resistance.

- Q8:** The included Western blots in Figure show a higher percentage of aggregated iRhom1 than monomeric iRhom1 (at least when probed with the anti-iRhom1 antibody). Since such aggregates are known to trigger the unfolded protein response, this could indirectly affect the phosphorylation of HSP27 or MAPK shown in the revised Figure 1G (the effect on MAPK14 on the Western blot is quite subtle). This experiment therefore also does not necessarily support the authors' interpretation.
- A8:** We appreciate the reviewer's previous suggestion of examining iRhom1 in high MW region. Indeed, as mentioned by the reviewer, we saw the iRhom1 aggregates in Western while it is unclear whether aggregates do exist in cells under natural condition. We have discussed the limitations of the iRhom1 OE experiments (**Page 21 line 22-23, Page 22 line 1-3**). However, we are confident about our proposed role of iRhom1/MAPK14/HSP27 axis as this was supported by various approaches in a number of experiments using 2 cell lines.
- Q9:** The revised Figure 2B shows a similar rescue effect of the control vector as the iRhom1 rescue vector. This reviewer attempted to understand this experiment (in the previous version, Figure 2B also showed a similar experiment in immunocompetent mice, in which a control vector experiment was similar to WT cells, with an "ns" added between the bars). This seemed to imply that the WT cells had been treated with a control vector, as opposed to a iRhom1 vector, but either way, this experiment does not make sense as presented. Since a rescue of iRhom1 KO cells with iRhom1 or a control was requested and this is presumably shown in the revised Figure 2B, the conclusion is that the control vector has a similar effect as the iRhom1 rescue.
- A9:** CRISPR gene editing and lentiviral transduction are known to potentially render the engineered cells more immunogenic. The use of control vector is to confirm that the retarded growth of KO tumor cells is not or not only attributed to the increased immunogenicity. The incomplete rescue of tumor growth of KO cells actually suggests that the KO cells may indeed become more immunogenic. We have elaborated this in the revised (R2) manuscript (page 8, line 9-11).
- Q10:** The authors did not perform the requested experiment, which was to include an iRhom2 rescue and perhaps a different multi-membrane-spanning protein, to show the effect is specific for iRhom1. The rescue experiments were to be performed in the iRhom1 ko cells.
- A10:** We apologized for having not done this experiment during R1 revision. We re-expressed the GFP-iRhom2 in iRhom1 KO cells (**Fig. S11**) as requested by the reviewer in R2

manuscript. Our data showed that iRhom2 rescue did not have significant impact on the enhanced Ag presentation seen in iRhom1 KO cells. The new data were included in the R2 manuscript as **Fig. S11**.

Q11: Overall, the discussion is improved. There is some discussion of the role of iRhoms in regulating ADAM17 and some caveats regarding alternative interpretations are included. However, the statement on page 19, lines 13 – 15, that “iRhom2 has more restricted distribution and exclusive functions in bone marrow and immune cells” is not correct, since the expression of iRhom1 and iRhom2 overlap in all tissues except in most immune cells, where little, if any iRhom1 is expressed, and in brain, where very little, if any iRhom2 is expressed in mice (except in microglial cells). In this reviewer’s opinion, the many different conclusions in this manuscript are not sufficiently supported by the results, so that publication in a high quality journal such as Nature Communications cannot be recommended by this reviewer.

A11: We appreciate the reviewer’s suggestion and have further revised the “Discussion” section accordingly (**page 19, line 16-23**). We have also deleted the bioinformatic data related to the comparative role of iRhom1 and iRhom2 (**Fig. S24** in R1), considering the very limited experimental data related to iRhom2 in our study.

Reviewer #2

Q1: Based on the BioPlex Network, MAPK14 is a candidate that likely interacts with iRhom1, and the MAPK14/HSP27 axis could play a key role in the drug resistance in certain types of cancer. However, as the authors noted in reference 53, the PI3K/AKT may be another pivotal pathway involved in drug resistance which is also supported by the data in the previous submission (Fig. 1F and 1G). It would thus be more meaningful to further clarify the contribution of these biological pathways, rather than just removing the previous data in the revised manuscript.

A1: We didn’t remove the data; the data was shown in **Fig. S3** in R1 and now **Fig. S3** in R2. In addition, our new data (**Fig. S23** in R2 and shown below) suggest that the iRhom1/MAPK14/HSP27 is regulated, at least in part by a mechanism that is independent of AKT signaling. We feel that detailed study on this is beyond the scope of this manuscript and will be further pursued in the future.

Changes in MAPK-HSP27 after inhibition of PI3K-AKT pathway by AKT inhibitor MK2206 in WT and iRhom1 KO cells.

Editorial note: with respect to your reply to Reviewer #4 original requests, Reviewer #2 is suggesting to provide better quality bright field images for (original) Figure 2F rather than deleting these data.

Answer to Editorial note:

We have generated images with improved quality, showing clearly the enhanced T-cell mediated toxicity (arrow pointed) in iRhom1 KO cell line (**Fig. S12** in R2).

REVIEWERS' COMMENTS

Reviewer #1 (Remarks to the Author):

Please find a list of remaining concerns outlined below under Q9R and Q10R.

Q9: The revised Figure 2B shows a similar rescue effect of the control vector as the iRhom1 rescue vector. This reviewer attempted to understand this experiment (in the previous version, Figure 2B also showed a similar experiment in immunocompetent mice, in which a control vector experiment was similar to WT cells, with an “ns” added between the bars). This seemed to imply that the WT cells had been treated with a control vector, as opposed to a siiRhom1 vector, but either way, this experiment does not make sense as presented. Since a rescue of iRhom1 KO cells with iRhom1 or a control was requested and this is presumably shown in the revised Figure 2B, the conclusion is that the control vector has a similar effect as the iRhom1 rescue.

A9: CRISPR gene editing and lentiviral transduction are known to potentially render the engineered cells more immunogenic. The use of control vector is to confirm that the retarded growth of KO tumor cells is not or not only attributed to the increased immunogenicity. The incomplete rescue of tumor growth of KO cells actually suggests that the KO cells may indeed become more immunogenic. We have elaborated this in the revised (R2) manuscript (page 8, line 9-11).

Q9R: The answer provided by the authors does not address the original Q9, but instead reinforces the concerns raised about this experiment. The control vector in Figure 2B, middle panel, has a similar, if not stronger effect on tumor formation of iRhom1 KO CT26 cells than the rescue with iRhom1 in the right panel. These findings suggest that the loss of tumor formation in immunocompetent mice inoculated with iRhom1 KO CT26 cells is not necessarily due to loss of iRhom1 but could also be due to other off target mutations caused by CRISPR gene editing. Alternatively, the rescue vector could increase immunogenicity independently of the expressed gene. Either way, this experiment does not support the authors' conclusions regarding the role of iRhom1 in tumor formation in mice.

Q10: The authors did not perform the requested experiment, which was to include an iRhom2 rescue and perhaps a different multi-membrane-spanning protein, to show the effect is specific for iRhom1. The rescue experiments were to be performed in the iRhom1 ko cells.

A10: We apologized for having not done this experiment during R1 revision. We re-expressed the GFP-iRhom2 in iRhom1 KO cells (Fig. S11) as requested by the reviewer in R2 manuscript. Our data showed that iRhom2 rescue did not have significant impact on the enhanced Ag presentation seen in iRhom1 KO cells. The new data were included in the R2 manuscript as Fig. S11.

Q10R The request was to include an iRhom2 rescue in addition to an iRhom1 rescue and perhaps a different multi-membrane-spanning protein, to show the effect on tumorigenesis was specific for iRhom1, in other words, to perform the same type of experiments as in Figure 2 B. Since the interpretation of the current Figure 2B is questionable for the reasons outlined above, it is not clear that the observed rescue effect depends on iRhom1 in the first place. Nevertheless, the requested experiment was an additional control performed like those in Figure 2B.

The overall assessment of this manuscript remains unchanged. In this reviewer's opinion, the many different conclusions in this manuscript are not sufficiently supported by the results, so that publication in a high quality journal such as Nature Communications cannot be recommended.

We would like to thank reviewer 1 again for spending time reviewing our manuscript again and for the comments. The following are our responses to each specific comment of the reviewer:

Response to Reviewer #1:

Q1: The answer provided by the authors does not address the original Q9, but instead reinforces the concerns raised about this experiment. The control vector in Figure 2B, middle panel, has a similar, if not stronger effect on tumor formation of iRhom1 KO CT26 cells than the rescue with iRhom1 in the right panel. These findings suggest that the loss of tumor formation in immunocompetent mice inoculated with iRhom1 KO CT26 cells is not necessarily due to loss of iRhom1 but could also be due to other off target mutations caused by CRISPR gene editing. Alternatively, the rescue vector could increase immunogenicity independently of the expressed gene. Either way, this experiment does not support the authors' conclusions regarding the role of iRhom1 in tumor formation in mice.

A1: The figure on the far right is a new experiment conducted during R2 revision to address the rescue effect. Unlike the KO cells that received lentiviral transduction once, the rescued cells received lentiviral transduction twice to re-express iRhom1. There was a significant increase in tumor growth after rescue. The incomplete rescue is likely due to enhanced immunogenicity especially in cells that are treated with lentiviral vector twice. This has been reported in the literature and we have cited the references.

We have been following standard KO and rescue protocols and we think the reviewer may have misinterpreted "Control vector group" as the **KO** cells (which are actually **WT** cells) transduced with a blank lentiviral vector and then compared the effect of control vector with the iRhom1 rescue effect in KO cells in the figure on the far right. The reviewer might have also misinterpreted "WT cells had been treated with a control vector, as opposed to a iRhom1 vector". This might have led the reviewer to the conclusion that the rescue effect was not specific to iRhom1 and was comparable to that of a control vector. We apologize that we did not provide more details in the manuscript and have provided more information in the Methods section about how these cell lines were generated (**Page 26, line 18-22**).

Q2: The request was to include an iRhom2 rescue in addition to an iRhom1 rescue and perhaps a different multi-membrane-spanning protein, to show the effect on tumorigenesis was specific for iRhom1, in other words, to perform the same type of experiments as in Figure 2 B. Since the interpretation of the current Figure 2B is questionable for the reasons outlined above, it is not clear that the observed rescue effect depends on iRhom1 in the first place. Nevertheless, the requested experiment was an additional control performed like those in Figure 2B.

A2: The major conclusion from **Fig. 2** is that iRhom1 negatively regulates ERAP1-mediated antigen processing and presentation through facilitating ERAP1 degradation. At the request of the 1st reviewer we have done a few experiments suggesting that iRhom2 does not play a role in iRhom1-ERAP1-mediated antigen presentation as: 1) there are no changes in the expression levels of iRhom2 in iRhom1 KD or KO cells (**Supplementary Fig. 6**); 2) KD of iRhom2 had no impact on the expression levels of ERAP1 (**Supplementary Fig. 15**); and 3) iRhom2 rescue showed no impact on the enhanced antigen presentation in iRhom1 KO cells (**Supplementary Fig. 11**).

We feel that the addition of another control of iRhom2 rescue in iRhom1 KO in **Fig. 2B** does not add to our conclusion about the role of iRhom1 in regulating immune response. It is possible that iRhom2 may also be involved in regulation of immune response but that deserves a separate study. Our data (as requested by the reviewer) suggests that iRhom2 is not involved in antigen presentation in our B16-OVA model, which is different from the effect of iRhom1. Again, the request for more control

experiments may also come from the misinterpretations of the different groups and the reviewer's disbelief that the rescue effect was specific to iRhom1. We hope that our explanations/responses are acceptable to reviewer 1 and the editor.